



# Ocean biogeochemistry in the Norwegian Earth System Model version 2 (NorESM2)

Jerry F. Tjiputra[1], Jörg Schwinger[1], Mats Bentsen[1], Anne L. Morée[2], Shuang Gao[3], Ingo Bethke[2], Christoph Heinze[2], Nadine Goris[1], Alok Gupta[1], Yanchun He[4], Dirk Olivié[5], Øyvind Seland[5], and Michael Schulz[5]

[1]NORCE Norwegian Research Centre and Bjerknes Centre for Climate Research, Bergen, Norway
[2]Geophysical Institute, University of Bergen and Bjerknes Centre for Climate Research, Bergen, Norway
[3]Institute of Marine Research and Bjerknes Centre for Climate Research, Bergen, Norway
[4]Nansen Environmental and Remote Sensing Centre, Bergen, Norway
[5]Norwegian Meteorological Institute, Oslo, Norway

**Correspondence:** Jerry Tjiputra (jerry.tjiputra@norceresearch.no)

**Abstract.** The ocean carbon cycle is a key player in the climate system through its role in regulating atmospheric carbon dioxide concentration as well as other processes that alter the Earth's radiative balance. In the second version of the Norwegian Earth System Model (NorESM2), the oceanic carbon cycle component has gone through numerous updates that include, amongst others, improved process representations, increased interactions with the atmosphere, and additional new tracers.

Oceanic dimethyl sulfide (DMS) is now prognostically simulated and its fluxes are directly coupled with the atmospheric component, allowing for a direct feedback to the climate. Atmospheric nitrogen deposition and additional external inputs of other biogeochemical tracers through riverine are recently included in the model. The implementation of new tracers such as 'preformed' and 'natural' tracers enables a separation of physical from biogeochemical drivers as well as of internal from external forcings and hence a better diagnostic of the simulated biogeochemical variability. Carbon isotope tracers have been imple-

mented and will be relevant for studying long-term past climate changes. Here, we describe these new model implementations and present the evaluation of the model's performance in simulating the observed climatological states of water column biogeochemistry as well as in simulating the transient evolution over the historical period. Compared to its predecessor NorESM1, the new model's performance has improved considerably in many aspects. In the interior, the observed spatial patterns of nutrients, oxygen, and carbon chemistry are better reproduced, reducing the overall model biases. A new set of ecosystem

parameters and improved mixed layer dynamics improves the representation of upper ocean processes (biological production and air-sea $CO_2$ fluxes) at seasonal time scale. Transient warming and air-sea $CO_2$ fluxes over the historical period are also in good agreement with observation-based estimates. NorESM2 participates in the Coupled Model Intercomparison Project phase 6 (CMIP6) through DECK (Diagnostic, Evaluation and Characterization of Klima) and several endorsed MIP-simulations.

## 1 Introduction

Up to the early 1990s, climate models consisted only of physical atmospheric general circulation models (AGCMs) with prescribed ocean surface state variables or simplified ocean modules (swamp ocean, slab ocean). As the ocean is a huge





reservoir and absorber of heat and the greenhouse gas $CO_2$ (carbon dioxide), an expansion of climate models to include ocean components and a representation of the carbon cycle became a necessity. Coupled atmosphere-ocean-land models (now referred to as Earth System Models - ESMs) were eventually developed to account for further reservoirs and feedbacks including biogeochemical cycles (Bretherton, 1985; Cubasch et al., 2013; Heinze et al., 2019). The inclusion of the ocean carbon cycle is

a first order requirement, because on timescales beyond a few decades, the ocean becomes the sole major sink for atmospheric excess $CO_2$ from anthropogenic emissions (Sarmiento and Gruber , 2006). This allows us to estimate historical carbon budgets as well as future carbon emission pathways under specified scenarios.

Today, modern ESMs are state-of-the-art tools that allow us to study the complex interactions and feedbacks between various components in the Earth system in a comprehensive way (Flato et al., 2013). They are applied regularly to simulate climate

evolutions across time scales and to study transient climate change, its drivers, and its impact on the environment (e.g., Bopp et al., 2013; Gehlen et al., 2014). By prescribing plausible scenarios of future emissions and land use, these models provide projections for possible future climate states. ESMs are typically upgraded every few years with new and improved process representations as well as adaptations to technical advancements (e.g., new hardware system, higher resolutions, etc.).

The ocean and its biogeochemistry play a crucial role in controlling the rate of anthropogenic climate change through a

substantial negative feedback (Friedlingstein et al., 2006; Arora et al., 2013). However, the absorption rates of heat and carbon by the ocean are non-linear in space and time, due to the complex interplay of internal climate variability (e.g., Tjiputra et al., 2012; Schwinger et al., 2014). Once passed the air-sea interface, the dissolved carbon dioxide reacts with seawater and is chemically transformed into different carbonate species. The ocean circulation and biological processes sequester parts of these into the deep ocean, where they stay isolated from the atmosphere, for decades to millennia (Sarmiento and Gruber ,

2006). In addition to carbon dioxide, the ocean biogeochemistry also influences the Earth's climate through other seawater chemical constituents, such as through dimethyl sulfide emissions. This could result in a number of feedback mechanisms within the Earth system that can either amplify or dampen the rate of climate change (Rhein et al., 2013). ESM projections are therefore critical for narrowing uncertainties in the future global carbon budget, and consequently its climate feedback.

ESMs have been increasingly used to estimate the evolution of marine ecosystem stressors, such as ocean acidification and

deoxygenation, under anthropogenic climate change (e.g., Gehlen et al., 2014; Tjiputra et al., 2018). ESM applications to support climate policy-making, such as quantifying future compatible anthropogenic carbon emissions under a set of predefined scenarios as well as addressing emerging questions associated with the United Nations sustainable development goals have become more regular. With its growing relevance, a more realistic representation of the observed contemporary ocean biogeochemistry in Earth system models is urgently necessary to increase the fidelity of future projections.

In this manuscript, we present improvements in the ocean biogeochemical component of the Norwegian Earth System Model (NorESM). This component is based on the Hamburg Oceanic Carbon Cycle model (HAMOCC5.1), which was originally developed by Maier-Reimer (1993) and has gone through several development iterations (e.g., Six and Maier-Reimer, 1996; Maier-Reimer et al., 2005). At the Bjerknes Centre for Climate Research in Norway, the model has been branched off, further developed (Assmann et al., 2010; Tjiputra et al., 2010, 2013; Schwinger et al., 2016), and coupled to the Bergen Layered

(isopycnic coordinate) Ocean Model (BLOM-iHAMOCC). Within the CMIP5 experiments, we have identified several short-





comings in the previous model version of NorESM (NorESM1-ME, hereafter referred to as NorESM1). Here, we discuss several developments that have been prioritized for the ocean biogeochemistry and are included in the current model version (NorESM2), which will contribute to CMIP6.

NorESM1 does not include interactively coupled DMS emissions from the ocean, which is the largest natural source of

atmospheric sulfur, and contributes to aerosol and cloud formation. Instead, the atmospheric chemistry module of NorESM1 prescribed mean fixed climatology oceanic fluxes of DMS, and there is no feedback associated with climate-induced change in marine biological production. However, a model study has suggested that the inclusion of interactive DMS in future climate projection could potentially result in spatially-varying changes in warming rate and a non-negligible feedback on the climate system (Schwinger et al., 2017). This has prompted us to include a fully-interactive DMS cycle (production, emissions, and

radiative impact) into NorESM2.

In the past years, the concept of emergent constraints was employed to identify key biogeochemical processes that lead to uncertanties in ESMs projections (e.g., Wenzel et al., 2014; Kwiatkowski et al., 2017). For instance, biases in the seasonal cycles of biological production and surface ocean $pCO_2$ have been identified as one of the factors contributing to the uncertainty in the projected carbon sinks and storage in the ESMs participating in CMIP5 (Kessler and Tjiputra, 2016; Goris et al., 2018).

These studies motivate further improvement in the representation of biological processes, particularly focusing on the high latitude regions, such as the Southern Ocean. The biological improvements in NorESM2 were mainly achieved through tuning of the poorly constrained parameters in the ecosystem module.

In NorESM1, as well as other ESMs participating in CMIP5, there is a large uncertainty in the simulated interior oxygen concentration when compared to observations. This is partly attributed to the short model spin-up and bias in the interior

remineralization processes (Cabré et al., 2015; Séférian et al., 2016). To alleviate this, we have tested several parameterizations of particulate organic carbon (POC) sinking schemes in our model (Schwinger et al., 2016). Currently, there are three different sinking schemes that can be selected in NorESM2. Based on our earlier assessments, we have used the scheme with sinking velocity that increases linearly with depth as the default parameterization in NorESM2.

Increasing atmospheric $CO_2$ and associated climate change will alter both the natural and anthropogenic components of

ocean inorganic carbon chemistry. Disentangling future dynamical responses of these components separately is therefore needed to identify which processes (e.g., biological and physical, among others) regulate the simulated changes in ocean carbon storage today and in future. In NorESM1, only a single dissolved inorganic carbon (i.e., total DIC) tracer was implemented, and there is no suitably accurate method to separate between the driving mechanisms of its changes. In NorESM2, we have introduced a new 'natural DIC' tracer, which simulates changes in the natural component of DIC (i.e., it only interacts

with fixed preindustrial $CO_2$ concentration during the air-sea gas exchange). Other key diagnostic tracers such as preformed phosphate, oxygen, and alkalinity have also been implemented, all of which allow us to better elucidate mechanisms driving the simulated ocean carbon chemistry changes (Bernardello et al., 2014). These tracers also allow to determine interior ocean circulation-induced changes in biogeochemical tracers.

Nutrients are important constraints of ocean primary productivity and hence atmospheric carbon sinks. In NorESM1, the nu-

trient cycle was assumed to be an approximately closed system with a long-term loss due to sedimentation. The only external





source is through atmospheric nitrogen fixation. Implementation of riverine input of nutrients in HAMOCC5.1 was initiated by Bernard et al. (2011) and a study by Gao et al. (in prep.) indicates that riverine nutrient inputs could improve the regional representation of marine biogeochemistry, with implications on future projections, i.e., they increase coastal carbon sinks and alleviate nutrient limitations due to warming-induced stratification. In NorESM2, we have accounted for riverine inputs of nu-

trients and other biogeochemical constituents. Other processes representing the external nitrogen sources such as atmospheric nitrogen deposition and nitrogen fixation have also been implemented and updated, respectively.

Since ESMs are used to project future climate change, questions have arisen whether or not these models are able to simulate past climate change. To this end, NorESM1 has been applied to study the sensitivity of climate states to different boundary conditions in the past, going back from the last glacial to as far back as the mid-Pliocene (Zhang et al., 2012; Guo et al., 2019).

The model was also used to study the sensitivity of the ocean carbon cycle to different background climates, which could provide insights on the ocean role in regulating past atmospheric $CO_2$ variability (Kessler et al., 2018; Luo et al., 2018). There has also been a growing interest to determine if such complex models are able to reproduce climate states directly inferred from proxy-records. These paleo proxies, e.g., carbon isotopes ($\delta^{13}C$ and $\Delta^{14}C$), typically measured from foraminiferal samples collected from seafloor sediment, are natural archives that store watermass properties and are used to infer large scale ocean

circulation patterns as well as ventilation time scales during past climate changes (Peterson et al., 2014). In NorESM2, we have now implemented $^{13}C$ and $^{14}C$ tracers, which will be used as an additional ocean circulation and biogeochemistry constraint when simulating past climate variability.

Beyond the above mentioned biogeochemical processes, other improvements in tracer initialization, iron chemistry, air-sea gas exchange parameterization, code readability, and documentations of the codes have been made. Moreover, there have been

several updates within the physical ocean component of NorESM2 and a selection of these updates are discussed in this paper as they have direct impacts on the ocean biogeochemistry. In addition to describing the model improvements, we also evaluate the performance of the ocean biogeochemical component of NorESM2. Whenever possible, we compare and describe the performance of NorESM2 model with the first generation model, NorESM1.

The manuscript is organized as follows: advancements in the physics and improvements in biogeochemical processes are

described in Section 2. Implementations of new tracers are presented in Section 3. The different model simulations performed and used in the paper are described in section 4. In Section 5, we present results from our coupled ESM simulations as well as that of the ocean-only (preindustrial) simulation to illustrate the performance of the simulated carbon isotope tracers. Summary and discussion are presented in section 6.

## 2 Model changes and improvements

### 2.1 General configuration changes

In NorESM2, the ocean-ice model adopts a tripolar grid instead of the bipolar grid of NorESM1. The tripolar grid was chosen because it is more isotropic at northern high latitudes and therefore allows for a more efficient time integration (longer time step) (Guo et al., 2019). Compared to NorESM1, NorESM2 has enabled a higher frequency ocean coupling to resolve the





diurnal cycle in the flux- and state-exchange between ocean and atmosphere (i.e., hourly instead of daily atmosphere-ocean coupling). We also now applied a full leap-frog time-stepping in both physical and ocean biogeochemical models to improve conservation of biogeochemical tracers (Schwinger et al., 2016).

There are several configuration options for NorESM2. For CMIP6, two configurations will mostly be used, differing only

in the horizontal resolution of the atmospheric and land models with otherwise identical model components. They are named NorESM2-LM and NorESM2-MM and have atmosphere-land resolution of approximately 2° and 1°, respectively. In addition, both versions have a different parameter tuning in the atmospheric component, necessary to achieve a top of the atmosphere radiative balance under preindustrial conditions. The ocean biogeochemistry performs similarly in both model versions. Unless explicitly specified, our analysis refers to the results from NorESM2-MM (hereafter referred to as NorESM2).

**2.2   Physical parameterizations**

The simulated Atlantic Meridional Overturning Circulation (AMOC) strength in NorESM1 is on the strong side (30.8 Sv at 26.5°N; 1 Sv = $10^6$ m$^3$ s$^{-1}$) compared to observation-based estimates and other CMIP5 models (Bentsen et al., 2013). The vigorous AMOC leads to a too warm and saline Atlantic Ocean at depth. Lack of upper ocean stratification at high latitudes is thought to contribute to this AMOC bias. Reformulating the eddy-induced transport (Gent and McWilliams, 1990) in order

to make it more efficient in the upper ocean non-isopycnic regime of the model and adjustments to the parameterized re-stratification by mesoscale eddies (Fox-Kemper et al., 2008) lead to improved high latitude stratification and a more realistic MLD, particularly at high latitudes.

NorESM1 has a cold bias in the depth range 200–1000 m between 50°S and 50°N (Bentsen et al., 2013) that could be related to insufficient upper ocean vertical mixing. Further, the new hourly ocean coupling and associated tuning of MLD

parameterizations lead to a cold bias in the Pacific equatorial thermocline that also indicates too little vertical mixing.

To improve this, two different approaches have been used. First, we allow surface turbulent kinetic energy (TKE) originating from wind stirring to penetrate below the mixed layer and be added to the TKE reservoir of the $k-\varepsilon$ turbulence model handling the shear instability mixing of the isopycnic interior. Secondly, with hourly ocean coupling, high frequency winds are available to parameterize wind work on near-inertial motions. Inspired by the approach of Jochum et al. (2013), a fraction of this energy

source is added to the mixed layer TKE reservoir, modifying the MLD estimation, while a fraction of the remaining energy is used to drive upper ocean mixing through assumed excitation and breaking of internal waves. Combining these approaches, we were able to reduce the above-mentioned upper ocean biases significantly. The mid-depth warm bias seen in NorESM1 is also reduced. More details on the physical improvements are described in Bentsen et al. (in prep.). As a result of these improvements, NorESM2 simulates a more reasonable AMOC strength of approximately 21 Sv, which is closer and within the

range of the observational estimates of 17.9±3.3 Sv (Srokosz et al., 2012).





## 2.3 Dimethyl sulfide

NorESM2 prognostically simulates DMS concentration according to the formulation of Six and Maier-Reimer (2006), as follows:

$$\frac{\partial DMS}{\partial t} = ADV(DMS) + S_{Phy} - S_{bac}DMS - S_{photo}DMS - S_{gas}, \tag{1}$$

where the first term on the right hand side denotes the physical transport, the second term denotes production due to phytoplankton degradation, the third, fourth, and fifth terms denote sinks due to bacterial activity, photolysis, and flux to the atmosphere. The production term depends on the temperature, pH, and opal and $CaCO_3$ (implicitly computed as a function of silicate concentration) productions:

$$S_{Phy} = (\gamma_d Phy_{dia} + \gamma_c Phy_{cal}) \cdot (1 + \frac{1}{(T+10)^2}) \cdot (1 + (pH - pH_{pi}) \cdot \alpha). \tag{2}$$

$\gamma_d Phy_{dia}$ and $\gamma_c Phy_{cal}$ represent production rates of biogenic opal and $CaCO_3$ within the mixed layer depth. The last term in Eq. 2 describes a pH-dependence of DMS production following (Six et al., 2013), which is parameterized based on the deviation of actual pH from the climatological monthly mean surface pH ($pH_{pi}$) calculated from the preindustrial control. In NorESM2, the atmospheric model receives the DMS emissions ($S_{gas}$) that are prognostically computed by the ocean biogeochemistry model. Due to uncertainties in the pH-control on DMS production, we have omitted the pH-dependency term by

setting $\alpha$ to zero in the presented simulations. Parameters $\gamma_d$ and $\gamma_c$ are set to 0.025 and 0.125, respectively, such that DMS production is more sensitive to calcifier production in the model. This assumption is due to the fact that haptophyte-class phytoplankton generally has a higher DMSP to cell carbon ratio than diatoms (Keller et al., 1989). Since the diatom and calcifier portions of phytoplankton are implicitly computed as a function of silicate concentration, e.g., diatom (calcifier) are high (low) when surface silicate is abundant and vice versa, the DMS production is also sensitive to the long-term trend in the silicate

concentration.

## 2.4 Riverine input

The influx of carbon and nutrients from over 6000 rivers to the coastal oceans has been implemented based on previous work of Bernard et al. (2011) with modifications. Bernard et al. (2011) have implemented the riverine fluxes based on the levels in year 1995, provided by an early version of Global-NEWS model (Seitzinger et al., 2005). In addition to the riverine DIC,

dissolved organic carbon (DOC), inorganic nitrogen and phosphorus, particulate organic carbon (POC), particulate nitrogen and phosphorus, and dissolved silicate, we have also included riverine alkalinity (ALK) and iron (Fe). Except for DIC, ALK and Fe, all data are provided by the more recent Global-NEWS2 model (Mayorga et al., 2010), which is a hybrid of empirical, statistical and mechanistic model components that simulate steady-state annual riverine fluxes of carbon and nutrients. The DIC and ALK fluxes are taken from the work by Hartmann (2009). Riverine Fe-flux is calculated as a proportion of global

gross dissolved Fe input of 1.45 Tg yr$^{-1}$ (Chester, 1990), weighted by water runoff of each river. Only 1% of the riverine dissolved Fe is added to the oceanic dissolved Fe, since approximately 80 to 99% of the fluvial gross dissolved Fe is removed





during estuarine mixing (e.g., Sholkovitz et al., 1981; Shiller and Boyle, 1991; Bruland and Lohan, 2014). Instead of releasing the riverine fluxes to the nearest ocean grid cell (Bernard et al., 2011), we have interpolated all fluxes at river mouths in the same way as the freshwater runoff (distributed as a function of river mouth distance with e-folding length scale of 1000 km and cutoff of 300 km) to the ocean grid. The fluxes are specified to be constant over time at contemporary levels (year 2000). In the model, carbon and nutrients are taken up by phytoplankton according to a Redfield ratio (C:N:P=122:16:1). Therefore, the least abundant riverine organic matter (both dissolved and particulate species) is used to calculate the riverine fluxes of DOC and POC (detritus, particulate organic matter) in the model. Any excess riverine organic matter following the Redfield ratio conversion is merged into its inorganic form (see also Bernard et al., 2011).

### 2.5 Atmospheric nitrogen deposition

We apply anthropogenic nitrogen deposition fluxes that potentially affect the simulated ocean carbon uptake (through an increase of the available nitrate for biological production) to the ocean biogeochemical model. The monthly input fields, spanning the years 1850-2014, are simulated by chemistry transport models and provided by CMIP6 (Jones et al., 2016). Conservative remapping is used to interpolate the input data from a regular $2.5° \times 1.9°$ latitude-longitude grid to the tripolar ocean model grid of NorESM2. Four species of wet or dry and oxidized or reduced nitrogen deposition rates are included in the input fields. All of them are assumed to be bio-available and added to the nitrate pool in the top-most ocean layer.

### 2.6 Particle exports

In NorESM1, the export of particulate organic matter is parameterized with a constant sinking speed and a constant remineralization rate at depth, when sufficient oxygen is available. This simplistic formulation has been shown to have difficulty in accurately representing the observed particle fluxes at depths (Kriest and Oschlies, 2008) and leads to biases in the interior distributions of biogeochemical tracers (e.g., as seen from remineralized phosphate concentrations). In NorESM2, we have developed two additional particle flux parameterizations that can be selected. They are referred to as WLIN and AGG schemes. The WLIN scheme is similar to the standard scheme but the sinking speed is a linear function of depth: $w_{POC} = \min(w_{min}+az, w_{max})$. Here, we use $w_{min} = 1\ md^{-1}$, $w_{max} = 60\ md^{-1}$, and $a = 60/2400\ md^{-1}m^{-1}$. When $w_{min}$ and $w_{max}$ are set to zero and infinity, respectively, this scheme is equivalent to the widely used Martin-curve formulation (Martin et al., 1987).

The AGG scheme implements a prognostic sinking speed, calculated according to a size distribution of sinking particles. The total concentration of particulate matter, formulated as a function of phytoplankton and detritus, forms sinking aggregates with explicitly computed size distribution that follows a power-law formula (Kriest and Evans, 1999; Kriest, 2002):

$$n(d) = Ad^{-\beta}, \quad l < d < \infty. \tag{3}$$

Here, $d$ represents the particle diameter and $A$ and $\beta$ represent parameters of the size distribution. The minimum particle diameter $l$ is set to 0.002 cm. The sinking speed of aggregates is computed according to the diameter of each aggregate. More details of the implementation of the AGG formulation are described in Schwinger et al. (2016). Based on the performance of the different schemes, we have set WLIN to be the default scheme and it is used in all experiments presented here. More details





on the differences in the simulated interior biogeochemistry in the different particle export schemes are available in Schwinger et al. (2016).

## 2.7 Nitrogen fixation

Nitrogen fixation by cyanobacteria is computed implicitly and directly converted into nitrate concentration. In NorESM1, nitrogen fixation is implemented such that it can occur anytime and anywhere in the surface ocean as long as the nitrate concentration is lower than phosphate (multiplied by the stoichiometric nitrate to phosphate ratio $R_{N:P}$) following Maier-Reimer et al. (2005). However, Breitbarth et al. (2007) provided some evidence that the large scale distribution of *Trichodesmium*, a type of cyanobacteria, is well constrained by seawater temperature between 20 and 30°C. Based on this, a temperature-dependent function $f(T)$ according to Kriest and Oschlies (2015) has been added to the nitrogen fixation formulation in NorESM2. The rate of change of nitrate owing to nitrogen fixation is formulated as follows:

$$\frac{\partial NO_3}{\partial t} = f(T) \cdot \mu . max(0, R_{N:P} \cdot PO_4 - NO_3), \tag{4}$$

with

$$f(T) = max(0, \frac{-0.0042 \cdot T^2 + 0.2253 \cdot T - 2.7819}{0.2395}), \tag{5}$$

where $\mu$ is the maximum $N_2$ fixation rate (0.005 day$^{-1}$). In addition, for each mole of N fixed to nitrate, 1.25 and 1 moles of dissolved oxygen and alkalinity are consumed and lost in the surface layer. The corrected formula essentially limits the occurrence of N-fixation to warm low latitudes.

## 2.8 Air-sea gas exchange

In NorESM2, the air-sea gas exchange of $CO_2$, $O_2$, $N_2$, $N_2O$, DMS, CFC-11, CFC-12, and $SF_6$ is computed prognostically according to the updated formulation of Wanninkhof (2014). Here, the air-sea flux $F$ of gas $X$ is computed as a function of surface wind speed $U$, Schmidt number $Sc$, gas solubility $K_o$, and partial pressure difference of gas $X$, based on a bulk formulation:

$$F_X = 0.251 \cdot U^2 \cdot (Sc/660)^{-0.5} \cdot K_o \cdot (pX_{sw} - pX_{air}). \tag{6}$$

The net fluxes are computed for the top layer of the ocean model (i.e., 10 m depth). The updated formulation includes a refitted temperature-dependent Schmidt number that can be applied for a temperature range of -2 to 40°C. The solubility of all gases is computed as a function of surface temperature and salinity following Weiss (1970) for $O_2$ and $N_2$, Weiss and Price (1980) for $CO_2$ and $N_2O$, Warner and Weiss (1985) for CFC-11 and CFC-12, and Bullister et al. (2002) for $SF_6$. The flux of DMS is only unidirectional (outgassing).

In the historical simulation, we use the annual atmospheric concentrations of CFC-11, CFC-12, and $SF_6$, divided into northern and southern hemispheric values, according to Bullister (2014). For the atmospheric $CO_2$ concentration, monthly global-mean values from the CMIP6 dataset are used. In the preindustrial simulation, a constant $CO_2$ concentration of 284.32 ppm is used.



## 2.9 Dissolved iron parameterization

Adjustments in the iron parameterization have been made to ensure that NorESM2 simulates the observed iron-limited primary production in the Southern Ocean, Equatorial and North Pacific. In the model, the consumption and release of dissolved iron associated with biological activities are determined using a fixed stoichiometry ratio ($R_{Fe:P}$ = 6.1e-4 mol Fe mol P$^{-1}$, previously 3.7e-4 mol Fe mol P$^{-1}$). In the surface layer, a constant fraction (3.13e-6 kmol Fe kg$^{-1}$ day$^{-1}$, previously 6.27e-6 kmol Fe kg$^{-1}$ day$^{-1}$) of aerial dust deposition (Mahowald et al., 2005) is instantaneously converted into bio-available iron. The updated global input of bio-available iron from the atmosphere is 2.8 Gmol Fe yr$^{-1}$, which is within the range of values used by other models (Tagliabue et al., 2016). Finally, throughout the water column, complexation of iron by organic substances (ligands) is assumed:

$$\frac{\partial Fe_{cmpl}}{\partial t} = -\lambda_{Fe} \cdot max(0, Fe - Fe_O) \tag{7}$$

where the strength if this complexation $\lambda_{Fe}$ is set to 0.05/365 (previously 0.005/365) day$^{-1}$ and the threshold $Fe_O$ has been reduced from 0.6 to 0.5 nmol Fe m$^{-3}$. The latter is motivated by the fact that the observed iron concentration in the deep Southern Ocean is lower than 0.6 nmol Fe m$^{-3}$ (Boyd and Ellwood, 2010). The higher complexation rate allows for faster relaxation toward this value.

## 2.10 Ecosystem parameterization updates

The underlying marine ecosystem parameterization in NorESM2 remains the same as in NorESM1, but many of the parameters have been adjusted to reduce biases that are present in NorESM1. An updated schematic diagram of the marine ecosystem module in NorESM2 is presented in Fig. 1. The main deficiencies of the spatial annual mean primary productivity pattern in NorESM1 are a too strong primary production (PP) in the Southern Ocean, the Eastern Tropical Pacific, and to a lesser degree the North Atlantic, contrasted by a very low PP in the subtropical gyres (most pronounced in the Pacific). The high productivity in the high latitudes is accompanied by an exaggerated annual cycle of PP showing a too strong spring bloom and a too fast decline of PP afterwards. Tuning of the ecosystem parameterization during the development of NorESM2 was focused on improving these regional shortcomings. The changes in ecosystem parameters listed in Table 1 mainly serve two purposes: (i) to increase top-down limitation by zooplankton grazing during phytoplankton peak bloom (but not before and after), and (ii) to increase the fraction of nutrients that is routed through dissolved organic matter (DOM).

The first point is achieved through a reduction in zooplankton mortality as well as increases in the maximum grazing rate, assimilation efficiency, and half-saturation constant for zooplankton grazing. The latter point is mainly achieved by reducing the production of detritus by zooplankton (fecal pellets, $1 - \varepsilon_{zoo}$ of grazing) and increasing exudation and excretion rates. This allows, in combination with a decrease of the DOM remineralization constant, more nutrients to be laterally transported out of regions where nutrient trapping occurs, i.e., the Tropical Pacific.

Additionally, $CaCO_3$ and silicate to phosphate ratios were tuned to remove the high alkalinity bias of NorESM1 (see Schwinger et al., 2016, for details). Note that some adjustments of parameters given in Table 1 were also necessary because



of the new sinking scheme (increasing sinking speed with depth, see Section 2.6), and the different physical circulation fields between NorESM1 and NorESM2.

In NorESM1, biological productivity was computed only for the top 100 m depth, which presumably represents the averaged euphotic layer. In an isopycnic model, there is no static interface separating this depth level. In cases where the bulk MLD

extends below 100 m, we virtually split this at 100 m and the biological production is simulated only down to this interface. In NorESM2, we have omitted this virtual layer splitting and the biological production is allowed to occur below 100 m as long as it is within the bulk mixed layer depth and has sufficient light (attenuated with depth and chlorophyll concentration) for phytoplankton growth.

## 3  New tracers

### 3.1  Preformed tracers

Four new preformed tracers have been implemented in NorESM2, namely preformed dissolved oxygen ($O_2^{pre}$), phosphate ($PO_4^{pre}$), alkalinity ($ALK^{pre}$), and dissolved inorganic carbon ($DIC^{pre}$). They are initialized to zero during the spin-up. During the model integration, in the bulk mixed layer of the model (i.e., the top 2 levels), the preformed tracers are set to the respective total, non-preformed values at each time step. Below the mixed layer, the preformed tracers are advected as passive

tracers by the physical processes and hence are a measure of transport-induced (e.g., circulation, ventilation, etc.) changes. The preformed tracers are included for diagnostic purposes, such as identifying the sources of model-data misfits (Duteil et al., 2012). The preformed oxygen can be used to quantify the total and apparent oxygen utilizations (TOU and AOU) as well as $O_2$ disequilibrium ($\Delta O_2$) in the interior ocean (Eqs. 8-9; Ito et al., 2004):

$$TOU = O_2^{pre} - O_2, \tag{8}$$

and

$$\Delta O_2 = TOU - AOU = O_2^{pre} - O_2^{sat}. \tag{9}$$

Here, the saturated oxygen ($O_2^{sat}$) is determined as a function of temperature and salinity (Garcia and Gordon, 1992). Due to its closed coupling with ocean circulation, the preformed oxygen can be used to separate biologically- from physically-induced biases in the simulated interior biogeochemical tracers.

Both preformed phosphate and alkalinity can be used to quantify the organic (soft tissue, $DIC^{soft}$) and inorganic (carbonate, $DIC^{carb}$) biologically-mediated carbon pump (Eqs. 10-12; Bernardello et al., 2014). We use a stoichiometry ratio of $P:N:C = 1:16:122$ in the model, such that $DIC^{soft}$ represents remineralized particulate and dissolved organic materials and $DIC^{carb}$ dissolution of particulate calcium carbonate in the water column:

$$DIC^{tot} = DIC^{pre} + DIC^{soft} + DIC^{carb}, \tag{10}$$



$$DIC^{soft} = r_{C:P}(PO_4^{tot} - PO_4^{pre}), and \tag{11}$$

$$DIC^{carb} = 0.5.[ALK^{tot} - ALK^{pre} + (r_{N:P} + 1) \cdot (PO_4^{tot} - PO_4^{pre})]. \tag{12}$$

The difference between total $DIC$ and biologically altered DIC ($DIC^{soft}$+$DIC^{carb}$, or the biological carbon pump) therefore represents the physical carbon pump ($DIC^{pre}$), which comprises saturated and disequilibrium components (Eq. 13). Here, a tracer of saturated DIC ($DIC^{sat}$) has been implemented. It is computed at the surface layer as a function of atmospheric $pCO_2$, total alkalinity, SSS, and SST. Below the first layer, $DIC^{sat}$ is treated as a passive tracer advected by the circulation. Since the equilibrium time-scale of the air-sea $CO_2$ fluxes is typically longer than the surface watermasses' residence time (e.g., in the

deep water formation regions), a disequilibrium term ($DIC^{diss}$) is computed by subtracting the saturated from the preformed DIC components. The DIC disequilibrium component is used to diagnose the importance of ventilation variability on the physical solubility pump, and to the overall DIC storage (Eggleston and Galbraith, 2018):

$$.DIC^{pre} = DIC^{sat} + DIC^{diss}. \tag{13}$$

### 3.2   Natural inorganic carbon tracers

In order to comply with the Ocean Model Intercomparison Project (OMIP) of CMIP6 (Orr et al., 2017), we have implemented a natural tracer of DIC ($DIC^{nat}$), which is formulated in the same manner as $DIC^{tot}$, except that the air-sea gas exchange is computed with a fixed preindustrial atmospheric $CO_2$ concentration of 284.32 ppm. In a transient simulation with increasing atmospheric $CO_2$ concentrations, the difference between $DIC^{tot}$ and $DIC^{nat}$ represents anthropogenic DIC ($DIC^{ant}$):

$$DIC^{ant} = DIC^{tot} - DIC^{nat}. \tag{14}$$

The inclusion of a $DIC^{nat}$ tracer also requires the implementation of respective 'natural' components for both the alkalinity and the particulate inorganic carbon ($CaCO_3$) tracers. Similarly, a 'natural' air-sea $CO_2$ flux has been added to the model output. These 'natural' tracers are only activated in simulations with atmospheric $CO_2$ that departs from the preindustrial value. Here, $DIC^{nat}$, $TALK^{nat}$, and $CaCO_3^{nat}$ are initialized in the same manner as $DIC$, $TALK$, and $CaCO_3$, respectively. In transient historical and future scenario simulations, these tracers provide insights of the natural carbon evolution under anthropogenic

climate change.

    In addition, $DIC^{nat}$ also provides a more precise estimate of $DIC^{ant}$ entering the ocean since the preindustrial period than simply taking the difference between $DIC^{tot}$ at a specific time and its preindustrial mean value. Alternatively, these 'natural' tracers could be computed with a parallel transient simulation but with fixed preindustrial atmospheric $CO_2$ as the ocean carbon cycle boundary condition. The natural tracers hence allow for substantial saving of computational time, especially for

high-resolution simulations. We note that the $DIC^{nat}$ tracer has not been implemented in the sediment module. Hence, in very long time-scale transient simulations where the sediment changes become substantial, the $DIC^{nat}$ tracer may include some uncertainties.



### 3.3 Carbon isotopes

The $^{13}C$, and $^{14}C$ carbon isotope tracers have now been implemented in NorESM2. Naturally occurring stable carbon isotopes ($^{12}C$, $^{13}C$, and $^{14}C$) and their relative abundances provide valuable information about both past and present climate. Changes in the $^{13}C/^{12}C$ ratio are used, e.g., to (i) study glacial-interglacial atmospheric $CO_2$ changes from ice core air samples (Broecker and McGee, 2013; Bauska et al., 2016), to (ii) reconstruct bottom water oxygen concentrations (Hoogakker et al., 2015), (iii) reconstruct paleo deep water circulation (Toggweiler, 1999; Curry and Oppo, 2005; Crucifix, 2005), (iv) investigate oceanic anthropogenic $CO_2$ uptake involving the Suess effect (Gruber and Keeling, 2001; Quay et al., 2003; Holden et al., 2013), and (v) evaluate model sensitivity and performance (Braconnot et al., 2012; Schmittner et al., 2013). The $^{14}C$:$^{12}C$ ratio is used as a circulation and age tracer (e.g., Skinner et al., 2017). Globally, the $^{12}C$:$^{13}C$:$^{14}C$ isotope ratio is about 99:1:10e$^{-12}$. In order to allow for a comparison between different carbon isotope studies, the $^{13}C$:$^{12}C$ ratio is standardized and expressed in permil as $\delta^{13}C$ (Eq. 22 for DIC) (Zeebe and Wolf-Gladrow, 2001). Our implementation of $^{13}C$ follows the OMIP guidelines (Orr et al., 2017). The $^{14}C$ is standardized as $\Delta^{14}C$ (see Sect. 3.3.3). Variations in $\delta^{13}C$ are due to isotopic fractionation processes during air-sea gas exchange, photosynthesis and $CaCO_3$ formation, but the latter is neglected in our implementation. For $^{14}C$, each fractionation factor is the quadratic of the respective $^{13}C$ value (i.e., $\kappa^{14}C = \kappa^{13}C^2$). Lastly, $^{14}C$ is radioactive and decays with a half-life of 5730 years to $^{14}N$ following:

$$dX^{14}C/dt = -\lambda \cdot X^{14}C, \tag{15}$$

where

$$\lambda[day^{-1}] = ln(2)/(5730 \cdot 365). \tag{16}$$

$X$ represents any $^{14}C$ state variable. In the model, all $^{14}C$ tracers are normalized to prevent numerical errors from carrying values close to the precision of the model.

The newly implemented marine carbon isotope code parallels the respective total DIC code, and in addition includes fractionation during photosynthesis and air-sea gas exchange (as well as decay for the $^{14}C$ tracers). In addition to the dissolved inorganic tracers, the following $^{13}C$ and $^{14}C$ state variables have also been implemented: dissolved organic carbon, particulate organic carbon, calcium carbonate, phytoplankton, and zooplankton. Therefore, 12 new tracers are added (isotopic DOC, POC, $CaCO_3$, zooplankton, and phytoplankton). Due to the long equilibration time, the isotopic tracers are only activated in the computationally efficient configurations, such as the ocean carbon cycle stand-alone configuration (NorESM2-OC) or the low resolution version of the coupled model. Equilibrium times of the carbon isotopes are long due to the slow air-sea gas equilibration (Broecker and Peng, 1974) and long-term transient effects from the balance between sediment burial and weathering (Roth et al., 2014). Realistic equilibration times are therefore currently only obtained when bypassing the sediment module of the model when running the carbon isotopes, as well as omitting carbon isotope input from rivers. When bypassing the sediment, mass balance is maintained by redistributing POC fluxes to the sediment over the entire overlying water column, and by dissolving inorganic carbon fluxes as well as opal fluxes at the bottom of the model immediately. Ongoing work will add the possibility for a fast sediment spin-up for use in future versions of the biogeochemical ocean model. Since $\Delta^{14}C$ is governed



mainly by radioactive decay and circulation, we focus on $\delta^{13}C$ in our description of isotopic fractionation effects (Sect. 3.3.1 and 3.3.2).

### 3.3.1 Air-sea gas exchange fractionation

During air-sea gas exchange, the lighter $^{12}C$ isotope preferentially escapes to the atmosphere. This fractionation process in-
5 creases $\delta^{13}C$ of surface ocean DIC, although the local net effect depends on the interplay between the local thermodynamic air-sea disequilibrium, the air-sea gas exchange rate, and the strength of the fractionation (Schmittner et al., 2013; Morée et al., 2018). The air-sea fractionation is a function of temperature ($T [^{\circ}C]$) and $CO_3^{2-}$ fraction ($fCO_3$), such that fractionation increases with decreasing temperatures, resulting in higher $\delta^{13}C$ in colder than in warmer surface water (Zhang et al., 1995). The fractionation during air-sea gas exchange, which varies between $\sim$8 and 10.5 ‰, is due to the combined effects of (1) the
10 fractionation between $CO_2$ and the different carbon species, $\alpha_{CT_g}$, (2) kinetic fractionation, $\alpha_k$, and (3) fractionation during gas dissolution, $\alpha_{aq_g}$. The net air-sea $^{13}CO_2$ flux is formulated as follows:

$$F^{13}CO_2 = k_w \cdot \alpha_k \cdot \alpha_{aq_g} \cdot (p^{13}CO_2^{atm} - p^{13}CO_2^{sw}/\alpha_{CT_g}). \tag{17}$$

Any fractionation $\alpha_i$ in Eq. 17 can be expressed as $\alpha_i = (\varepsilon_i/1000+1)$‰, where

$$\varepsilon_k = -0.88, \tag{18}$$

$$\varepsilon_{aq_g} = 0.0049 \cdot T - 1.31, and \tag{19}$$

$$\varepsilon_{CT_g} = 0.0144 \cdot T \cdot fCO_3 - 0.107 \cdot T + 10.53. \tag{20}$$

In Eq. 17, $k_w$ represents the gas transfer velocity for $CO_2$ according to Wanninkhof (2014) and T is in degrees C.

### 20 3.3.2 Biological fractionation

Phytoplankton prefers the lighter ($^{12}C$) isotope during photosynthesis, thereby increasing $\delta^{13}C$ of DIC in the surface ocean and producing low-$\delta^{13}C$ POC. In the interior ocean, the low-$\delta^{13}C$ POC is released back into the water column during remineralization/respiration, though without fractionation (Laws et al., 1995; Sonnerup and Quay, 2012). The 'biological isotope pump' thus creates a gradient of higher surface water $\delta^{13}C$ and lower deep water $\delta^{13}C$. The average fractionation during pho-
25 tosynthesis is approximately 19 ‰ (Lynch-Stieglitz et al., 1995; Tagliabue and Bopp, 2008). Even though many relationships for biogenic isotope fractionation have been proposed (e.g., Rau et al., 1996; Keller and Morel, 1999; Popp et al., 1998), the modelled $\delta^{13}C$ distributions are not very sensitive to the different parameterization, especially not in the surface ocean (Schmittner et al., 2013; Jahn et al., 2015). In addition, some relationships may be unsuitable for global scale modelling applications due to the dependency on unknown parameters (e.g., specific species, cell size, and cell membrane permeability).





A parameterization by Laws et al. (1997) is chosen in NorESM2, where the biological fractionation $\varepsilon_{bio}$ depends on the ratio between phytoplankton growth rate $\mu$ [day$^{-1}$] at every model time step and the aqueous $CO_2^*$ concentration [$\mu mol kg^{-1}$]:

$$\varepsilon_{bio} = \frac{6.03 + 5.5\mu/CO_2^*}{0.225 + \mu/CO_2^*} \qquad (21)$$

The growth rate $\mu$ is the brutto growth rate, uncorrected for losses such as mortality and exudation. $\varepsilon_{bio}$ increases with increas-
ing $CO_2^*$ and decreasing $\mu$. Fractionation values are kept within a realistic range of 5-26 ‰ to correct for the influence of $\mu/CO_2^*$ extremes (similarly as done by Tagliabue and Bopp, 2008). This nonlinear parameterization of Laws et al. (1997) as described in Eq. 21 is preferable over the linear formulation by Laws et al. (1995), because the linear formulation could result in unrealistic fractionation at high growth rates. Isotope equilibrium fractionation during $CaCO_3$ formation increases $\delta^{13}$C of $CaCO_3$ and decreases seawater $\delta^{13}$C. Nevertheless, the fractionation effect during $CaCO_3$ formation is relatively small compared to
the effects of air-sea gas exchange and photosynthesis and therefore is often omitted in modelling (Lynch-Stieglitz et al., 1995; Schmittner et al., 2013). There are additional uncertainties related to temperature- and species-dependencies (Grossman and Ku, 1986; Zeebe and Wolf-Gladrow, 2001). Taking this into consideration, we have chosen not to implement fractionation during $CaCO_3$ formation in NorESM2.

### 3.3.3  Diagnostic and initialization

In order to evaluate the carbon isotopes against observations, we compute the diagnostic variables $\delta^{13}$C and $\delta^{14}$C according to their standard formulations, as follow:

$$\delta^{13}C = \left[ \frac{DI^{13}C/DI^{12}C}{(^{13}C/^{12}C)_{PDB}} - 1 \right] \cdot 1000‰ \qquad (22)$$

$$\delta^{14}C = \left[ \frac{DI^{14}C/DIC}{NBS_{std}} - 1 \right] \cdot 1000‰ \qquad (23)$$

$$\Delta^{14}C = \delta^{14}C - 2 \cdot (\delta^{13}C + 25) \cdot (1 + \delta^{14}C/1000) \qquad (24)$$

where $(^{13}C/^{12}C)_{PDB}$ is the established Pee Dee Belemnite standard ratio of 0.0112372, and $NBS_{std}$ is 1.170e$^{-12}$ (Orr et al., 2017). The $DI^{12}C$ is computed as the difference between total $DIC$ and $DI^{13}C$.

In the model, the initialization of the carbon isotope tracers is done as follows: first, the model is spun up with preindustrial
boundary conditions until the non-isotope carbon chemistry reaches a quasi-equilibrium states. Next, we compute $DI^{13}C$ initial values by solving Eq. 22 for $^{13}$C using the simulated AOU and DIC together with the $\delta^{13}$C:AOU relationship reported by Eide et al. (2017), i.e., $\delta^{13}$C=-0.0075·AOU+1.72. The $DI^{14}C$ is initialized by first calculating $\delta^{14}$C by solving Eq. 24 using pre-industrial $\delta^{13}$C from Eide et al. (2017) (with the missing upper 200 m copied from the 200 m depth values) and the observational-based estimate of pre-industrial $\Delta^{14}$C (Key et al., 2004). This $\delta^{14}$C is then converted to absolute model $DI^{14}C$
using Eq. 23. The remaining isotope tracers are initialized as $C \cdot R \cdot \zeta$, with $C$ as the total carbon counterpart of respective





isotope tracer, $R$ as DI$^{13}$C/DI$^{12}$C or DI$^{14}$C/DI$^{12}$C ratio, and $\zeta$ = 0.98, a typical value for biological fractionation (applied for organic carbon components only).

Here, the prognostic atmosphere $pCO_2$ was initialized at 278 ppm from the start of the simulation. At initialization of the carbon isotopes, atmospheric $\delta^{13}C$ and $\Delta^{14}C$ are set to -6.5 ‰ and 0 ‰, respectively. Lastly, the results are presented as
calibrated $\Delta^{14}C$ to account for long-term drift. That is, $DI^{14}C$ is multiplied with a factor $F$ before standardization to $DI^{14}C$ corresponding to an atmospheric $\Delta^{14}C$ of zero permil ($^{14}C_{atmzero}$).

$$F = \frac{^{14}C_{atmzero}}{^{14}C_{atm}}, \tag{25}$$

where$^{14}C_{atm}$ in our simulation is found from the pre-calibrated $\Delta^{14}C_{atm}$ of approximately 36 ‰ and $^{14}C_{atmzero}$ is likewise determined by solving Eqs. 23 and 24 for $\Delta^{14}C$=0 ‰ in both cases using model $\delta^{13}C_{atm}$ (-7.5 ‰) and atmospheric pCO$_2$ (294
ppm). This leads to $F$=~2.5%.

## 4   Model simulations

Due to the long time scale of the large-scale ocean thermohaline circulation (flushing time is in the order of 1000-1500 years), a sufficiently long model integration in the order of at least 1000 years is required to achieve a quasi-equilibrium biogeochemical state in the interior ocean (Séférian et al., 2016). Prior to running any transient simulations, we have spun the model up for 1200
model years in a fully coupled configuration so that the ocean biogeochemistry reaches a quasi equilibrium state. In the spin-up simulation, all boundary conditions are fixed to constant preindustrial values, following the CMIP6 protocols (Eyring et al., 2016a). Due to the oceanic DMS fluxes, there is an internal feedback between the ocean biogeochemistry and the atmospheric radiative forcing during the spin-up. At the end of the model spin-up, the simulation is branched off into (1) preindustrial control simulation (*piControl*, years 1850-2349), (2) transient *historical* simulation (years 1850-2014), and (3) *esm-piControl-spinup*
simulation (100 model years). Simulation (3) is furthermore branched off into (3a) an *esm-piControl* simulation (for 250 years), and (3b) a transient *esm-hist* (years 1850-2014) simulation. In all *esm-experiments*, the atmospheric CO$_2$ is prognostically computed from ocean-atmosphere and land-atmosphere CO$_2$-fluxes, as well as from prescribed anthropogenic emissions (for *esm-hist*). Here, the atmospheric CO$_2$ is transported by the atmospheric circulation model (see also Eyring et al., 2016a). In the *esm-piControl-spinup* the atmospheric CO$_2$ concentration is relatively stable with a small drift of -0.002 ppm yr$^{-1}$ and a
long-term mean of 280.6±0.4 ppm.

For the newly implemented carbon isotopes, a considerably longer spin-up is required. Therefore, the carbon isotope tracers were not activated in our fully coupled simulations. Instead, simulations with carbon isotopes switched on have been performed using the ocean carbon cycle stand-alone configuration of NorESM2. Additionally, a lower resolution ocean grid (using a similar tripolar grid as described above but with a nominally 2° resolution) has been used. This configuration avoids the high
computational cost of running a long spin-up in a fully coupled configuration. The atmospheric forcing of the spin-up is the CORE normal year forcing (Large and Yeager, 2004), which represents a climatological mean year with a smooth transition between end and start of the year. Atmospheric CO$_2$, $^{13}$CO$_2$, and $^{14}$CO$_2$ concentrations are kept track of with a box-atmosphere





(i.e., assuming 100% instant mixing), which is updated each time-step according to the modelled air-sea $CO_2$ fluxes using a conversion factor of 2.13 ppm $Pg^{-1}$). These simplified prognostic atmospheric fields are simulated from the start of the spin-up (and the subsequent start of the carbon isotope simulation). As mentioned above, we have not implemented the carbon isotopes in the sediment compartment of the model. Therefore, the sediment module of HAMOCC was switched off for the carbon

isotope simulations presented here. Otherwise, the setup of the ocean carbon cycle stand-alone configuration is as described in Schwinger et al. (2016). The model spin-up was run for 5000 years, of which the first 1000 years were run without carbon isotopes. At year 1000 (4000), after the largest transient changes in biogeochemical tracers have flattened out, we initialised the the $^{13}C$ ($^{14}C$) tracers as described above.

Except for the carbon isotopes, all analyses shown in this paper were extracted from the two transient historical simulations
(prescribed $CO_2$-*historical* and prognostic $CO_2$-*esm-hist*). Since both simulations produce nearly identical climatology states, for the majority part of the mean state evaluation, we only present results from the *historical* simulation.

## 5    Results

Here, we evaluate the model performance in simulating the mean climatological state of key biogeochemical tracers as well as the evolution of air-sea $CO_2$ fluxes from the transient *historical* and *esm-hist* simulations.

### 5.1    Statistical performance summary

We assess, relative to the observational estimates and the earlier version NorESM1, the simulated long-term annual mean of global hydrography and biogeochemical tracer distributions. Where relevant, the mean seasonal cycles are also evaluated for the surface values. The list of parameters, the averaging periods, and the respective observational data used to assess the model performance are given in Table 2. We use spatial correlation and normalized-RMSE (root mean squared error) metrics
to measure the model-data difference and determine whether or not the current model version has improved and performed better than the earlier version. The normalization was done by dividing the RMSE with the standard deviation of the respective observations.

Figure 2 shows that, except for the surface layer, the normalized-RMSE of most of the NorESM2 variables' mean climatology is either noticeably improved or relatively similar to that of NorESM1. At subsurface 500 m, all variables but salinity have
lower RMSE values. For many of the biogeochemical tracers (phosphate, nitrate, oxygen, and DIC), NorESM2 shows improvements in model-data deviation throughout the water column. Similarly, the respective spatial correlations with the observations are considerably improved for most variables, especially for nitrate, DIC, and alkalinity. For surface seasonality, NorESM2 performs fairly comparable to NorESM1, with improvements in all seasonal metrics (NRMSE and spatial pattern) for surface salinity and net primary production. NorESM2 simulates slightly larger NRMSE in its surface nutrients (phosphate, nitrate,
and silicate) in both their annual and seasonal average values relative to NorESM1. This is attributed to the anomalously high concentration in the Arctic Ocean and parts of the Southern Ocean (just north of the Antarctic Circumpolar Current). More details on the performance of each specific variable are discussed in the following subsections.





## 5.2 Temperature

NorESM2 simulates a warm bias for both preindustrial conditions and the contemporary surface ocean (Fig. 3d), where a warm bias as high as 5°C is simulated in some regions of the Southern Ocean. Cold biases are simulated in parts of the north and equatorial Pacific, North Atlantic subpolar gyre, and the Arctic Ocean. The climatological mean temperatures are broadly

similar between both NorESM versions with NorESM2 having less spatial correlations at 1 and 1.5 km depths (Fig. 2a, b). At depths below 2 km, with the exception of North Pacific, NorESM2 simulates less biases. This improvement is related to the more accurately simulated Antarctic Bottom Water (AABW) in both the Pacific and Atlantic basins, as also depicted in Fig. 3. While NorESM1 simulates a too strong AMOC strength (roughly 31 Sv at 26°N), NorESM2 has a more reasonable strength of approximately 21 Sv. Nevertheless, the warm bias of 1-2°C in the North Atlantic Deep Water (NADW) remains

when compared to observations. The cold bias seen previously in the intermediate watermass in the Southern Ocean and North Pacific has been improved in the new version. In the Atlantic sector just south of 30°N at 1 km depth, there is a warm bias of up to 5°C, which could be related to the anomalously strong overflow watermass from the Mediterranean Sea. Nevertheless, the cold bias in the Atlantic and Pacific tropical/subtropical thermocline simulated in NorESM1 is now improved in NorESM2.

## 5.3 Salinity

The RMSE in surface salinity is reduced in NorESM2, mostly owing to improvements in the Arctic, where the previous model version simulates too saline water. Also at the surface, NorESM2 simulates anomalously too fresh and too saline biases in the Pacific and the Atlantic basins, respectively. In the subtropical south Pacific, the negative bias is as high as -2 psu, and is consistent with a too strong precipitation rate in this region simulated by the atmospheric component (not shown). Positive biases are simulated in the northern Indian Ocean, and along the Benguela current extension. In the interior, the salinity bias

in the Pacific and the Atlantic is in the order of ±0.11 and +0.4 psu, respectively. In the NADW, NorESM2 displays a positive bias, which is, however, smaller than that of NorESM1 (bias greater than 0.5 psu). Similar to temperature, NorESM2 produces too much saline water around 30°N at 1 km depth, due to overflow of high salinity Mediterranean watermasses. In the interior Pacific and Southern Ocean, both models' performances are generally comparable.

## 5.4 Mixed layer depth

Seasonally-averaged mixed layer depth (MLD) from NorESM2 and observations are shown in Fig. 5. To be consistent with the observational estimates (de Boyer Montégut et al., 2004), we have computed our MLD using the $\sigma_t$ (density) criteria, which is the first depth where the change from surface $\sigma_t$ of 0.03 has occurred. In NorESM1, the MLD is generally too deep throughout most ocean regions. The new improvements in the MLD parameterization in NorESM2 reduce this bias considerably, especially in the low- and mid-latitude and the North Pacific regions. In the North Atlantic subpolar and parts of the Southern Ocean, the

simulated MLD is still deeper than the observation-based estimates. In the Weddell Sea, NorESM2 also persistently simulates deep MLD throughout the year, a feature not seen in the observations.





## 5.5 Ocean ventilation

To assess the simulated ventilation, we compare the passive chlorofluorocarbon (CFC-11) tracer distribution in the interior Atlantic and Pacific Oceans as simulated by NorESM2 with that from observations, as shown in Fig. 6. We compare the simulated CFC-11 from calendar year 2000 with climatological estimates of Key et al. (2004). Similar to the observations, the

simulated CFC-11 concentrations in both the Atlantic and the Pacific are mostly confined to the upper 1 km depth, with the exception of the North Atlantic, where up to 0.5 nmol CFC-11 m$^{-3}$ penetrates down to 5 km depth. In the mid-latitude North Atlantic (i.e., 30°N), the model is unable to simulate the observed high values at around 1 km depth. This is likely due to the discrepancy in the watermass origin between the model and observations. Here, the watermass in the model is too old, as also seen in the too high Apparent Oxygen Utilization and dissolved inorganic carbon (see below subsections on oxygen and DIC).

The model simulates too strong deep water ventilation in both the Atlantic and Pacific sectors of the Southern Ocean as well as in the high latitudes of the North Atlantic.

## 5.6 Nutrients

In general, the climatological nutrient distributions (phosphate, nitrate, and silicate) in NorESM2 are either comparable or improved compared to those of NorESM1, both in spatial correlation and normalized RMSE (Fig. 2a, b). At depth, these

improvements are due to the new particulate organic carbon sinking scheme, where the sinking velocity increases linearly with depth. In NorESM1, the sinking speed is constant with depth, leading to more organic materials being remineralized in the upper 1500 m. The sinking scheme in NorESM1 produces anomalously high regenerated phosphate in intermediate depth (approximately between 100 and 1500 m depths). A consequence of this is a too strong oxygen minimum zone (OMZ; see also next subsection) and denitrification in the low latitudes, where the nitrate concentration becomes overly depleted. On the other

hand, at depths greater than 1500 m, NorESM1 exhibits too low concentrations in all nutrients relative to the observations.

With the new sinking scheme, more organic material reaches the deep ocean and gets remineralized there. As a result, model data biases in phosphate and nitrate in the interior are noticeable improved (Fig. 2b). In the Pacific OMZ, a reduced export production in NorESM2 also contributes to the reduced nutrient bias. In the interior North Atlantic, despite strong positive spatial correlation with the observations, NorESM2 shows a positive bias at depth below 3000 m and north of 30°S. Through

analyzing the quasi-conservative 'PO' tracer (Broecker, 1974), we are able to attribute this to the bias in the simulated watermass (Supplemental Figure S1). In NorESM2, this region is dominated by Antarctic Bottom Water, whilst the observations indicate NADW watermass. As with NorESM1, the current model still simulates too low nutrient concentrations in the North Pacific intermediate watermass. This is likely associated with the too low surface primary production simulated in this region (see also Sect. 5.8), leading to limited organic matter available for remineralization at depth. We note that the circulation bias

in this region could also contribute to the nutrient bias. In the Southern Ocean, too strong ventilation potentially leads to the negative nutrient biases in both Atlantic and Pacific sectors.





Less export production and deeper distribution of POC through the changed sinking scheme lead to a reduced rate of denitrification in the Pacific oxygen minimum zone and a greatly improved nitrate distribution. In NorESM1-ME, far too much nitrate was consumed for denitrification (see also Fig. 8e).

In NorESM, the phytoplankton growth rate is limited by multiple nutrients (i.e., phosphate, nitrate, and dissolved iron), in
addition to temperature and light. The inclusion of iron is critical to simulate the observed HNLC (High Nutrients Low Chloro-phyll) regions in the world ocean where year-long elevated concentrations of both nitrate and phosphate are not exhausted by phytoplankton (de Baar et al., 1995; Martin and Fitzwater, 1988). The three major HNLC regions are the subarctic North Pacific, the eastern and central equatorial Pacific, and the Southern Ocean. Here, low levels of bio-available iron concentrations limits the phytoplankton growths. In NorESM1, only the subarctic North Pacific is simulated to be iron-limited, with the other
two HNLC regions being mostly limited by nitrate, as shown in Fig. 10. Iron limitation is also incorrectly simulated in parts of the subtropical North Pacific, South Pacific, and most parts of the western Pacific. Following improvements in the iron parameterization, the three HNLC regions are now shown to be iron-limited in NorESM2. Nevertheless, the iron limitation bias in the western equatorial Pacific remains.

### 5.7 Dissolved oxygen

At surface level, dissolved oxygen is mostly constrained by solubility and hence sea surface temperature. Both NorESM1 and NorESM2 perform similarly for dissolved oxygen at the surface with subtle improvements in the seasonal spatial correlation coefficient of NorESM2. As with nutrients, the climatological distributions of interior oxygen improved considerably when compared to NorESM1, which is one of the CMIP5 ESMs with too strong OMZ (Cabré et al., 2015). In NorESM2, the OMZ volume is reduced considerably in both the equatorial Atlantic and Pacific, and the absolute dissolved oxygen values are also
much improved. The remaining bias is related to the nutrient trapping issue (Six and Maier-Reimer, 1996), as can also be seen from too high apparent oxygen utilization (AOU; Fig. 9). In the interior Pacific and Atlantic roughly at 2 km and deeper, NorESM2 simulates considerably higher oxygen concentration than the observations by as much as 50% (North Pacific deep water). In the North Pacific deep water, this is primarily due to the still too little organic matter flux reaching below the mesopelagic zone, leading to too low apparent oxygen utilization (Fig. 12). The bias in the dissolved organic carbon, which
is currently not well constrained due to lack of observational data, could also play a role in this. In addition to biological constraints, the remaining interior oxygen bias can be attributed to the too strong Antarctic Bottom Water (AABW) ventilation, which carries a colder (Fig. 3) and higher oxygen saturated watermass into both the Atlantic and Pacific bottom waters.

### 5.8 Biological production

In NorESM1, the primary production is generally too strong in the equatorial Pacific and in the high latitude summer months.
In the subtropical oceans, the simulated production is too low relative to the observational estimates. With the newly tuned ecosystem parameters of NorESM2, the spatial productivity patterns are improved considerably with an average absolute bias in the open ocean of less than 5 mol C m$^{-2}$ yr$^{-1}$ (Fig. 13). In NorESM1 these biases reach more than 10 mol C m$^{-2}$ yr$^{-1}$, especially in the Equatorial Pacific, the South Atlantic and the Southern Ocean. Seasonally, the spatial patterns are also





improved when compared to the observations with a correlation of approximately $r = 0.5$ (Fig. 2). The RMSE has also reduced considerably, particularly in the October-November-December months, where NorESM1 simulates a much too strong spring bloom in the Southern Ocean which is overestimated by as much as a factor of eight. Similarly, the spring bloom in the North Atlantic and North Pacific are overestimated in NorESM1 during the boreal spring months. The improved spring bloom

characteristics at high latitudes, however, comes at the cost of a too low annual mean productivity in the North Atlantic and North Pacific.

Due to unresolved physical dynamics, the model is still unable to reproduce the observed high productivity in coastal regions. Globally, the contemporary total primary production in NorESM2 (NorESM1) is 35.3 (39.9) Pg C yr$^{-1}$. These estimates are lower than the observational (MODIS, excluding coastal grids) estimates of 46.1 Pg C yr$^{-1}$ but within the broad range of

CMIP5 models (Bopp et al., 2013). In our model, export production is estimated as the flux of particulate organic carbon exiting the base of the euphotic zone (here assumed to be 100 m depth), as a function of zooplankton and phytoplankton mortalities and zooplankton fecal materials. Table 3 shows that the export production in NorESM2 is 5.39 Pg C yr$^{-1}$, as compared to 7.90 Pg C yr$^{-1}$ simulated by NorESM1. Current estimates of the global export production remain highly uncertain, from 5 to >12 Pg C yr$^{-1}$, with a more recent satellite-based approach that accounts for food-web processes revealing values of 6±1.2 Pg C

yr$^{-1}$ (Siegel et al., 2014).

Table 3 also summarizes other export production metrics from both model versions. As a consequence of the new particle sinking scheme, there is more POC exported into the deep ocean. This is especially reflected by the values of transfer efficiency ($T_{eff-1km}$), which is calculated as a ratio between POC fluxes at a depth of 1000 m and of 100 m. The $T_{eff-1km}$ in NorESM2 (0.24) is increased by a factor of four relative to that in NorESM1 (0.06). Compared to a recent estimate of transfer efficiency

reconstructed from observed interior biogeochemistry, NorESM2 still simulates strong values of >50% in the eastern Equatorial Pacific (Fig. 13 Weber et al., 2016). On the other hand, the transfer efficiency in the Southern Ocean and northern high latitudes are comparable. This represents a significant improvement when compared to NorESM1, which simulates $T_{eff-1km}$ values of less than 0.1 in nearly all ocean regions with the exception of eastern Equatorial Pacific and Equatorial Atlantic.

Figure 14 shows the seasonal cycle of biological production rates as simulated by NorESM1 and NorESM2 in different ocean

regions together with observation-based estimates. The regional mean values in NorESM2 are generally lower than those in NorESM1, except in the tropics and the North Pacific. Nevertheless, Fig. 2c shows that NorESM2 simulates lower normalized RMSE than NorESM1 in all seasons. As stated above, this bias could be reduced even more if we excluded the continental shelf regions. Nevertheless, Fig. 14 shows that the seasonal phase and amplitude of biological production rates in NorESM2 in the North Atlantic, North Pacific, and the Southern Ocean regions are closer to the observations when compared to NorESM1.

In the Southern Ocean (south of 20°S), NorESM2 no longer produces the large summer bias seen in NorESM1.

## 5.9  DMS production and fluxes

In NorESM2, DMS production is tuned towards the climatological observations from Lana et al. (2011). Figure 15 shows the surface DMS concentration for each season together with the observation-based estimates. To first order, the DMS distribution follows the spatial pattern of the simulated primary production, with higher concentrations during spring/summer periods in





high latitude regions. In the Equatorial Pacific and Atlantic upwelling regions, the model overestimates DMS concentrations. Similarly, in the North Pacific, where the model underestimates productivity during the boreal spring bloom (see also Fig. 13), the DMS concentration is lower than observed.

The simulated DMS concentration in the North Atlantic is comparable with observations. In the Southern Ocean, the model produces the observed high concentrations during summer months (JFM) along the 45°S latitude band well. In winter periods, the DMS concentration approaches zero in the model owing to the too low productivity, whilst the observations still indicate values of approximately 1 $\mu$mol S m$^{-3}$. For the present day period (1971-2000), the simulated global DMS flux is 19.76±0.19 Tg S yr$^{-1}$, which is in the lower end of observation-based estimates (17.6-34.4 Tg S yr$^{-1}$).

### 5.10   Dissolved inorganic carbon and alkalinity

Except for the polar Southern Ocean, the surface alkalinity concentrations in NorESM1 are approximately 100 $\mu$mol L$^{-1}$ higher than the observational estimate. And since the spin-up was forced with constant preindustrial atmospheric $CO_2$ concentrations, the surface DIC concentration adjusted (also anomalously high bias) to yield the 'correct' surface $pCO_2$. Consequently, biases in surface DIC and alkalinity compensate one another to yield the approximately 'correct' carbonate ion concentration (i.e., alkalinity minus DIC), seawater $CO_2$ buffer capacity, and air-sea $CO_2$ fluxes.

In NorESM2, we alleviate the surface alkalinity bias by increasing the surface sink associated with calcium carbonate production. This is done by increasing the $CaCO_3$ to phosphate uptake ratio ($R_{CaCO_3:P}$ in Table 1). As a result, the export of particulate $CaCO_3$ increases from 0.49 Pg C yr$^{-1}$ in NorESM1 to 0.66 Pg C yr$^{-1}$, with both estimates still within the range of the observation-based synthesis of 0.52±0.15 Pg C yr$^{-1}$ (Dunne et al., 2007). This modification increases the PIC:POC export ratio to 0.13 (0.06 in NorESM1, see also Table 3) and considerably improves the surface DIC and alkalinity concentrations
in NorESM2 (Figs. 16 and 17). At the same time, the spatial correlations with observations are also improved as well as the normalized RMSE (see Fig. 2a, b). We note that in few regions, e.g., the subtropical South Pacific, the model still underestimates the observed DIC and alkalinity concentrations. We note that the PIC:POC ratio is still within the large range of other studies (0.03 to 0.25; Koeve, 2002; Sarmiento et al., 2002)

As in the surface, the spatial distributions and normalized bias of interior DIC and alkalinity in the NorESM2 improve
considerably throughout the water column relative to NorESM1 (Fig. 2a,b). In both the interior Atlantic and Pacific basins, the spatial distribution and magnitude of DIC biases closely resembles that seen in the nutrients (e.g., phosphate ; Fig. 7e,f), when multiplied by the constant stoichiometry in the model ($R_{C:P}$=122). This suggests that the mechanisms driving the nutrient bias are also responsible for the bias in interior DIC. For instance, the positive anomaly in the equatorial Pacific between 1 and 3 km depths originates from too much biological remineralization in the model.
The simulated interior alkalinity concentrations in NorESM2 between 1 and 3 km depths are anomalously high, which can partially be attributed to too low interior remineralization in the Atlantic (see also AOU values Fig. 12) or too high $CaCO_3$ export production (Table 3).





### 5.11 Surface pCO$_2$ and sea-air CO$_2$ fluxes

The NorESM2 pCO$_2$ spatial correlation relative to the observations is improved in almost all seasons when compared to NorESM1, while the NRSME is reduced for JFM and AMJ months (Fig. 2c,d). Within these months, improvements are mostly seen in the Southern Ocean, where too strong surface mixing in NorESM1 leads to anomalously high pCO$_2$. The

climatological mean of contemporary surface pCO$_2$ compares well with the observational compilations, as seen in Fig. 18a. Similar improvements are also seen in the CO$_2$ fluxes, where NorESM2 simulates the broad spatial patterns relatively well (Fig. 18c). NorESM2 also produces moderately weaker annual mean carbon flux in the northern mid- to high-latitudes than NorESM1. Figure 18b,d shows the zonally-averaged monthly surface pCO$_2$ and CO$_2$ fluxes in NorESM2. Here, the model also agrees well with the observation-based estimates (contour-lines) in term of amplitude and temporal variability, with distinct

seasonal cycle poleward of 20° latitude. In the extra-tropical oceans (between 30–60°N and south of 60°S), the simulated winter (January-March and July-September in northern and southern hemispheres, respectively) pCO$_2$ is considerably lower than that seen in the observations. This feature also translates to a stronger carbon sinks during these periods.

Figure 19 depicts the mean seasonal cycle of sea-air CO$_2$ fluxes from NorESM models and observations, averaged over different regions. In the North Atlantic, North Pacific, and the mid-latitude Southern Ocean (between 20–40°S), the seasonal

phase and mean integrated annual CO$_2$ fluxes in NorESM2 compare better with the observations than those in NorESM1. Nevertheless, NorESM2 simulates a stronger than observed carbon sink in the Labrador Sea and along the Kuroshio current extension during boreal winter months. We note that the Landschützer et al. (2014) data estimates higher surface pCO$_2$ (hence a weaker carbon sink) in these regions than the observational estimates from Takahashi et al. (2009). Despite the warm SST bias, NorESM2 simulates a moderately weaker than observed CO$_2$ outgassing throughout the year. A potential explanation

for this is the weaker upwelling rates of DIC-rich deep watermass in the model. In the Southern Ocean (south of 40°S), the strong bias in the seasonal amplitude in NorESM1 is now considerably improved, due to the improvements in the biological production in this region. Nevertheless, here the seasonal phase in NorESM2 is approximately reversed: the strongest sink is simulated in the austral winter months whereas observational estimates indicate summer months.

### 5.12 Distribution of $\delta^{13}C$ and $\Delta^{14}C$

Figure 21a-c shows that the spatial pattern of $\delta^{13}C$ is in fair agreement with the gridded preindustrial observational estimate of Eide et al. (2017). Nevertheless, The ocean component of NorESM2 simulates weaker variability than observed, i.e., it is unable to reproduce both very low (negative) and high (positive) values. Therefore, the simulated $\delta^{13}C$ concentrations in the relatively younger watermasses, such as the North Atlantic Deep Water (NADW) and the Antarctic Intermediate Water (AAIW), have negative bias relative to the observations (Fig. 21e). On the contrary, positive biases are depicted in the older

Antarctic Bottom Water (AABW) and in the deep North Pacific (Fig. 21e-f). The $\delta^{13}C$ can be decomposed into biological and residual (air-sea gas exchange and circulation) components (Broecker and Maier-Reimer, 1992; Eide et al., 2017):

$$\delta^{13}C = \delta^{13}C_{BIO} + \delta^{13}C_{AS}, \tag{26}$$





where $\delta^{13}C_{BIO}$ can be estimated as a function of phosphate and DIC distribution and their global mean values:

$$\delta^{13}C_{BIO} = \frac{-19 \cdot r_{C:P}(PO_4 - \overline{PO_4})}{\overline{DIC}} + \overline{\delta^{13}C}. \tag{27}$$

Applying this decomposition, we can attribute the majority of the differences between model and observational estimates of $\delta^{13}C$ to biases in the biological component (Fig. 22). The $\delta^{13}C_{BIO}$ reveals that in the Southern Ocean (south of $\sim$55°S), as well

as in its exported deep waters, NorESM2-OC simulates too high $\delta^{13}C$ due to the too small regenerated signal in these waters. The negative $\delta^{13}C$ anomaly in the North Atlantic of about -0.5 ‰ is a combination of a too low $\delta^{13}C_{BIO}$ signature in NADW (Fig. 22e), combined with a too negative influence from air-sea gas exchange and circulation as compared to observational estimates (the residual term $\delta^{13}C_{AS}$). In the Pacific the positive anomaly from the Southern Ocean persists throughout most of the basin due to too weak remineralization, except for at intermediate depths where remineralization is high (see also AOU,

Fig. 12).

The simulated $\Delta^{14}C$ distribution is in general agreement with the pre-industrial observational estimate by Key et al. (2004). Nevertheless, interior model-observation biases are in the range of $\pm$50 ‰. Waters south of 60°S are positively biased as compared to observations (Fig. 23), indicating too young and well-ventilated waters in this region. This positive bias of 20-50 ‰ is carried into the Atlantic and Pacific basins (Fig. 23e-f) and is equivalent to an underestimation of radiocarbon age by

200-400 years. North of the Equator, a negative bias of similar magnitude indicates too old water masses in the Atlantic basin below 3 km depth as well as throughout the water column in the northern half of the Pacific basin. The top 800m of the water column has a negative bias of 100 ‰, which we attribute to the too negative atmospheric p$^{14}$CO$_2$ before calibration (-36 ‰) and biases in air-sea equilibration and fractionation. Calibrated $\Delta^{14}C_{atm}$ is by definition 0 ‰, $\delta^{13}C_{atm}$ equilibrates at -7.5 ‰, and atmospheric pCO$_2$ is 294 ppm at the end of the simulation.

## 5.13   Transient changes

In this subsection, we describe the global mean surface air temperature and oceanic carbon uptake over the historical period (1850–2014) as simulated by NorESM2. For this, we discuss the two transient simulations: (i) *historical* and (ii) *esm-hist*. Similar to SST (see Fig. 3), there is a warm-bias in the simulated surface air temperature. For the period of 1850-1859, the NorESM2 global mean surface air temperature is 14.5°C compared to 13.7°C from the HadCRUT4 dataset (Morice et al.,

2012). Nevertheless, the simulated warming between 2005-2014 and 1850-1859 is 0.7°C, relatively comparable with that from the observations of 0.8°C. The warming in the *esm-hist* is lower, 0.6°C. Figure 20a shows that NorESM2 simulates warming rates that are comparable to observations between 1970 and 2010, but shows a stronger cooling between 1950 and 1970 (*historical* and *esm-hist*).

In both simulations, the global mean surface ocean pCO$_2$ follows the respective atmospheric CO$_2$ trend over the simulation

periods. Towards the end of the historical period, the oceanic pCO$_2$ diverges from (i.e., lower than) the atmospheric counterpart by approximately 20 ppm (Fig. 20b). Consistent with the lower ocean (than the atmosphere) pCO$_2$ growth rates, the ocean carbon uptake continues to increase over most of the transient period in both simulations (Fig. 20c). In the *historical* simulation, the carbon sink stabilizes at roughly 0.7 Pg C yr$^{-1}$ between 1910 and 1960. This flattening of the uptake strength is associated





with the slow down in the atmospheric $CO_2$ growth rate. Figure 20d shows that the atmospheric growth rate weakens during this time window (orange-bars in Fig. 20d). On the contrary, the carbon sink in *esm-hist* steadily increases during the same period, consistent with the steady increase in the simulated prognostic atmospheric $CO_2$ (green-bars in Fig. 20d).

From 1960 onward, the atmospheric $CO_2$ growth rates in both simulations increase almost linearly in time from 0.8 to more than 2.0 ppm yr$^{-1}$. Consequently, the ocean carbon uptake strengths increase as well, from approximately 1 Pg C yr$^{-1}$ in the 1960s up to 2.5 Pg C yr$^{-1}$ in 2014 (in both *historical* and *esm-hist*). Rapid strengthening in ocean carbon uptake is also evident in estimates from the Global Carbon Project (Fig. 20, Le Quéré et al., 2018). For the decades of 1980s and 1990s, the simulated carbon uptake is also well within the IPCC (Intergovernmental Panel for Climate Change, Denman et al., 2007) estimates (Table 4). In NorESM2, a new tracer $DIC^{nat}$ has been implemented (see also Sect. 3.2), allowing for a more accurate estimate of the anthropogenic carbon uptake and storage in the ocean. Using this tracer, we can also estimate the net flux of anthropogenic carbon into the ocean during the historical period, as depicted by the purple line in Fig. 20c). The strong resemblance between the orange and purple lines suggests that the long-term trend in oceanic carbon sinks is associated with the increasing atmospheric $CO_2$, rather than changes in climate states (as expected for the historical runs where the change in the climate state is relatively small). We note that, in the preindustrial control simulation, the ocean is a weak net carbon sink, of approximately 0.1 Pg C yr$^{-1}$. Table 4 also shows that the cumulative carbon uptake by the ocean for the 1850-1994 and 1994-2011 is well within the observation-based estimates. The cumulative carbon uptake in NorESM2 is lower than that of NorESM1 and compares better with observations.

## 6  Summary and discussion

The ocean biogeochemical component of the Norwegian Earth System Model (NorESM) has been updated from version 1 (as used in CMIP5) to version 2 (NorESM2). These developments focus on alleviating known biases in mean states and seasonal cycles of key variables when compared to the observations. This paper describes new and improved processes, newly implemented diagnostic and carbon isotope tracers, and highlights the model improvements relative to the earlier model version.

On the biogeochemistry side, we have introduced a revision to the particulate organic carbon vertical sinking scheme, an improved tuning of ecosystem parameters, riverine inputs of nutrients and other biogeochemical constituents, an updated air-sea gas exchange parameterization, and atmospheric deposition of nitrogen. In NorESM2, the ocean biogeochemistry also simulates (i) fully interactive oceanic DMS fluxes coupled to the atmospheric model, (ii) newly implemented preformed tracers, which will allow for a more detailed diagnosis of physical and biogeochemical sources and sinks in future studies, and (iii) capability to simulate carbon isotope, e.g., in extended paleo time-scale simulations.

On the physical side, the simulated Atlantic Meridional Overturning Circulation strength is considerably improved. An hourly coupling (previously daily) across the ocean-atmosphere-ice interfaces has been implemented. Intensive fully coupled testing prompted further attempts to optimize mixed layer physics parameters, which slighly reduce the bias in MLD when compared to our earlier model version.





In connection to our contribution to the CMIP6 simulations, two model configurations have been prepared, NorESM2-LM and NorESM2-MM, where the latter adopts a higher atmospheric resolution. In both versions, the ocean components are identical. Following the CMIP6 protocols (Eyring et al., 2016a), we have performed preindustrial control (*piControl*, years 1850-2349) and transient climate (*historical*, years 1850-2014) simulations. Here, we analyzed the simulations performed

using both model versions.

As a result of the above developments, the overall performance of NorESM2 in simulating the climatological state of interior ocean biogeochemistry has improved considerably. Some of the key improvements include (i) better representation of equatorial Pacific oxygen minimum zone, (ii) improved interior nutrient distributions, and (iii) representation of iron limitation in the Southern Ocean, equatorial and North Pacific.

With respect to transient ocean carbon sinks, previous studies have indicated the importance of the background biogeochemical state, such as alkalinity, dissolved inorganic carbon, and buffer capacity, in order to adequately simulate the long-term trends (Hauck and Völker, 2015; Fassbender et al., 2018; Lebehot et al., 2019). NorESM1 exhibits a considerable bias for the background biogeochemical state, especially in surface alkalinity concentrations. These biases have been substantially reduced in NorESM2, allowing for a more adequate representation of the background buffer factor and improved model fidelity in

simulating the temporal evolution of the ocean carbon sink.

The seasonality of upper ocean biogeochemistry has been identified as critical for quantifying a model's uncertainty in its long-term projections of oceanic carbon uptake (Fassbender et al., 2018). Further, with ongoing anthropogenic $CO_2$-emissions, the seasonal variability of regional sea-air $CO_2$-fluxes is expected to amplify. This amplification, which is linear for the thermal component and nonlinear for the biophysical component of oceanic $pCO_2$ (Fassbender et al., 2018), can also impact the long-

term evolution of sea-air $CO_2$-fluxes. In this aspect, NorESM1 simulates considerable bias in the seasonal cycle of biological production and air-sea $CO_2$ fluxes, particularly in the Southern Ocean. The updated set of ecosystem parameters in NorESM2 is able to mitigate these biases to some extent. In the North Atlantic, North Pacific, and most parts of the Southern Ocean, the seasonal cycle (both in phase and amplitude) of primary production and $CO_2$-fluxes in NorESM2 reflects observational estimates and their driving mechanisms more accurately (Landschützer et al., 2018). We note however that there is a mismatch

between the $CO_2$-fluxes seasonal cycle in observations and NorESM2 in the Southern Ocean between 40°N and 60 °N. A closer attention of the associated physical (e.g., solubility) and biogeochemical (e.g., biological productivity) drivers in this region is needed when considering future scenarios in the future. However, general seasonal characteristics are improved and we are confident that NorESM2 will provide a better estimate of future changes.

Despite numerous model improvements, several biases remain in the updated model. A long-standing issue of nutrient

trapping in the intermediate depth of equatorial Pacific persists. Here, the simulated phosphate concentration is still higher than observed, but the bias is less severe than that of NorESM1. Consistently, dissolved inorganic carbon is still anomalously high and oxygen too low in the same region. The simulated Southern Ocean still has a too strong ventilation, leading to biases in the newly formed bottom watermass. Other physical biases such as the Antarctic Bottom Water penetrating too northward also contribute to the simulated model-data differences in the interior North Atlantic.



Currently, evaluations of ocean biogeochemical component in ESMs have been largely focused on assessing the model ability in simulating the mean climatology state, however, further options exist (Eyring et al., 2016b; Heinze et al., 2019). The ocean biogeochemical observing community has put a lot of efforts in sustaining key monitoring networks, such as the Surface $CO_2$ Atlas (SOCAT; Bakker et al., 2014) and the Global Ocean Data Analysis Project (GLODAP; Olsen et al., 2016). This

critical observational network has allowed for additional constraints for ESMs, such as by confronting the model ability to simulate seasonality of surface ocean $pCO_2$ as well as long-term trends in surface ocean $pCO_2$ (Tjiputra et al., 2014), ocean acidification (Lauvset et al., 2016), and deoxygenation (Tjiputra et al., 2018). As ESMs simulate emerging climate change signals in the coming decades (Henson et al., 2017), sustaining these observations into the future is becoming more important than ever.

In order to tackle the challenges associated with climate change, ocean modules in Earth system models could be further upgraded to include interactive methane ($CH_4$), nitrous oxide ($N_2O$), and biogenic volatile organic compounds (BVOCs) cycles for enabling model projections in respective emission driven frameworks. Moreover, NorESM2 still has a relatively simple marine biology that would benefit from more key functional groups as well as their sensitivity to climatic stressors. In preparation for the anticipated increasing spatial resolution in the physical components of the model, different data assimilation

methods (combined with state-parameter estimation on the basis of observations) are currenty being explored to allow for efficient optimization of the current ecosystem parameterization (Tjiputra et al., 2007; Gharamti et al., 2017).

Further NorESM-internal improvements may consider online estimated or progonostic organic carbon emissions from the ocean biogeochemistry to the atmosphere. Currently, we use a fixed particulate organic carbon emissions from the ocean, linked to an upper-ocean chlorophyll-a climatology. The spatial distribution and temporal characteristics of DMS emissions may have

an impact on the strength of the feedback involving DMS and aerosol-cloud interactions. Dust and iron or phosphate contained therein as well as nitrogen deposition will be computed in the future version of the atmospheric NorESM module and may provide more realistic nutrient input and may thus lead to feedbacks in a coupled ESM. The biogeochemical exchange processes are also dependent on sea ice cover and require better knowledge of air-sea-interaction processes in polar and subpolar regions. Mineralisation processes in land regions may change under climate change conditions, leading to important changes in nutrient

fluxes into the ocean biogeochemical systems. Biogenic Volatile Organic Compounds (BVOC) and halocarbons emitted from the ocean have been suspected to influence atmospheric chemistry, e.g., in the stratosphere. It will certainly be the ambition of further developments to investigate sensitive of these processes to climate change to increase complexity where needed.

Beyond their current applications, future ESMs should be developed further to provide societal relevant information that is directly relevant for policy decisions on climate mitigation and adaptation measures. This includes potential climate hazards

to societies (such as marine heatwaves, sudden pH drops, and spreading low oxygen zones in the oceans) as well as the risks emerging from the combination of hazards, vulnerabilities, and exposures (Oppenheimer et al., 2014). Future inclusion of socio-economic dynamics in ESMs and their feedback to climate through societal transformative action remains a big challenge (Giupponi et al., 2013) while a new ocean related socio-economic scenario framework (OSPs) has been developed mirroring the established Socioeconomic Pathways or SSPs (Maury et al., 2017). On the other hand, refining process complexity and

increasing resolution may not be the only alternative for further model development, as the results of a more complex model



simulations become more difficult to interpret (Bony et al., 2011). Therefore, simplified models and ESMs of intermediate complexity including the respective simplified ocean biogeochemistry modules, will need to be developed and used in parallel (for instance Steinacher et al., 2013).

*Data availability.* All observational data used in this paper are publicly available: World Ocean Atlas (https://www.nodc.noaa.gov/OC5/
5  woa13/), Global Ocean Data Analysis Project version 2 (https://www.glodap.info/), surface mixed layer depth (http://www.ifremer.fr/cerweb/
deboyer/mld/home.php), surface $pCO_2$ and air-sea $CO_2$ fluxes (https://www.nodc.noaa.gov/ocads/oceans/SPCO2_1982_2011_ETH_SOM_
FFN.html), ocean primary production (https://www.science.oregonstate.edu/ocean.productivity/), Dimethyl Sulphide (https://www.bodc.ac.
uk/solas_integration/implementation_products/group1/dms/), $\Delta^{14}C$ and CFC-11 (https://www.nodc.noaa.gov/ocads/oceans/glodap/), and $\delta^{13}C$
(https://www.bcdc.no/data-products.html).

10  *Code and data availability.* The codes of Bergen Layered Ocean Model and isopycnic-based Hamburg Oceanic Carbon Cycle (BLOM-
iHAMOCC) can be downloaded from https//github.com/NorESMhub/BLOM. The NorESM CMIP5 and CMIP6 model outputs can be accessed through the Earth System Grid Federation (ESGF) decentralized database (https://esgf-node.llnl.gov).





**Table 1.** Parameter values of the ecosystem parameterisation that have been changed between NorESM1, NorESM-OC1.2, and NorESM2. NorESM-OC1.2 was an intermediate model version (Schwinger et al., 2016) and is included here for completeness and traceability. Naming of parameters follows Ilyina et al. (2013).

| Parameter | Symbol | NorESM1 | NorESM-OC1.2 | NorESM2 | Unit |
|---|---|---|---|---|---|
| Half-saturation constant PO$_4$ uptake | $K_{PO_4}$ | $2 \times 10^{-7}$ | $4 \times 10^{-8}$ | $4 \times 10^{-8}$ | kmol P m$^{-3}$ |
| Half-saturation constant grazing | $K_{zoo}$ | $4 \times 10^{-8}$ | $4 \times 10^{-8}$ | $8 \times 10^{-8}$ | kmol P m$^{-3}$ |
| Half-saturation constant silicate uptake | $K_{Si}$ | $1.5 \times 10^{-6}$ | $1 \times 10^{-6}$ | $5 \times 10^{-6}$ | kmol Si m$^{-3}$ |
| Phytoplankton mortality rate | $\lambda_{phy}$ | 0.008 | 0.008 | 0.004 | d$^{-1}$ |
| Zooplankton mortality rate | $\lambda_{zoo}$ | $5 \times 10^{-6}$ | $3 \times 10^{-6}$ | $3 \times 10^{-6}$ | (kmol P m$^{-3}$ d)$^{-1}$ |
| Phytoplankton exudation rate | $\beta_{phy}$ | 0.03 | 0.03 | 0.04 | d$^{-1}$ |
| Zooplankton excretion rate | $\beta_{zoo}$ | 0.03 | 0.06 | 0.06 | d$^{-1}$ |
| Maximum grazing rate | $\mu_{zoo}$ | 1 | 1 | 1.2 | – |
| Zooplankton assimilation efficiency | $\omega_{zoo}$ | 0.5 | 0.6 | 0.7 | – |
| Silicate to phosphorus uptake ratio | $R_{Si:P}$ | 25 | 30 | 33 | mol Si mol P$^{-1}$ |
| CaCO$_3$ to phosphorus uptake ratio | $R_{CaCO_3:P}$ | 35 | 40 | 45 | mol CaCO$_3$ mol P$^{-1}$ |
| Fraction of grazing ingested | $\varepsilon_{zoo}$ | 0.8 | 0.8 | 0.85 | – |
| Detritus remineralization rate | $\lambda_{det}$ | 0.03 | 0.025 | 0.025 | d$^{-1}$ |
| DOC remineralization rate | $\lambda_{DOC}$ | 0.025 | 0.025 | 0.004 | d$^{-1}$ |





**Table 2.** List of the simulated variables and the respective historical simulation periods at which their climatological values are averaged over. The last column indicates the observational reference used to validate the model output.

| Model variables | Periods | Observations and references |
|---|---|---|
| Temperature | 1971–2000 | World Ocean Atlas 2013 (Locarnini et al., 2013) |
| Salinity | 1971–2000 | World Ocean Atlas 2013 (Zweng et al., 2013) |
| Mixed layer depth | 1971–2000 | de Boyer Montégut et al. (2004) |
| Oxygen | 1971–2000 | World Ocean Atlas 2013 (Garcia et al., 2013a) |
| Phosphate | 1971–2000 | World Ocean Atlas 2013 (Garcia et al., 2013b) |
| Nitrate | 1971–2000 | World Ocean Atlas 2013 (Garcia et al., 2013b) |
| Silicate | 1971–2000 | World Ocean Atlas 2013 (Garcia et al., 2013b) |
| Dissolved inorganic carbon | 1997–2007 | Global Ocean Data Analysis Project (GLODAPv2; Lauvset et al., 2016) |
| Alkalinity | 1997–2007 | Global Ocean Data Analysis Project (GLODAPv2; Lauvset et al., 2016) |
| Surface $pCO_2$ | 1982–2011 | Landschützer et al. (2015) |
| Sea-air $CO_2$ fluxes | 1982–2011 | Landschützer et al. (2015) |
| Net primary production | 2003–2012 | Averaged of three remote sensing products (VGPM, Eppley-VGPM, and CbPM) from Moderate Resolution Imaging Spectroradiometer (Behrenfeld and Falkowski, 1997; Westberry et al., 2008) |
| Dimethyl sulfate | 1971–2000 | Lana et al. (2011) |
| CFC-11 | 2000 | Key et al. (2004) |
| $\delta^{13}C$ | Last 10 yrs of piControl | Eide et al. (2017) |
| $\Delta^{14}C$ | Last 10 yrs of piControl | Key et al. (2004) |

**Table 3.** Annual mean primary production metrics simulated by NorESM1-ME, NorESM2-LM, and NorESM2-MM.

| | NorESM1-ME | NorESM2-LM | NorESM2-MM | Units |
|---|---|---|---|---|
| Primary production | 39.86 | 33.27 | 35.29 | Pg C yr$^{-1}$ |
| POC export | 7.90 | 4.94 | 5.39 | Pg C yr$^{-1}$ |
| $CaCO_3$ export | 0.49 | 0.66 | 0.74 | Pg C yr$^{-1}$ |
| Opal export | 105.93 | 76.89 | 81.38 | Tmol Si yr$^{-1}$ |
| $f$-ratio | 0.20 | 0.15 | 0.15 | - |
| $CaCO_3$ to POC ratio | 0.06 | 0.13 | 0.14 | - |
| $T_{eff-1km}$ | 0.06 | 0.24 | 0.24 | - |





**Table 4.** Annual and cumulated ocean carbon uptakes simulated by NorESM1, NorESM2, and from observation-based estimates.

|  | NorESM1-ME | NorESM2-LM | NorESM2-MM | Observations |
|---|---|---|---|---|
| Annual uptake 1980s [Pg C yr$^{-1}$] | 2.03 | 1.80 | 1.85 | 1.8±0.8 (Denman et al., 2007) |
| Annual uptake 1990s [Pg C yr$^{-1}$] | 2.24 | 2.04 | 2.05 | 2.2±0.4 (Denman et al., 2007) |
| Cumulated uptake (1850-1994) [Pg C] | 127.91 | 111.02 | 107.82 | 111±21* (Gruber et al., 2019) |
| Cumulated uptake (1994-2007) [Pg C] | 33.68 | 30.57 | 31.04 | 29±5 (Gruber et al., 2019) |

*1800-1994





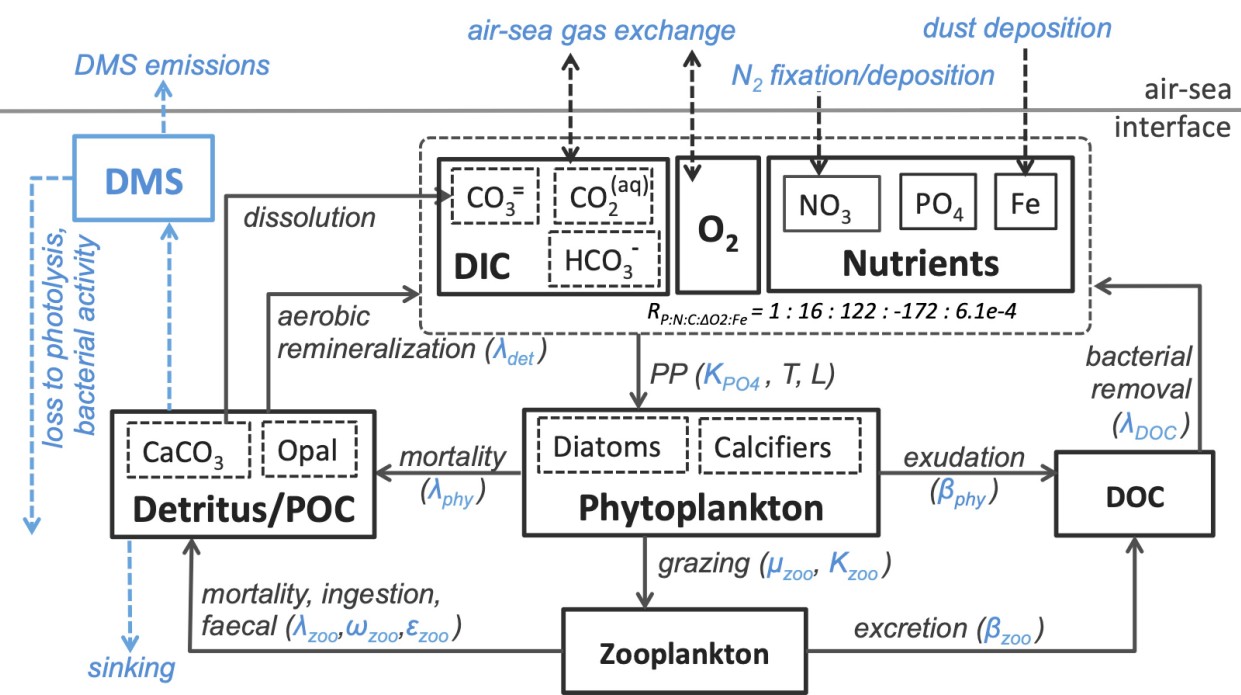

**Figure 1.** Schematic diagram of the ecosystem module in the ocean biogeochemical component of the NorESM2 model. The diagram is an updated version of a similar diagram in Six and Maier-Reimer (2006). Blue colors depict component, processes, and parameters that have been modified in NorESM2 relative to the previous model version NorESM1-ME.



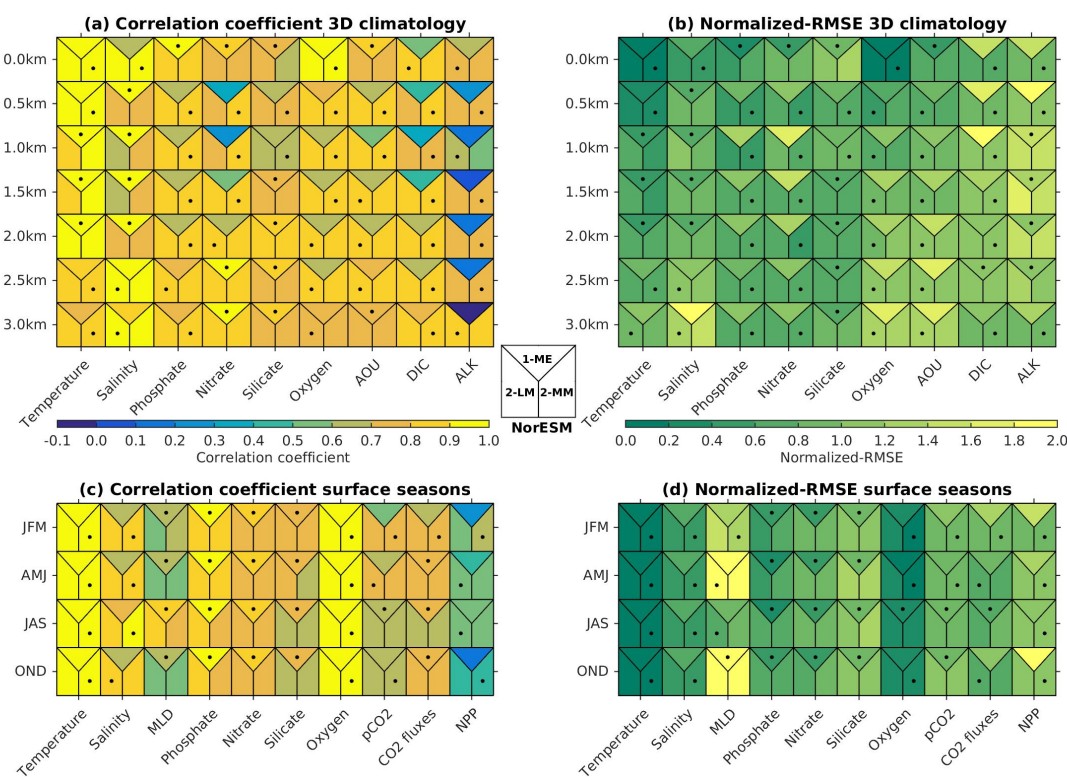

**Figure 2.** Summary of (left) spatial correlation coefficients and (right) normalized-RMSE between the observations and models (NorESM1-ME, NorESM2-LM and NorESM2-MM). In all (top) 3D fields, the metric values are computed over a 2D horizontal global domain ranging from the surface to 3km depth for 3D fields, while for (bottom) seasonal metrics, only surface values are evaluated. Each square contains three metric scores, i.e., for NorESM1-ME (top), NorESM2-LM (bottom-left), and NorESM2-MM (bottom-right). Black dots indicate which model performes the best.





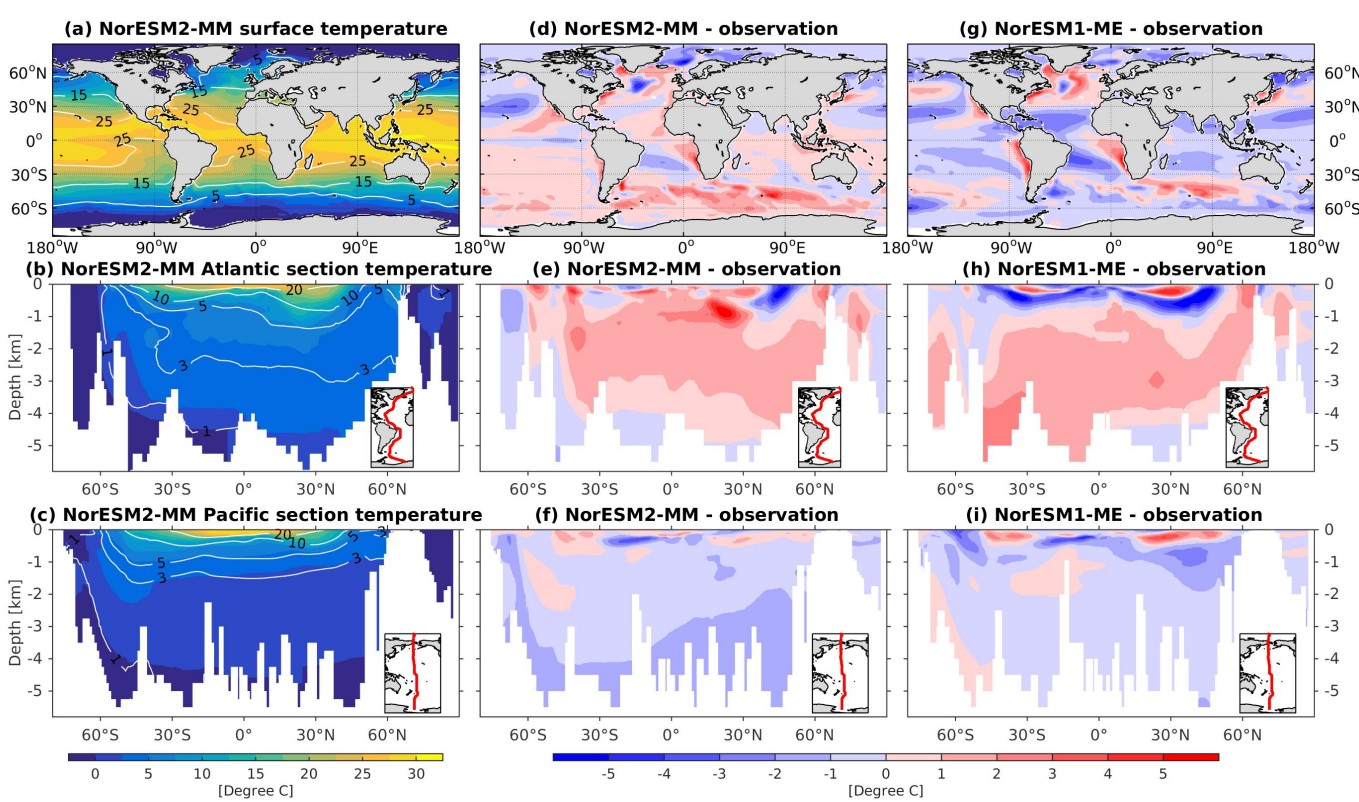

**Figure 3.** Climatological temperature values at the (a) surface and across the vertical sections of (b) Atlantic and (c) Pacific in NorESM2-MM (color-shadings) and from observations (contour lines). Difference between NorESM2-MM (or NorESM1-ME) and observations are shown in panels (d)-(i).



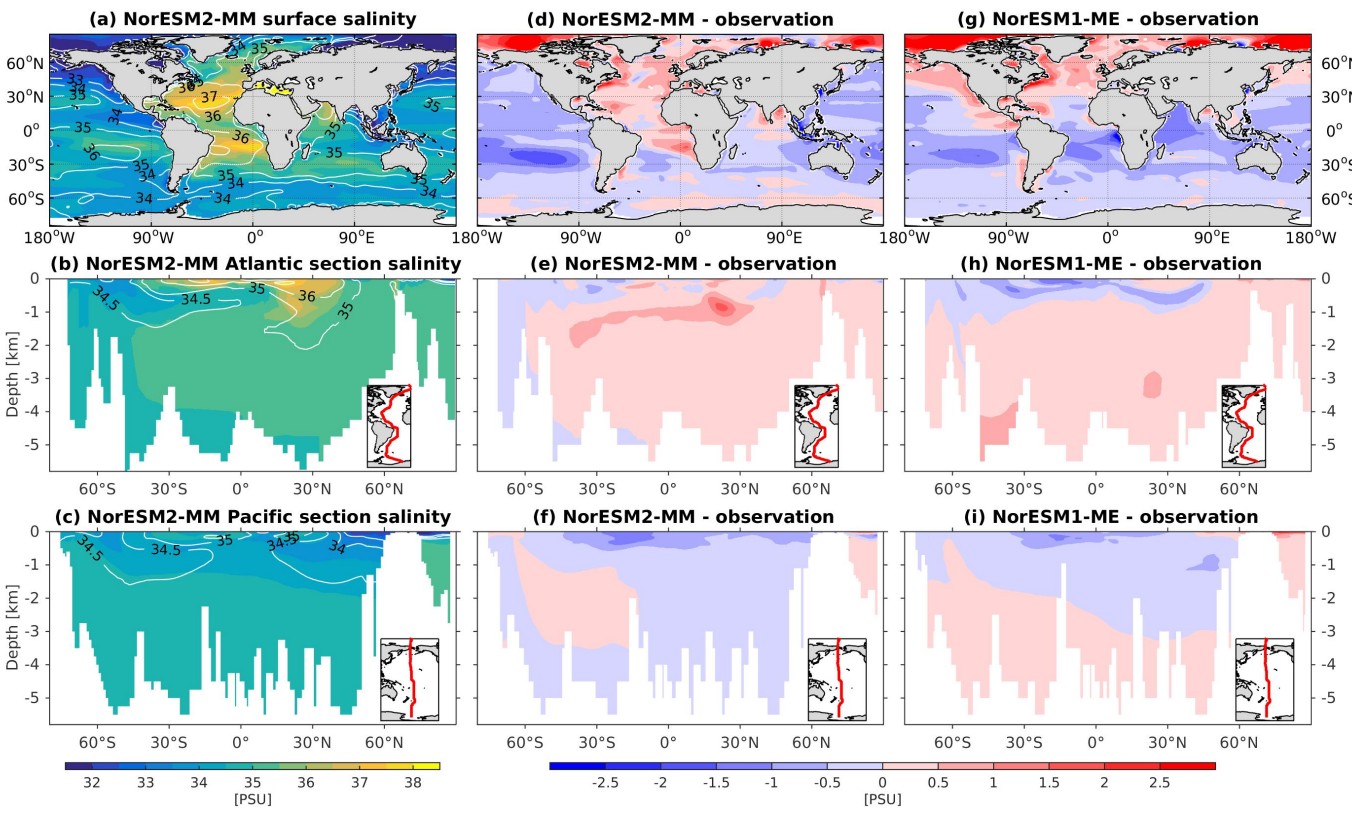

**Figure 4.** Climatological salinity values at the (a) surface and across the vertical sections of (b) Atlantic and (c) Pacific in NorESM2-MM (color-shadings) and from observations (contour lines). Difference between NorESM2-MM (or NorESM1-ME) and observations are shown in panels (d)-(i).

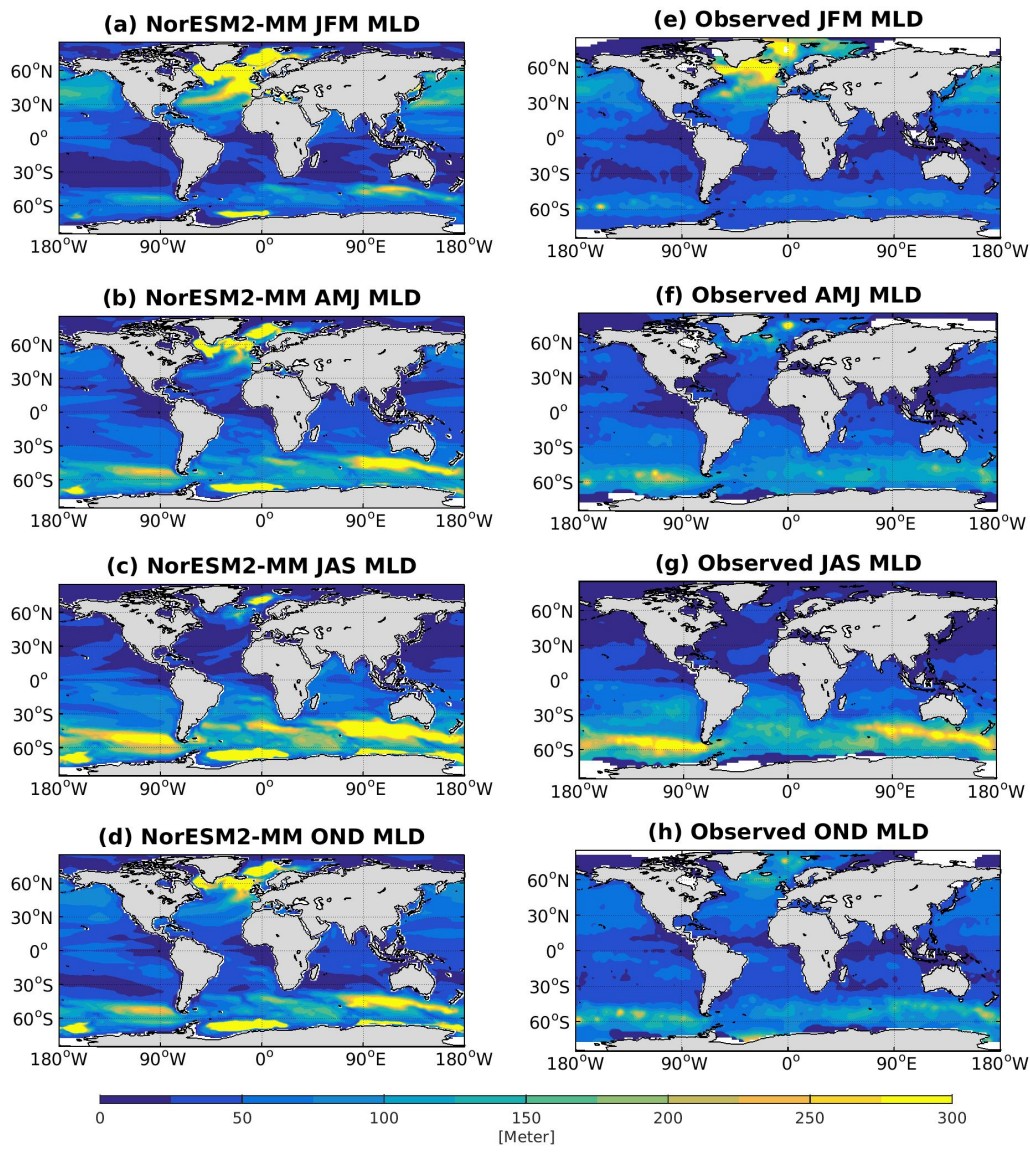

**Figure 5.** Climatological seasonal mixed layer depth as (left-column) simulated in NorESM2-MM and (right-column) estimated from observations.



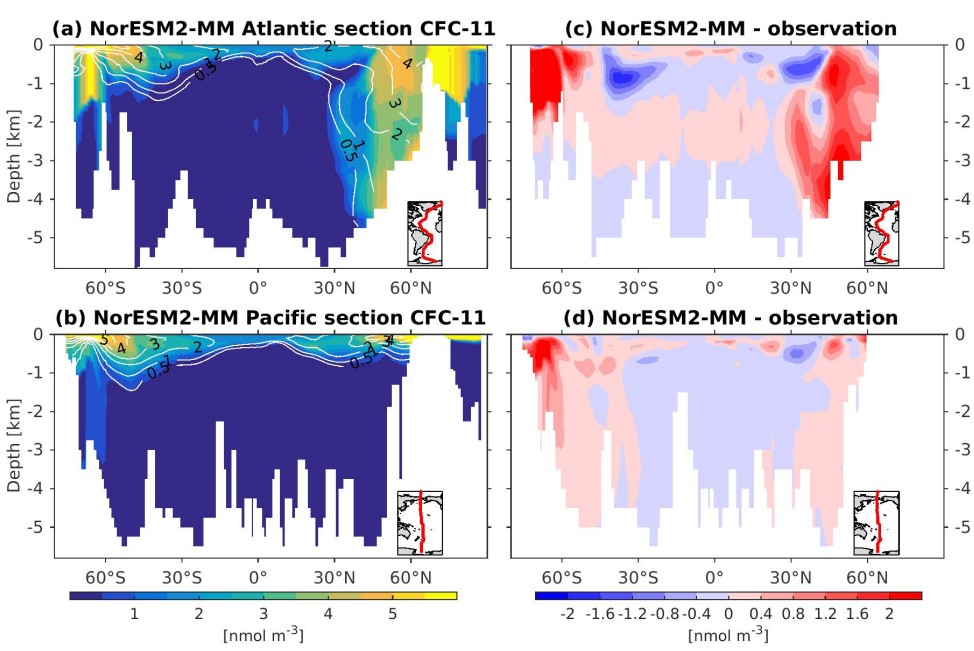

**Figure 6.** Concentration of CFC-11 across the vertical sections of (a) Atlantic and (b) Pacific from NorESM2-MM (color-shadings) and observations (contour lines). Differences between NorESM2-MM and observations are shown in panels (c)-(d).





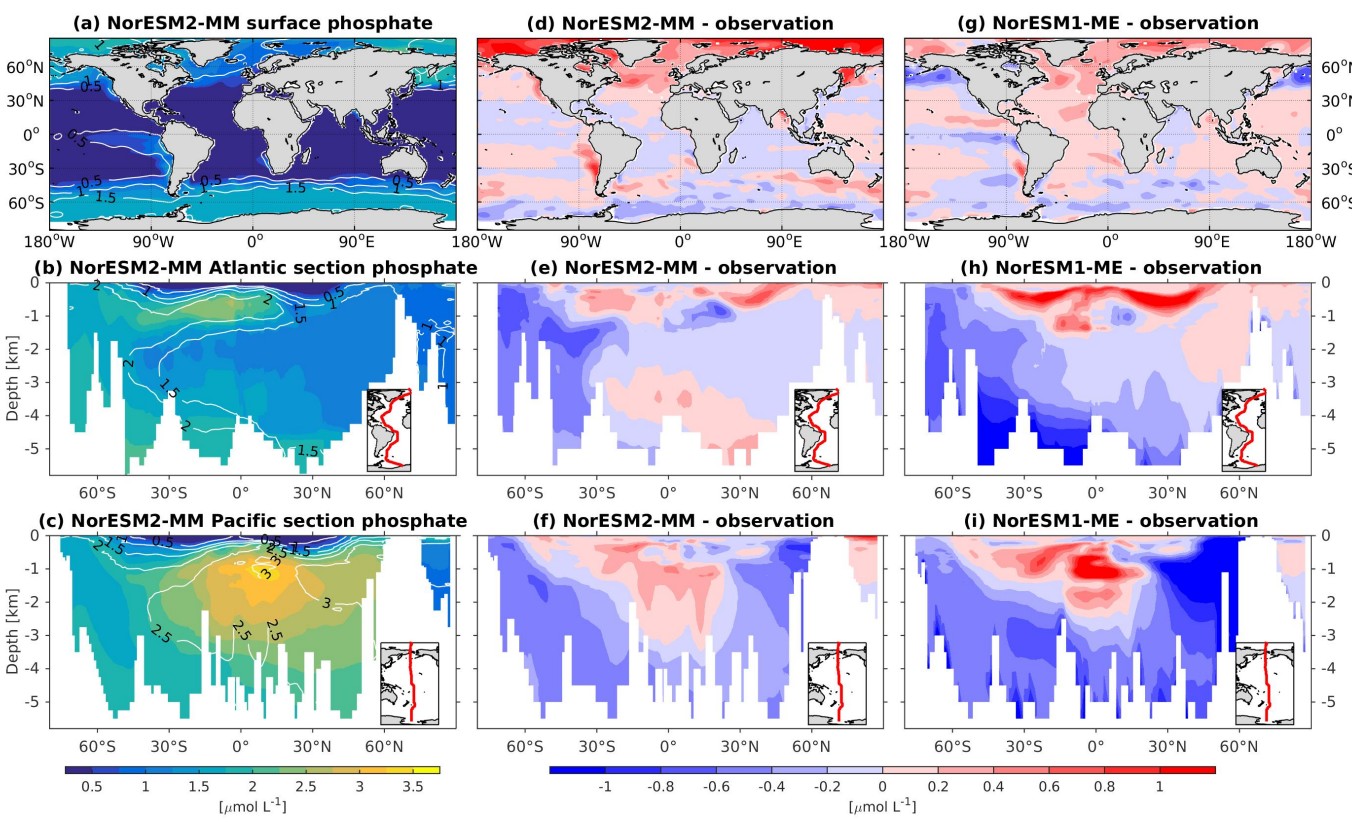

**Figure 7.** Climatological phosphate values at the (a) surface and across the vertical sections of (b) Atlantic and (c) Pacific in NorESM2-MM (color-shadings) and from observations (contour lines). Difference between NorESM2-MM (or NorESM1-ME) and observations are shown in panels (d)-(i).





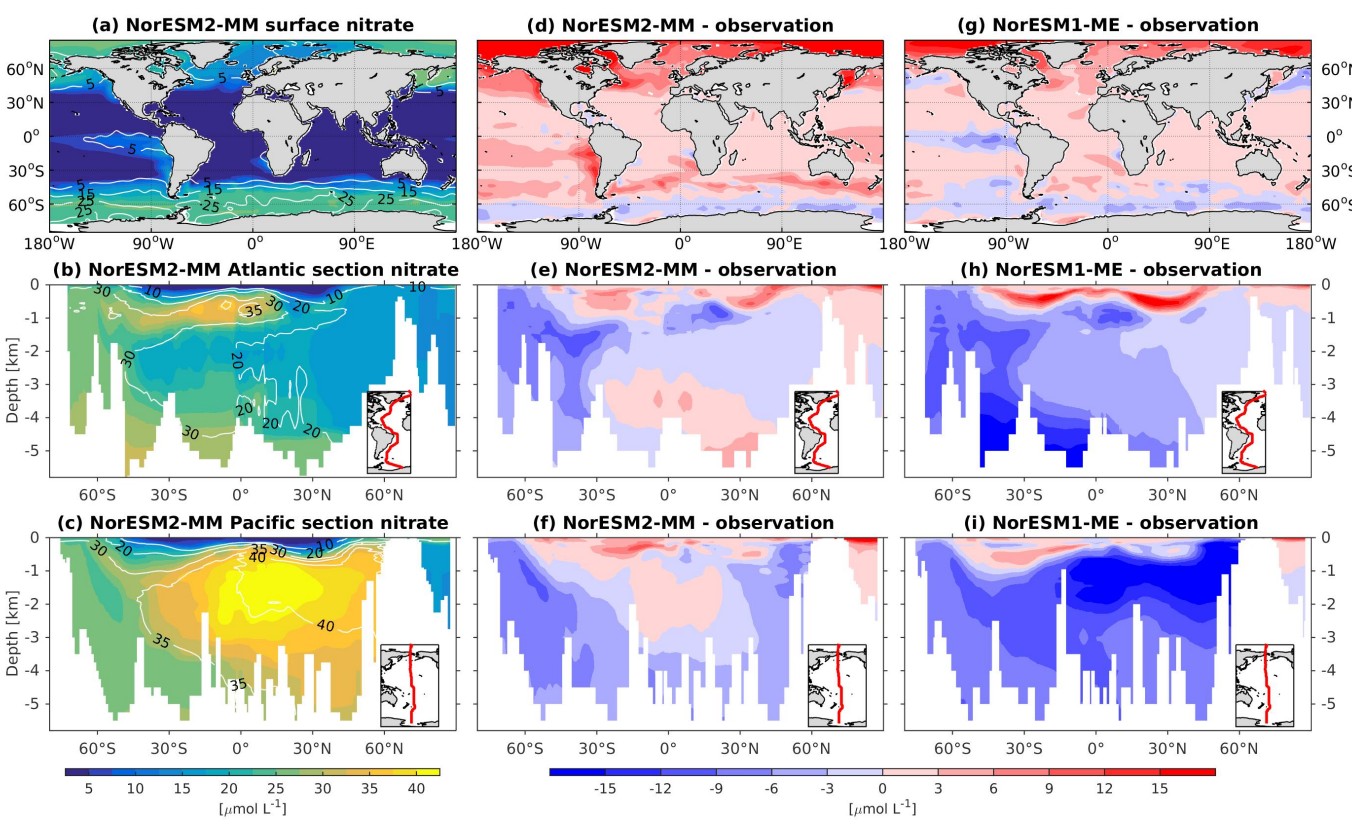

**Figure 8.** Climatological nitrate values at the (a) surface and across the vertical sections of (b) Atlantic and (c) Pacific in NorESM2-MM (color-shadings) and from observations (contour lines). Difference between NorESM2-MM (or NorESM1-ME) and observations are shown in panels (d)-(i).



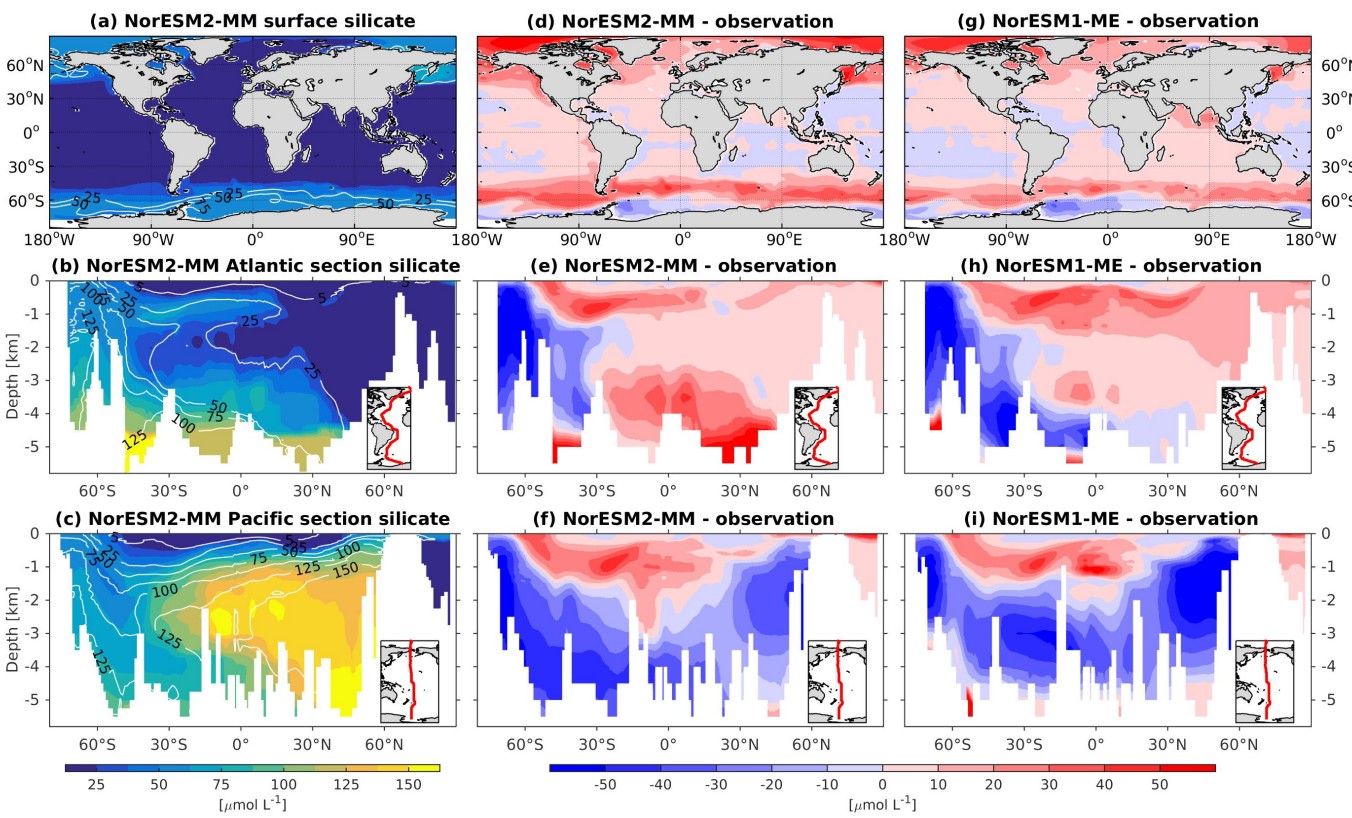

**Figure 9.** Climatological silicate values at the (a) surface and across the vertical sections of (b) Atlantic and (c) Pacific in NorESM2-MM (color-shadings) and from observations (contour lines). Difference between NorESM2-MM (or NorESM1-ME) and observations are shown in panels (d)-(i).





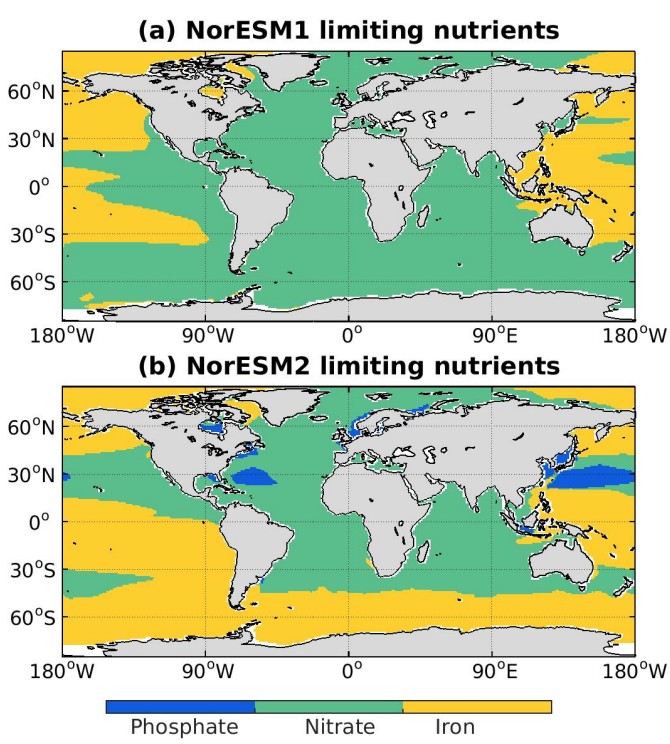

**Figure 10.** Maps of limiting nutrients for biological productivity as simulated in (a) NorESM1 and (b) NorESM2.

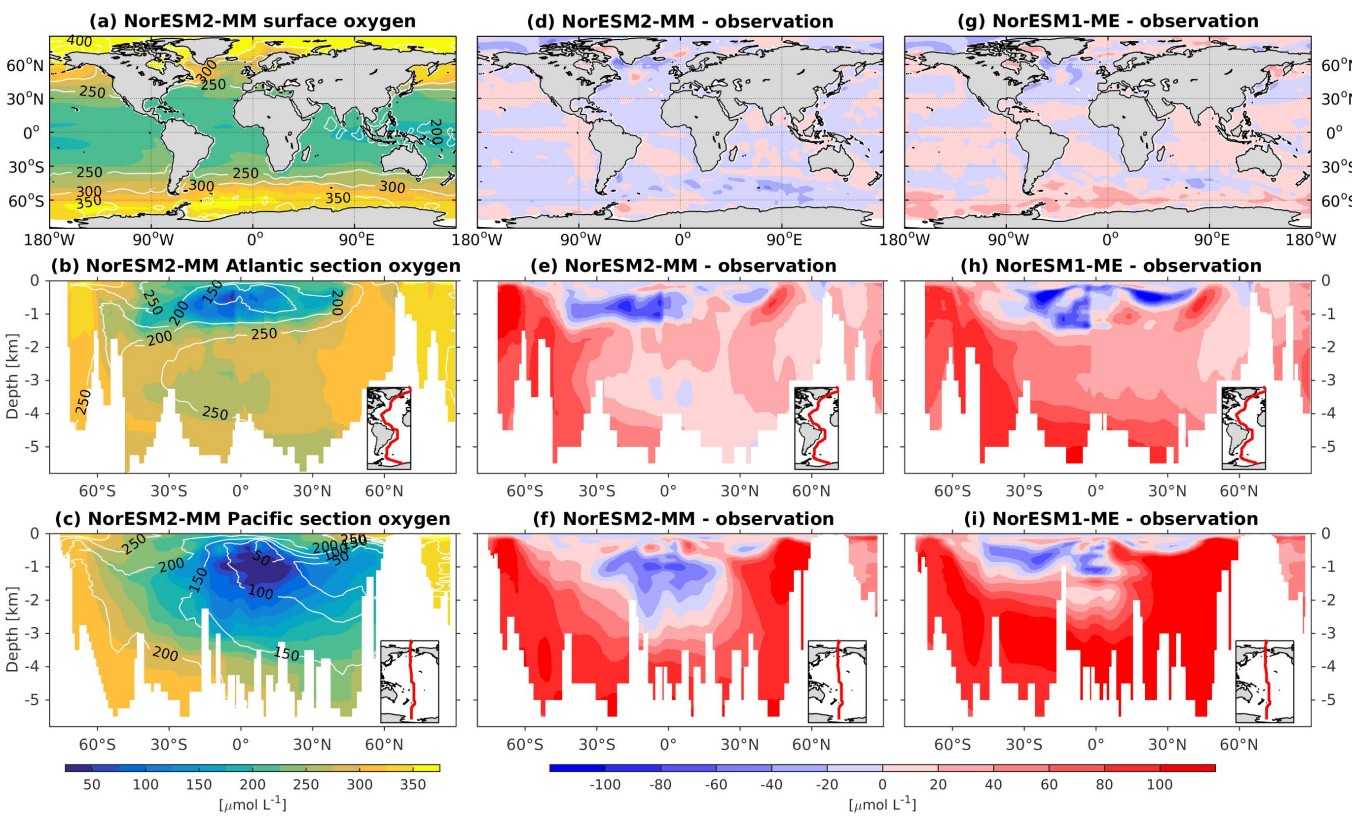

**Figure 11.** Climatological dissolved oxygen values at the (a) surface and across the vertical sections of (b) Atlantic and (c) Pacific in NorESM2-MM (color-shadings) and from observations (contour lines). Difference between NorESM2-MM (or NorESM1-ME) and observations are shown in panels (d)-(i).



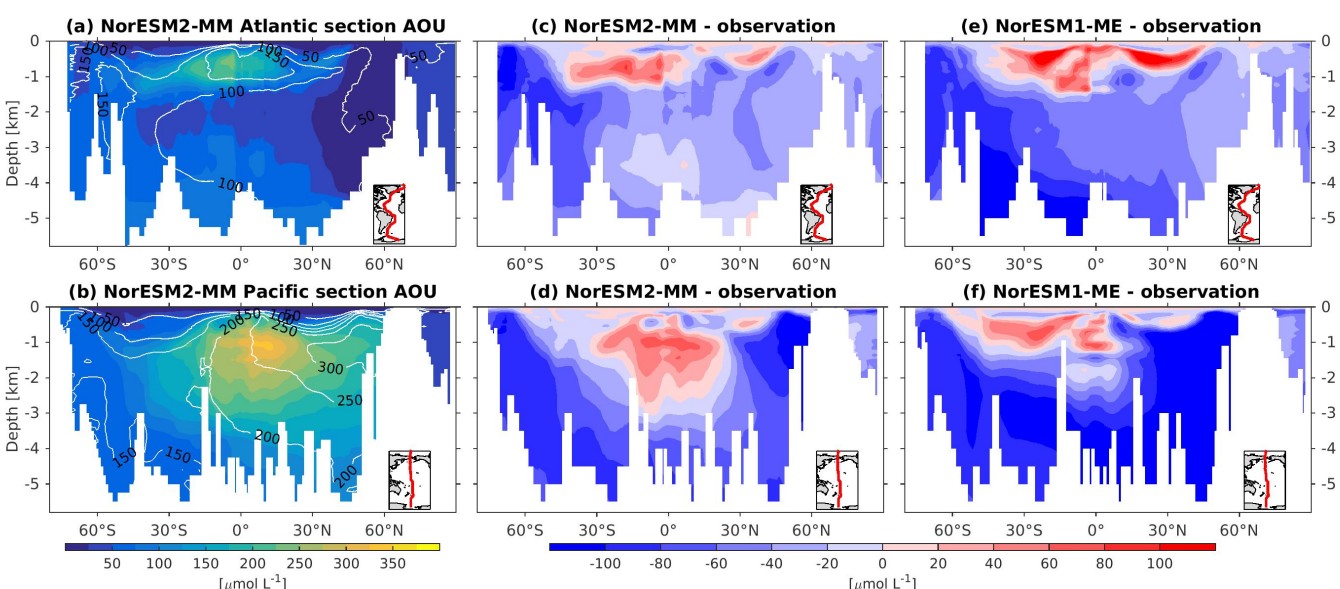

**Figure 12.** Climatology of apparent oxygen utilization (AOU) values across the vertical sections of (a) Atlantic and (b) Pacific in NorESM2 (color-shadings) and from observations (contour lines). Differences between NorESM2-MM (or NorESM1-ME) and observations are shown in panels (c)-(f).



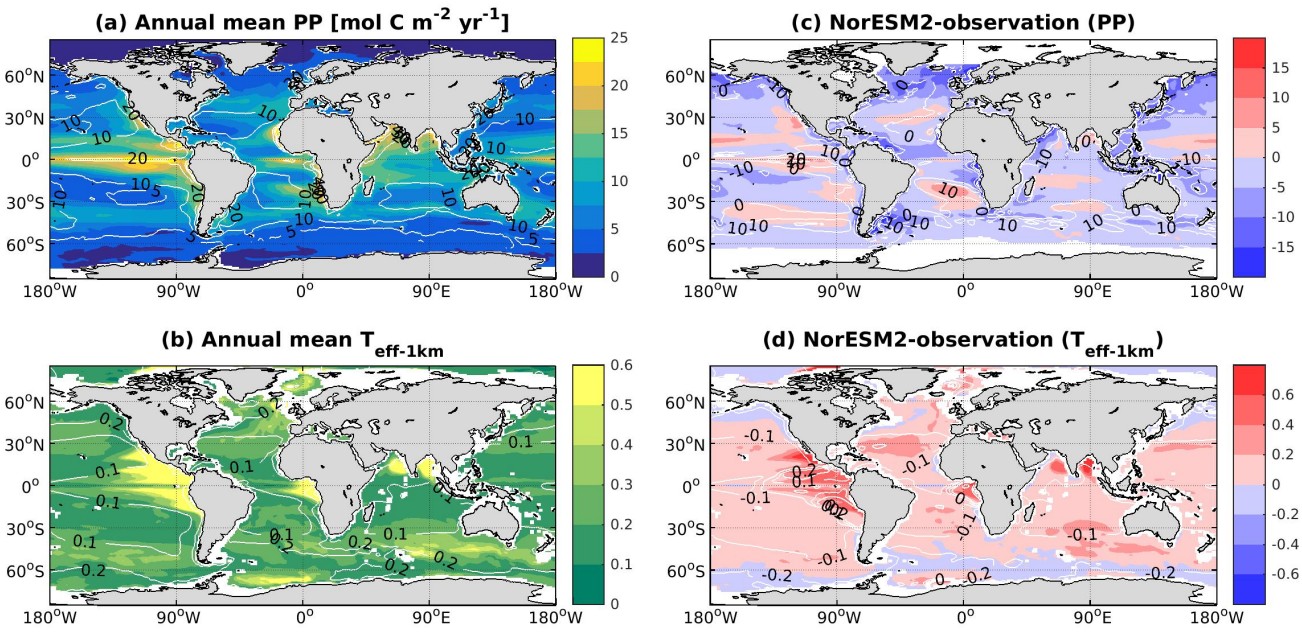

**Figure 13.** Annual mean (a) vertically-integrated contemporary primary production and (b) transfer efficiency ($T_{eff-1km}$) as simulated in NorESM2 (color-shadings) and from observations (contour-lines). Respective difference between NorESM2-MM (NorESM1-ME) and observations are shown in color-shadings (contour-lines) in panels (c)-(d).



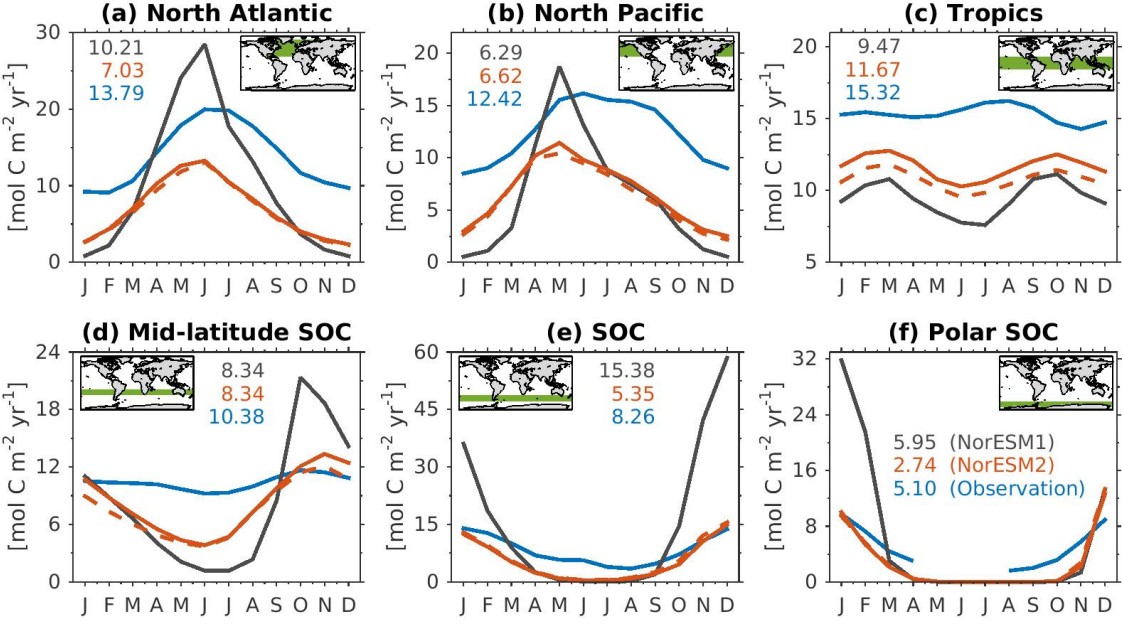

**Figure 14.** Mean seasonal cycle of vertically-integrated contemporary primary production in the (a) North Atlantic (north of 20°N), (b) North Pacific (north of 20°N), (c) tropics (20°S-20°N), (d) mid-latitude Southern Ocean (20°S-40°S), (e) Southern Ocean (40°S-60°S), and (f) polar Southern Ocean (south of 60°S). Shown are values from NorESM1-ME (black), NorESM2-MM (red) and observational estimates (blue). Numbers within each panel depict the regional mean values averaged for all months.





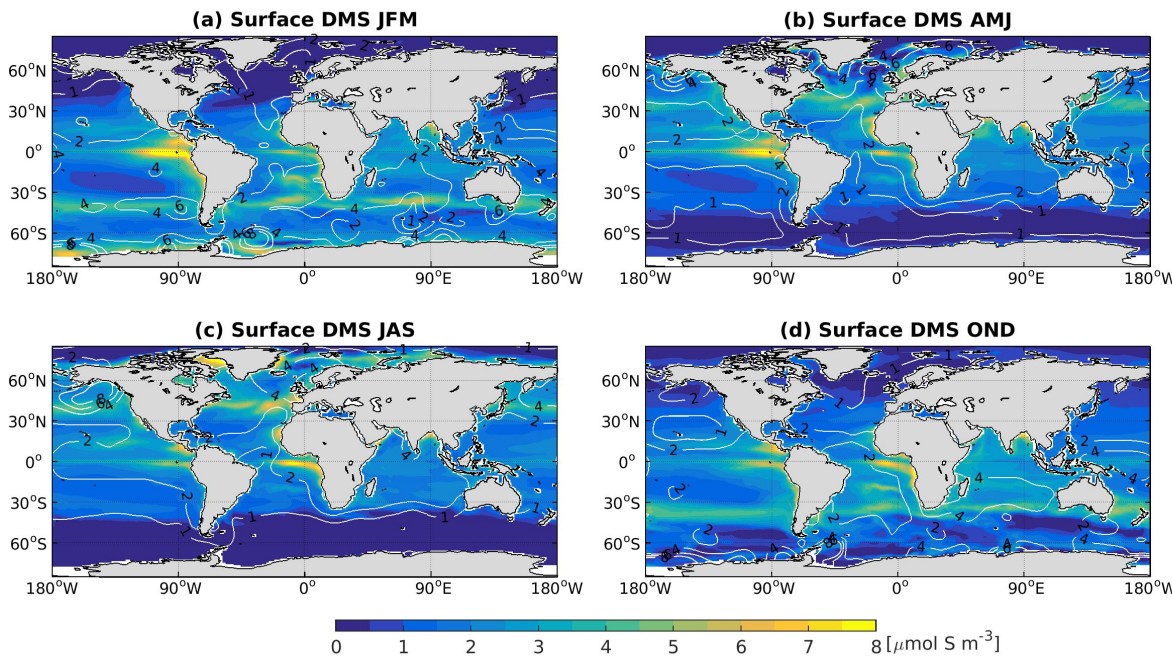

**Figure 15.** Seasonally-averaged surface concentration of DMS as simulated in NorESM2-MM (color-shadings) and from observations (contour-lines).



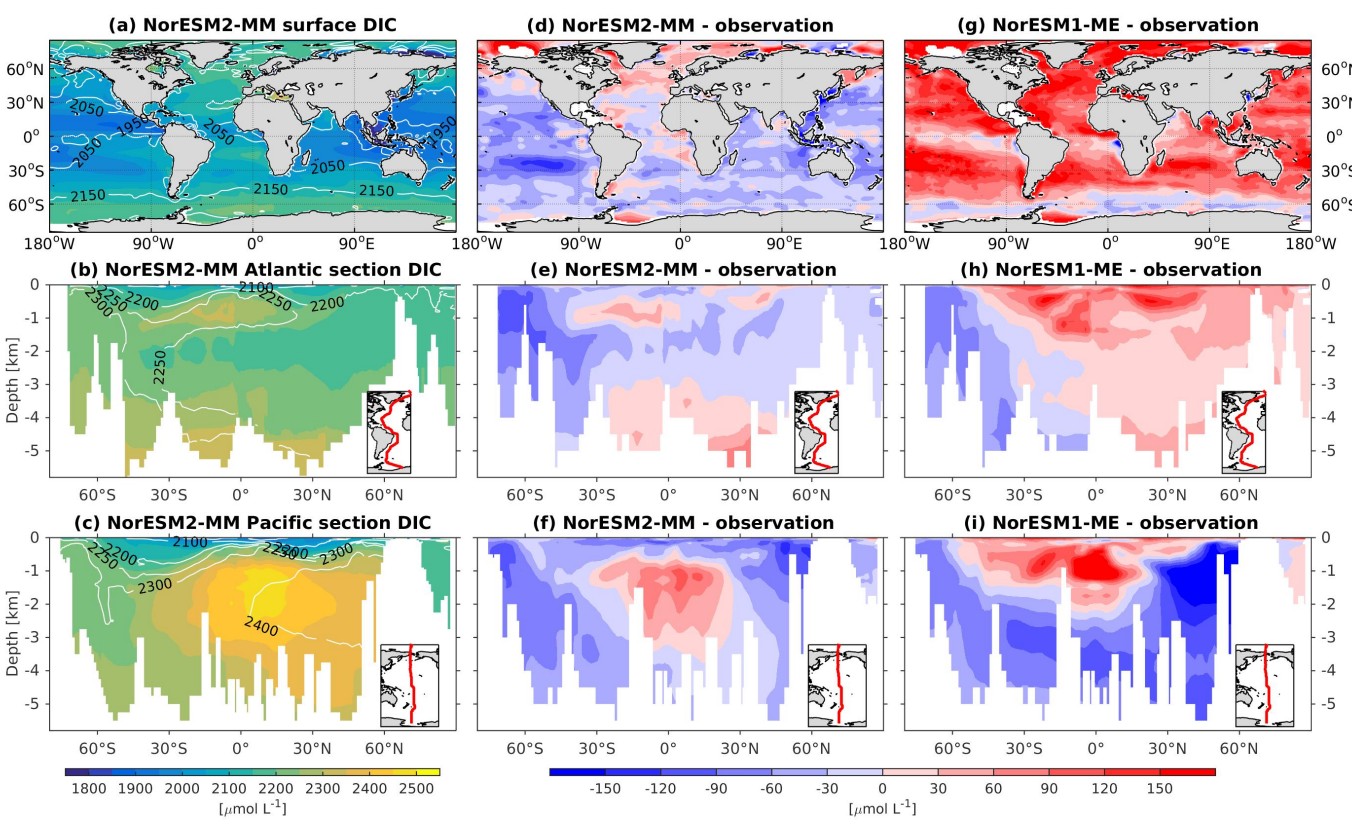

**Figure 16.** Climatological dissolved inorganic carbon (DIC) values at the (a) surface and across the vertical sections of (b) Atlantic and (c) Pacific in NorESM2-MM (color-shadings) and from observations (contour lines). Difference between NorESM2-MM (or NorESM1-ME) and observations are shown in panels (d)-(i).



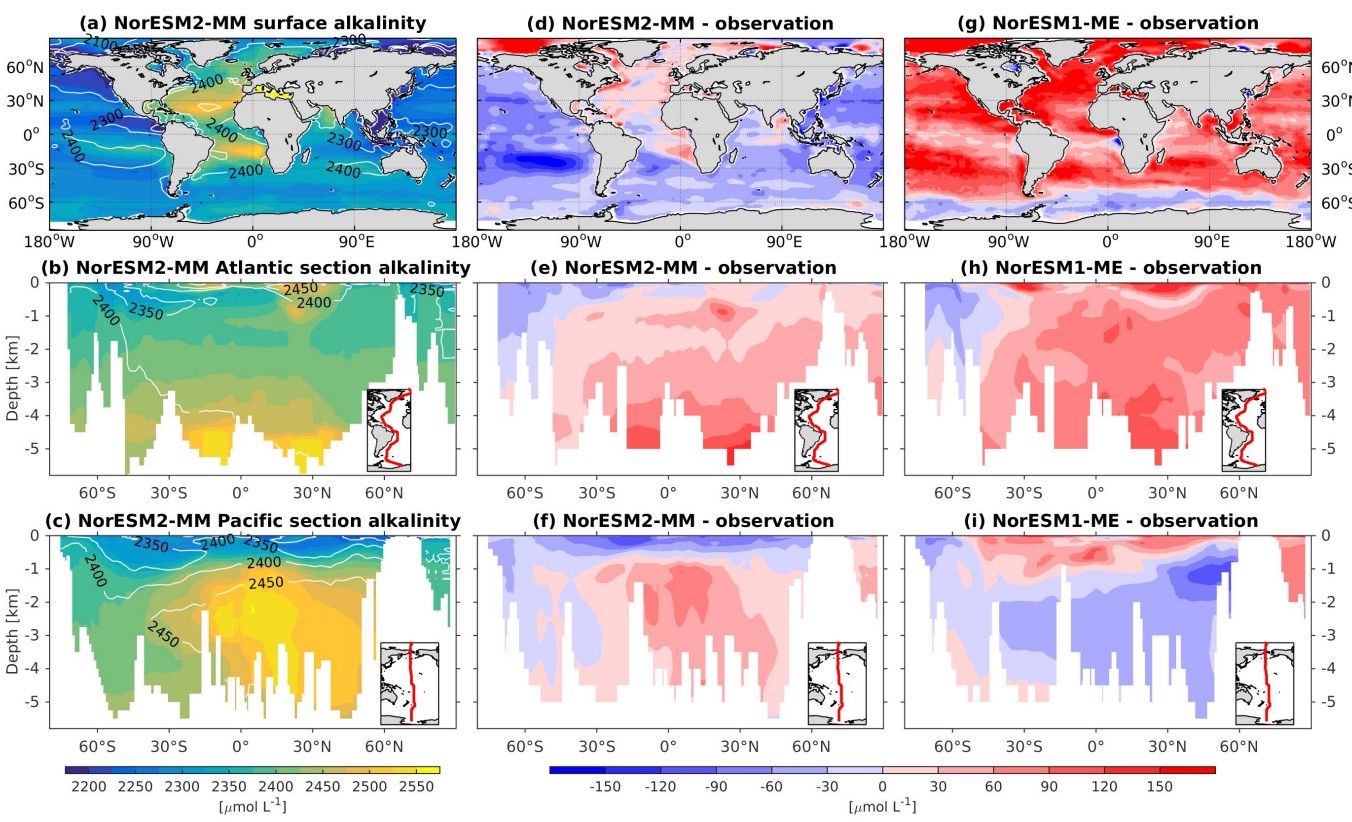

**Figure 17.** Climatological alkalinity values at the (a) surface and across the vertical sections of (b) Atlantic and (c) Pacific in NorESM2-MM (color-shadings) and from observations (contour lines). Difference between NorESM2-MM (or NorESM1-ME) and observations are shown in panels (d)-(i).





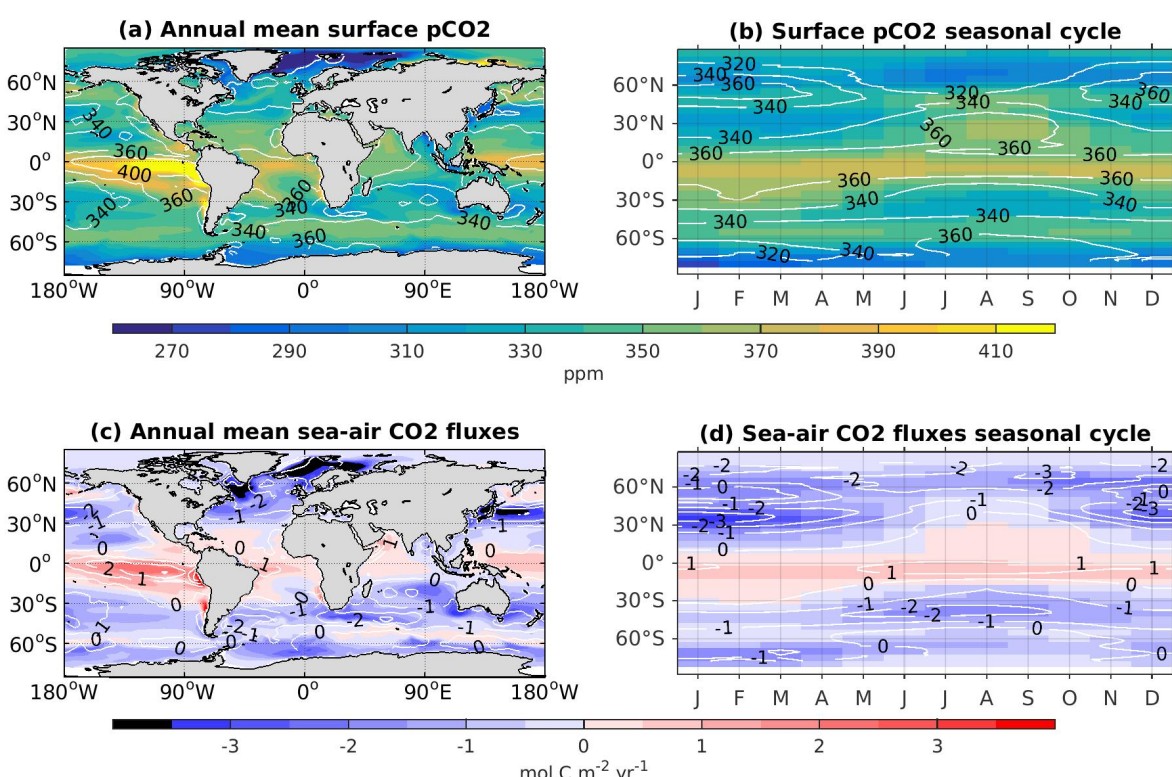

**Figure 18.** Surface climatology maps of (a) partial pressure of $CO_2$ and (c) sea-air $CO_2$ fluxes together with Hovmöller plots of zonally-averaged seasonal cycle of (b) surface $pCO_2$ and (d) sea-air $CO_2$ fluxes as simulated in NorESM2-MM (color-shadings) and from observations (contour-lines).



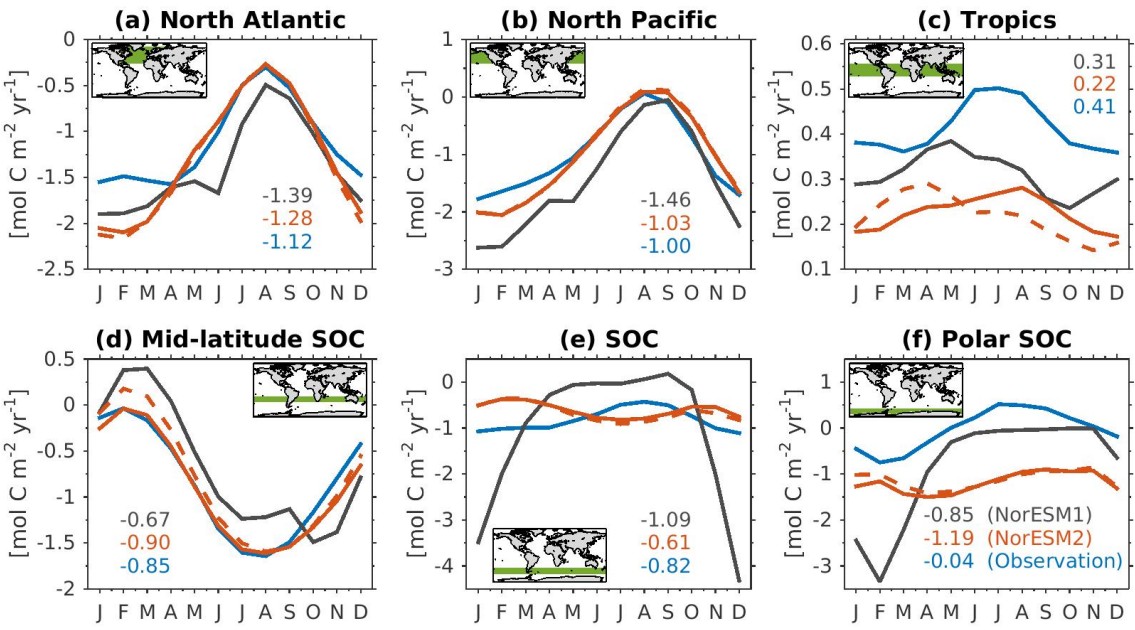

**Figure 19.** Mean seasonal cycle of vertically-integrated contemporary sea-to-air $CO_2$ fluxes in the (a) North Atlantic (north of 20°N), (b) North Pacific (north of 20°N), (c) tropics (20°S-20°N), (d) mid-latitude Southern Ocean (20°S-40°S), (e) Southern Ocean (40°S-60°S), and (f) polar Southern Ocean (south of 60°S). Shown are values from NorESM1-ME (black), NorESM2-MM (red) and observational estimates (blue). Numbers within each panel depict the regional mean values averaged for all months.





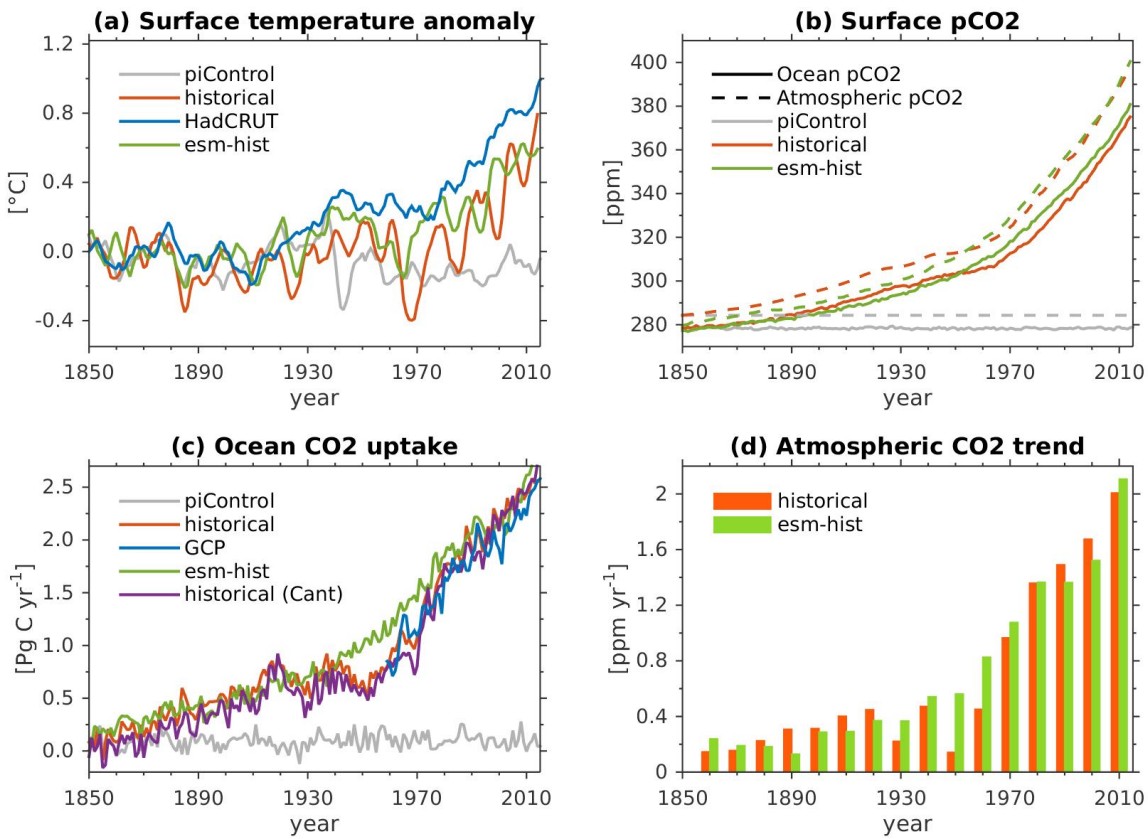

**Figure 20.** Time-series of global mean annual (a) surface air temperature anomalies (relative to the 1850-1879 periods), (b) surface ocean and atmospheric $pCO_2$, (c) ocean $CO_2$ uptake, and (d) atmospheric $CO_2$ concentration. Shown are values from NorESM2-MM (grey) preindustrial control and (orange) historical simulations. Green colors depict values from emissions prescribed historical simulations with NorESM2-LM. Purple line in panel (c) represent the anthropogenic carbon uptake estimated by substracting the natural carbon uptake from the total uptake (see also Sect. 3.2). Data for temperature and ocean carbon sinks are estimates from HadCRUT4 and the Global Carbon Project (Morice et al., 2012; Le Quéré et al., 2018), respectively, in blue colors.



**Figure 21.** Climatological $\delta^{13}C$ values at (a) 500 m depth and across the vertical sections of (b) Atlantic and (c) Pacific from NorESM2-OC (color-shadings) and observations (contour lines). Difference between NorESM2-OC and observations are shown in color-shadings in panels (d)-(f).





**Figure 22.** Climatological fields of the biological component of $\delta^{13}$C (i.e., $\delta^{13}C_{BIO}$) at (a) 500 m depth and across the vertical sections of (b) Atlantic and (c) Pacific from NorESM2-OC (color-shadings) and observations (contour lines). Difference between NorESM2-OC and observations are shown in color-shadings in panels (d)-(f).





**Figure 23.** Climatology $\Delta^{14}$C values at (a) 500 m depth and across the vertical sections of (b) Atlantic and (c) Pacific from NorESM2 (color-shadings) and observations (contour lines). Difference between NorESM2 and observations are shown in color-shadings in panels (d)-(f).





**Appendix A**

**A1**

*Author contributions.* JT, JS, AM, MB, CH, IB, and SG contributed to developing and implementing the iHAMOCC model code changes. JT, JS, MB, AM, AG, DO, and ØS designed, tested, and performed the experiments described in the manuscript. JT, JS, AM, and YH

5  analyzed the model simulations. All authors contributed to the writing of the manuscript.

*Competing interests.* The authors declare that they have no conflict of interest.

*Acknowledgements.* We acknowledge support from the Research Council of Norway funded projects EVA (229771), INES (270061), and KeyClim (295046), Horizon 2020 European Union's Framework Programme for Research and Innovation project CRESCENDO (Coordinated Research in Earth Systems and Climate: Experiments, Knowledge, Dissemination and Outreach, no. 641816), and funding from the

10  Bjerknes Centre for Climate Research. JT also acknowledges project Biodiversa (295340) and COLUMBIA (275268). High performance computing and storage resources were provided by the Norwegian infrastructure for computational science (through projects nn2345k, nn2980k, nn1002k, nn9560k, ns2345k, ns2980k, ns1002k, and ns9560k). We also acknowledge Johan Liakka for providing technical assistance and Jöran Maerz for the T$eff - 1km$ data.



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
