# Peer review of "Ocean biogeochemistry in the Norwegian Earth System Model version 2 (NorESM2)"

_Geoscientific Model Development, 2019_

## Referee Comment (RC1) · Joachim Segschneider (Referee) · 6 Feb 2020

Review of GMD-2019-347

Ocean biogeochemistry in the Norwegian Earth System Model version 2 (NorESM2)

by Jerry F. Tjiputra, Jörg Schwinger, Mats Bentsen, Anne L. Moree, Shuang Gao, Ingo Bethke, Christoph Heinze, Nadine Goris, Alok Gupta, Yanchun He, Dirk Olivie, Öyvind Seland, and Michael Schulz

In their study, the authors report on the further development of the ocean biogeochemical model iHAMOCC as component of the Norwegian Earth System Model version 2 (NorESM2). In particular, the changes of the model compared to the version used in NorESM1 are first described, and then the performance of iHAMOCC in NorESM2 is

evaluated in relation to observations and the predecessor model NorESM1 in a subset of CMIP6 experiments (control and historical).

Overall, the authors find that the numerous changes lead to a better match of the climatological mean state of various biogeochemical parameters with observations, and thus reduce the overall model bias.

The changes to the iHAMOCC model code are fairly substantial, ranging from the introduction of a depth-dependent settling velocity for detritus, the simulation of DMS emissions, updated riverine input (including nutrients and DIC/ALK), atmospheric nitrogen deposition, a changed formulation for nitrogen fixation and air-sea gas exchange, adjustments to the iron cycle and various ecosystem parameters, to the simulation of carbon isotopes (intended to be used in paleo studies), and the introduction of new 'preformed' and 'natural' tracers (the latter operating at pre-industrial atmospheric CO2 levels) for additional analyses. In summary this justifies a new documentation/publication in GMD.

The paper is generally quite well written, with some room for improvement mainly in the 'Results' section, the figure captions, and the supplementary material.

The supplementary material could be better justified and introduced, hardly any of the figures are discussed in the main text.

Also, the reader's life could be made easier by a more specific referencing to the figures (like pointing to the specific panel, not only the Fig., and perhaps the particular feature like 'blue curve in Fig X.x').

In summary I recommend publication of the manuscript after a careful revision addressing the below comments. Due to the length of the paper, my comments are quite numerous, but by nature I would consider the requested changes as minor revisions.

. .

==============================

Specific comments

p5 ln 19 and 23

better: 'atmosphere-ocean coupling' instead of 'ocean coupling'

Section 2.3 p6 ln 6-ln10

I was a bit confused by this (DMS production) description, in particular there seems to be an error in the use of degradation/production ?

after reading through several papers, I now assume production should be changed do degradation in line 10 - or it should be 'detritus production' of opal/CaCO3

In Six et al. 2013 the k refers to sulphur to carbon ratios in cells of opal/CaCO3

In Six and Maier-Reimer 2006 the gammas to degradation rates of opal/CaCO3

not clear to me what is meant in Eq. 2

please double-check and revise Section 2.3

p6 Eq (1) capital P in Phy irritating - I suggest to use 'phyto' to avoid mixing up with physics

also, is there a reason why some terms carry the 'DMS', others not?

p9 ln16/18

perhaps the ref. to Fig. 1 could be moved to the end of the para, or even toward the end of the section, otherwise one is trying to connect the following statements to Fig. 1

p10 ln25 I presume what is meant here is that preformed phosphate can be used to estimate the organic carbon pump, and preformed alkalinity to estimate the inorganic carbon pump? As the sentence stands now, either can be used to estimate both pumps.

–> Preformed phosphate can be used to quantify the organic (), and preformed alkalinity to quantify the inorganic () carbon pump ().

p10 ln26 (Eqs. 10-12; Bernardello et al.) is a bit misleading, perhaps (Eqs. 10-12, based on Bernardello et al. ) is better suited (the corresponding eqs. are 1-4 in Bernardello e al. 2014)

p11 eq.12 I do not quite follow how eq. 12 is derived (where does the +1 originate from?) +1 not in Bernardello et al. 2014 Eq. 4

p15 ln 18 'branched off into' is not a valid formulation. rephrase to sth like: the PI, control and historical exps are branched off from the spin-up. or 'the simulation is used as a starting point'....

p15 ln 20 see above

p16 ln24 rephrase 'at subsurface 500 m'

p16 ln25/26 'improvements ... in deviation' does not sound good.

–> improvements in agreement, better agreement

p18 ln 5 what is meant by : the simulated .... concentrations are mostly confined to the upper 1 km?

p23 ln 31 ct Consistent with the lower oceanic (than atmospheric) pCO2 partial pressure

(I guess partial pressure is meant here, since the oceanic growth rates are not shown in Fig. 20)

Fig. 2 since the panels are labelled a-d, better use 'a,c' in the caption instead of 'left', etc

Fig. 9 is never discussed in the text

Fig. 14 I am a bit sceptical about the term 'Southern Ocean' for latitudes between 20 deg S and 40 deg S. Why not just call it southern hemisphere mid latitudes.

The content description of the supplementary material is full of errors.

——————————————————————————

Technical errors

general:

Check all occurrences of 'to allow for' (which means to consider s.th. when planning for s.th., see e.g. dictionary.cambridge.org) and replace by 'allows' or 'enables us to' or similar

change all occurrences of 'insight to' to 'insight into'

check for sgl/pl and past/present mismatches

———————————————————————————-

in the following 'ct' stands for 'correct to'

p1 ln 5 correct 'allow for'

p1 ln 7 riverine 'input'

p1 ln 7 'are recently' does not make sense -> have recently been ...

p2 ln 6 remove 'us' (who is 'us' here?) or better reformulate whole sentence

p2 ln 8 remove 'us'

p2 ln 13 ct ...hardware systems, higher resolution

p2 ln 16 complex interplay between what and what?

p2 ln 21 remove 'through'

p4 ln 3 check use of 'implications on' (correct: implications of sth. for sth.)

should be 'consequences for', 'impact on', or similar

p4 ln 11 ct 'insight into the ocean's role' ....

p4 ln 26 ct ....'in Section 4.'

p4 ln 28 ct ....'in Section 6.'

p4 ln 32 delete 'for'

p5 ln 2 ct we also now apply (or we also applied)

p5 ln 29 ct 'closer to' (or better, 'which is within the range'....)

p6 ln 24 ct ... an early version of 'the' Global-NEWS model ...

p7 ln 15 ct .... and 'are' added to the nitrate pool .... (otherwise it is not clear if this is
only assumed)

p7 ln 16 ct Particle export (without plural s)

p8 ln 1 ct ....interior biogeochemistry 'using' the different .... (not 'in')

p8 Eq. 4 dot between mu and max misplaced

p8 ln 14/15 change to: 1.25 moles dissolved oxygen and 1 mole of alkalinity...

p9 ln 11 ct where the strength 'of' this ....

p 11 Eq. (12) dot after 0.5 misplaced

p11 Eq. (13) remove leading dot

p11 ln24 insight into

p11 ln29 correct 'allow for'

p12 ln 5 remove 'to' before (ii)

p12 ln10 correct 'allow for'

p13 ln 27 ct ... different parameterizations, (add s)

p14 ln 16 ct , as follows (add s)

p14 ln 25 ct 'a quasi-equilibrium state' (remove pl. s)

p14 ln 30 ct ...carbon counterpart of 'the' respective ...

p15 ln 1 ct ...applied 'to' organic ... (not 'for')

p15 ln 3 –> atmospheric pCO2

p16 ln 8 correct 'the the'

p16 ln11 either 'the majority' or 'the major part' (but not 'majority part')

p17 ln 29 ct 'In the subpolar North Atlantic'....

p17 ln 31 ct ....'a' deep MLD ....

p18 ln 5 delete 'depth' after upper 1 km

p18 ln 16 ct .... material being ....

p18 ln 17 ct ... 'at' intermediate depth

p18 ln 18 ct ... 100 and 1500 m depth).

p18 ln 20 ct ... concentrations of all nutrients ...

p18 ln 27 ct ....NADW as the main watermass ...

p19 ln 3 ct ...(see also Fig 8i). - check if 8i was meant, 8e is Atlantic NorESM2

p19 ln 8/9 ct: ...low levels.... limit the phytoplankton growth.

p19 ln 21 correct 'Fig. 9' to 'Fig. 12'

p19 ln 29 ct ... and at high latitudes during summer months.

p20 ln 3 ct ...the spring blooms....

p20 ln 4 delete 'during the boreal spring months' (redundant in sentence)

p20 ln 9 coastal areas (not grids)

p20 ln 20/21 move ref to Weber after 'biogeochemistry' ct ....still simulates too high transfer efficiencies...

p20 ln 22 ct ...are comparable with observations.

p20 ln 26 correct 'Fig. 2c' to Fig. 2d

ct ....simulates a lower ...

p21 ln 8 ct ...at the lower end...

p21 ln 24 ct As at the surface...

p22 ln 12 ct ...translates into stronger carbon sinks...

p22 ln 20 ct of DIC-rich deep watermasses...

p22 ln 21 ct ...strong bias ... is considerably reduced, ....

p22 ln 22 ct ... is approximately reversed compared to observations:

p22 ln 26 ct Nevertheless, the ....

p23 ln 25 ct 0.7 oC, comparable to that from obs...

p23 ln 26 ct the warming in esm-hist ....

p23 ln 30 ct (i.e., is lower than)...

p24 ln 7 ct For the 1980s and 1990s ...

p24 ln 9 correct allowing for

p24 ln 15 ct ...by the ocean for 1850-1994 and 1994-2011 is ....

p24 ln 27 correct allow for

p24 ln 28 ct , and (iii) carbon isotopes that can be used e.g. ....

p25 ln 8 ct the equatorial Pacific OMZ

p25 ln 9 ct ...Southern Ocean, and the equatorial and North Pacific.

p25 ln 14 replace allowing for by 'resulting in'

p25 ln 20 ct simulates a considerable bias

p25 ln 25 ct the $CO_2$-fluxes' seasonal cycle...

p25 ln 26 attention to (not of)

p25 ln 30 ct ... depth of the equatorial Pacific ....

p25 ln 33 ct penetrating too far north..

p26 ln 1 ct biogeochemical components in ESMs...

p26 ln 2 ct the mean climatological state. However, .... (check if the references have to be moved)

p26 ln 3 ct a lot of effort

p26 ln 5 replace allowed for by e.g., 'provided'

p26 ln 15 correct allow for

p26 ln 19 ct ..currently, we use fixed particulate organic carbon emissions ...

p26 ln 27 ct to investigate the sensitivity of ....

p26 ln 35 ct as the results of more complex model simulations

Table 1: ct ... ecosystem parameterisations that have been changed....

Table 2: ct ... simulation periods over which their climatological values have been averaged.

ct Average of the three remote sensing products...

Table 3: ct Annual mean biology-related metrics... (not only primary production is listed)

Fig. 1 ct Blue depicts components, processes, .... (there is only one blue)

Fig. 3 caption: ct 'Differences between ... are ....

also Fig. 4, 7, 8, 9, 11, 13, 16, 17, 21, 22, 23

Fig. 14 averaged over all months.

also Fig. 19

Fig. 20 ct relative to the 1850-1879 period, (b)...

Green depicts results from the .... simulation with NorESM2. The purple line in panel (c) represents .... (only one green, only one esm-hist with NorESM2-LM)

---

## Referee Comment (RC2) · J.,. Palmieri (Referee) · 22 Feb 2020

General comments:

This paper "Ocean biogeochemistry in the Norwegian Earth System Model version 2 (NorESM2)" is a description and evaluation of the Norwegian Earth system model that participates to the CMIP6 exercise. This kind of paper is extremely useful to the CMIP community that will use and compare the models taking part to the project, and who need to know the different model characteristics, mains results, biases. I appreciate the massive work needed to write this kind of paper, which are not the most exiting to write, but needs to be written. Although the paper would need some more polishing, the results are well described and clearly presented.

[Figure]

The paper mainly describes the improvements made to the model since the previous version used for the CMIP5 exercise, comparing both to observation, and showing that the new version is mainly better the its predecessor. Some difficulties come from this presentation choice as sometime the authors consider that some description being made in the NorESM1 description papers, the reader is assumed to know them. In results, some basic description are almost missing. For example, the ocean grid description of which we only know it is tripolar.

Apart some little corrections of that kind, listed bellow, which should hopefully help to improve the paper, i have one concern about the isotope run. The isotopes are run on an ocean-only run, with a lower resolution grid than the coupled run, and a different atmosphere to force the run. I don't think this run should be used to evaluate the coupled run, unless the authors show that both model DIC steady states are comparable, or mention in the isotope results paragraph that these results are only informative, because from a different run. This needs - at least - to be reminded to the reader. Written as it is, the isotope results are in the middle of historical and esm-hist results. The reader can easily misunderstand and think they all are from the same simulation.

Overall, i have positive feelings about this paper. Most comments are minor like rephrasing or asking for missing details, what should translate in minor revision.

List of specific remarks :

Being not native english speaking, i will not be the best one to correct the English formulation, if needed. Fortunately, from what i have seen, the paper looks well written.

1 Intro

P2 l1 : rephrasing : absorber of heat and the greenhouse gas CO2 to absorber of heat and of the greenhouse gas CO2 or absorber of heat and CO2 greenhouse gas ?? But the one in the paper sounds weird to my ears.

P2 l30 to P3 l3 : Is there an equivalent paper to this one for NorESM1 ? It seems

there is no reference description paper for the version 1 of the model. You list different papers of different development stages, but which one is the one that would describe the version of NorESM1 you use in this paper ?

P3 l3 : "which will contribute to CMIP6". You can probably write it at present tense.

P3 l24-33 : This can not replace having 2 separate runs. Which nutrient does the biology see ? How important could that be within the scenario ? (PI carbonate system evolves with biology that feels high CO2 world, pH,... ) probably not important for historical runs. that's a problem we faced within OMIP, and we could not have a definite answer (should run both to be sure...).

P3 l34 to p4 l6 : 2 remarks : you say " The only external source [in NorESM1] is through atmospheric nitrogen fixation" you mean there is no dust (iron/Si/P) deposition in NorESM1 ? from this paragraph it sounds like you only add riverine nutrients, but you talk of dust somewhere else... could you clarify ? Also, what is included in riverine nutrients ? (actually you answer later on - maybe add a "see paragraph 2.4 for details")

2 Model changes and improvements

2.1 : you give several details in this paragraph but we miss some details of the ocean grid, like the basic ones: resolution, number of vertical levels,...

p7 l4-7 : i don't understand what you explain there to get the riverine fluxes of DOC and POC, maybe you could try to make it clearer ?

P9 paragraph 2.9 – So aerial dust deposition was already in NorESM1 - don't forget to add it P3.

P9 l11 - typo : strength *of* this

3 New tracers

P10 l12 : "during the spin-up" might be misleading. Maybe "for the spin-up" or "to start the spin-up" or just "initialized to zero" might be enough.

P12 l14 : "... and CaCO3 formation, but the latter is neglected in our implementation". Why ? any idea of the impact ?

P14 l24 to P15 l2 : The method to initialize the C isotopes is very clever, and must save a lot of computing time. But i wonder the impact of initializing them this way. Don't you introduce bias compare to a proper model integration? Do you note drifts in the isotopes concentration during the historical period ?

4 Model simulations

P15 l19-23 : might need a bit of rephrasing here. explanation about (3) are a bit confusing. i would suggest to call it emission-driven simulation as (1) and (2) are also "esm" runs. Or if you absolutely want to call it esm-... explain straight what it is like " (3) esm-experiments, the atmospheric CO2 is prognosticaly computed from ocean-atmosphere and land-atmosphere CO2 -fluxes, as well as from prescribed anthropogenic emissions (for the historical period). (3) is composed of a esm-piControl-spinup simulation (100 model years) which is then branched off into (3a) an esm-piControl simulation (for 250 years), and (3b) a transient esm-hist (years 1850-2014) simulation." But calling this experiment "esm" is confusing for the reader.

P15 l30 : "The atmospheric forcing of the spin-up is the CORE normal year forcing (Large and Yeager, 2004), which represents a climatological mean year with a smooth transition between end and start of the year" Why did you use the CORE climatology ? that's a surprising choice, it removes all substantial link between your C-isotope/ocean-only run and the esm. Using CORE remove all potential isotope insights on the esm runs... there are other ways that could provide this insight possibility: you could extract atm fields from the end of the spin-up and use them as forcing field for this experiment, for example. Using CORE, the ocean dynamics will be different from the coupled, the spin-up steady state will have nothing to do with the esm's steady state. Already changing the ocean resolution affect the dynamic, but resolution plus forcing completely change the ocean. I don't think you can use the isotopes to evaluate the esm run with

such a different set-up.

P15 l30 : Subsequent question concerning the isotope run. What happens after the spin-up ? Is there anything special to run the historical period ? Do you change the atm forcing field (i really wished you used atmospheric fields extracted from the coupled historical run)? or do you simply follow the historical atmospheric CO2 records for the last ~150years ?

5 Results

5.3 Salinity - P17 - no ref to Fig. 4

5.6 Nutrients - P18 - No ref to Fig. 7 and 9 The fig 2 is understandably well discussed as it is a very instructive figure, but all following figures are almost forgotten, and sometimes not introduced at all.

P19 l3 - if you refer to NorESM1, you should point to fig 8h and 8i

5.7 Dissolved oxygen - P19 - No ref to Fig. 11 .

P19 l21 - Fig. 9 is the silicic acid ; AOU is Fig. 12

5.12 - P22 : The Analysis done here are really great, but coming from a stand alone run forced with a different forcing field, these results do not inform about the main NorESM2 MM and LM runs i am afraid.

5.12 and 5.13 - double-check the figure order, you switched both section figures .

6 Figures

Fig.1 some features are missing. For instance the riverine inputs (with the different materials they contain). Also the silicic acid is missing.

Fig. 10 : The modeled Diatoms are never Si limited ? You say biological productivity, how is that done ? as both phyto are not necessarily limited by the same nutrient. Might be good to say 2 words in the figure caption about how you calculate that.

Fig. 13 : I have difficulties to realize what's going on with contours superposed to colormaps and showing different things. It is OK-ish on vertical sections, but on global horizontal maps it is difficult to appreciate the differences, and spot what we should see. it is not straight forward. could you add a colored pictures for NorESM1 so it is not superposed as contour, for fig 13, 15, 18. I would ask to do that for all fig of that kind, but that would mean almost all pictures... i really find it difficult to interpret that way. it save space, but doesn't make it simpler for the reader, i think.

Fig 14. and all alike : what is the difference between the solid red and dotted red lines ? i guess the dotted one is NorESM-LM but it is not written in the caption. you probably can remove the dotted line as there is no reference to it at all.

---

## Author Comment (AC1) · 22 Mar 2020

We thank Dr. Joachim Segschneider for his prompt and very thorough review of our manuscript. We have now addressed all of his comments (in **bold**) with point-by-point (i) responses (in *italics*) and/or (ii) description of changes made in the revision (in blue) detailed below. Changes in the revision can also be seen in the attached revised manuscript with highlighted changes (red and blue depicts older and revised version, respectively).

**The paper is generally quite well written, with some room for improvement mainly in the 'Results' section, the figure captions, and the supplementary material. The supplementary material could be better justified and introduced, hardly any of the figures are discussed in the main text.**

*We agree with this assessment and have accordingly revised the 'Results' section, figure captions and supplementary material.*

In the revised manuscript, we have made some improvements in the abovementioned components of the manuscript, add more references to the (revised) supplementary materials. Since many of the figures in the previous supplement materials, especially those showing results from NorESM2-LM version, closely resemble those of NorESM2-MM in the main manuscript, we have removed these figures.

**Also, the reader's life could be made easier by a more specific referencing to the figures (like pointing to the specific panel, not only the Fig., and perhaps the particular feature like 'blue curve in Fig X.x'). In summary I recommend publication of the manuscript after a careful revision addressing the below comments. Due to the length of the paper, my comments are quite numerous, but by nature I would consider the requested changes as minor revisions.**

*We agree and have followed your advice accordingly. We've gone through the revised manuscript and inserted specific figure references where they fit.*

Changes are applied throughout the manuscript, especially in the Results section.

**Specific comments**
**p5 ln 19 and 23 better: 'atmosphere-ocean coupling' instead of 'ocean coupling'**

*Done.*

Both changed to 'atmosphere-ocean coupling'.

**Section 2.3 p6 ln 6-ln10: I was a bit confused by this (DMS production) description, in particular there seems to be an error in the use of degradation/production? after reading through several papers, I now assume production should be changed do degradation in line 10 - or it should be 'detritus production' of opal/CaCO3. In Six et al. 2013 the k refers to sulphur to carbon ratios in cells of opal/CaCO3. In Six and Maier-Reimer 2006 the gammas to degradation rates of opal/CaCO3; not clear to me what is meant in Eq. 2 please double-check and revise Section 2.3**

*Thank you for noticing these inconsistencies. We have updated and expanded Section 2.3, as follows:*

- Changed the reference from Six and Maier-Reimer (2006) to Kloster et al. (2006), which is the updated parameterization currently used in our model.
- Revised equation (1) and provided an improved description for source and sink terms associated with detritus export production, bacterial consumption, and photolysis.
- Revised equation (2), through renaming the different terms.
- Added equations for loss terms due to bacterial consumption and photolysis are now added in the revision.

**p6 Eq (1) capital P in Phy irritating - I suggest to use 'phyto' to avoid mixing up with physics; also, is there a reason why some terms carry the 'DMS', others not?**

*We follow your suggestion. Some terms initially carry 'DMS' as it is computed as a function of the DMS concentration in the model.*

The term 'Phy' has been replaced with simply 'prod' (for production) and 'DMS's are removed in all terms in the revised eq. (1).

**p9 ln16/18: perhaps the ref. to Fig. 1 could be moved to the end of the para, or even toward the end of the section, otherwise one is trying to connect the following statements to Fig. 1**

We agree.

Moved the sentence referencing Fig. 1 to the end of the 3$^{rd}$ paragraph in the same section.

**p10 ln25 I presume what is meant here is that preformed phosphate can be used to estimate the organic carbon pump, and preformed alkalinity to estimate the inorganic carbon pump? As the sentence stands now, either can be used to estimate both pumps.**
**–> Preformed phosphate can be used to quantify the organic (), and preformed alkalinity to quantify the inorganic () carbon pump ().**

*Correct. We have revised the sentence to:*

"Following Bernardello et al. (2014), $PO_4^{pre}$ is used to quantify the organic (soft tissue, $DIC^{soft}$), whereas both $PO_4^{pre}$ and $ALK^{pre}$ are used to quantify the inorganic (carbonate, $DIC^{carb}$), biologically-mediated carbon pump (Eqs. 12-14)".

**p10 ln26 (Eqs. 10-12; Bernardello et al.) is a bit misleading, perhaps (Eqs. 10-12, based on Bernardello et al. ) is better suited (the corresponding eqs. are 1-4 in Bernardello e al. 2014)**

We agree. The sentence has been revised (see previous response).

**p11 eq.12 I do not quite follow how eq. 12 is derived (where does the +1 originate from?) +1 not in Bernardello et al. 2014 Eq. 4**

*In our updated code, we have assumed that during organic production and remineralization, both changes in nitrate and phosphate alter the proton concentration and therefore change the alkalinity (see also Section 3.1.3 of Paulmier et al., 2009). We have included the following statement in the revision:*

Equation 14 is slightly different than that in Bernardello et al. (2014), $r_{N:P} + 1$ instead of $r_{N:P}$, because, in our updated code, we do not neglect the contribution of phosphate to alkalinity changes during biological production and remineralization. For instance, both nitrate and phosphate produced during organic remineralization increase the concentration of proton and therefore reduce the alkalinity (see also Section 3.1.3 of Paulmier et al., 2009).

**p15 ln 18 'branched off into' is not a valid formulation. rephrase to sth like: the PI, control and historical exps are branched off from the spin-up. or 'the simulation is used as a starting point'....**

*We have rephrased the sentence as suggested to:*

"The end of the model spin-up is used as a starting point for …"

**p15 ln 20 see above**

*Rephrased to:*

"Simulation (3) is furthermore used as a starting point for …. "

**p16 ln24 rephrase 'at subsurface 500 m'**

*Rephrased to:*

"At 500 m depth, …"

**p16 ln25/26 'improvements ... in deviation' does not sound good.**
**–> improvements in agreement, better agreement**

*Rephrased to:*

"… shows better agreement with data throughout …"

**p18 ln 5 what is meant by: the simulated .... concentrations are mostly confined to the upper 1 km?**

*It was meant to describe that most of the high CFC-11 values are mostly simulated between the surface and 1 km depth. We have revised the sentence to:*

"Similar to the observations, high values of CFC-11 are generally simulated in the upper 1 km in both the Atlantic and Pacific, with the exception of the North Atlantic … "

**p23 ln 31 ct Consistent with the lower oceanic (than atmospheric) pCO2 partial pressure (I guess partial pressure is meant here, since the oceanic growth rates are not shown in Fig. 20)**

*Yes, thank you. We have rephrased the sentence to:*

"Consistent with the lower surface ocean (than the atmospheric) $pCO_2$, the ocean carbon uptake continues to increase …"

**Fig. 2 since the panels are labelled a-d, better use 'a,c' in the caption instead of 'left', etc**

*Done.*

Replaced left, right, top, and bottom with 'a,c', 'b,d', 'a,b', and 'c,d'.

**Fig. 9 is never discussed in the text**

*In the revised section 5.6, towards the end, we have added the following (previously missing) discussion on Silicate (Fig. 9):*

"The silicate spatial distribution in NorESM2 resembles closely the phosphate distribution (Figs. 9a-c and 7a-c), with similar biases across the vertical sections of Atlantic and Pacific (Figs. 9e-f and 7e-f). At high-latitudes, NorESM2 overestimates the surface silicate concentration (Fig. 9d), which could be attributed to the reduced opal export sinking speed relative to our earlier model version (30 instead of previously 60 m day$^{-1}$)."

**Fig. 14 I am a bit sceptical about the term 'Southern Ocean' for latitudes between 20 deg S and 40 deg S. Why not just call it southern hemisphere mid latitudes.**

*We followed the reviewer's suggestion.*

We have revised both figures 14 and 19 and renamed the region to "Mid-latitude southern hemisphere". The corresponding texts have been revised too.

**The content description of the supplementary material is full of errors.**

*The supplementary material has been revised.*

In addition, we have also reduced the number of supplemental figures (from 16 to 7; Figures S6 and S7 are new), since many of the figures (i.e., distributions of biogeochemical tracers, maps of limiting nutrient, surface $pCO_2$ and $CO_2$ fluxes, DMS concentration, and mean primary

production and transfer efficiency) from NorESM2-LM are qualitatively and visually very similar to those of NorESM2-MM (shown in the main manuscript).

**Technical errors**
**general: Check all occurrences of 'to allow for' (which means to consider s.th. when planning for s.th., see e.g. dictionary.cambridge.org) and replace by 'allows' or 'enables us to' or similar change all occurrences of 'insight to' to 'insight into' check for sgl/pl and past/present mismatches**

We have checked for and revised the above phrases as suggested.

**in the following 'ct' stands for 'correct to'**
**p1 ln 5 correct 'allow for'**

Corrected to: "allow for".

**p1 ln 7 riverine 'input'; p1 ln 7 'are recently' does not make sense -> have recently been ...**

The sentence has been revised to: "have recently been…".

**p2 ln 6 remove 'us' (who is 'us' here?) or better reformulate whole sentence**

Rephrased to: "ESM simulations can therefore be used to estimate historical carbon budgets and future carbon emission pathways under specified scenarios."

**p2 ln 8 remove 'us'**

Rephrased to: " … ESMs are state-of-the-art tools used to study …".

**p2 ln 13 ct ...hardware systems, higher resolution**

Done. "(… new hardware systems, higher resolution, etc.)".

**p2 ln 16 complex interplay between what and what?**

Replaced "… complex interplay of internal climate variability." with "… complex interactions with the internal climate variability."

**p2 ln 21 remove 'through'**

Done.

**p4 ln 3 check use of 'implications on' (correct: implications of sth. for sth.) should be 'consequences for', 'impact on', or similar**

Replaced with 'consequences for'.

**p4 ln 11 ct 'insight into the ocean's role' ....**

Done. "… insights into the ocean's …".

**p4 ln 26 ct ....'in Section 4.'**

Corrected.

**p4 ln 28 ct ....'in Section 6.'**

Corrected.

**p4 ln 32 delete 'for'**

Removed 'allows for a'.

**p5 ln 2 ct we also now apply (or we also applied)**

Revised to 'We also applied …'.

**p5 ln 29 ct 'closer to' (or better, 'which is within the range'....)**

Used 'which is within the range'.

**p6 ln 24 ct ... an early version of 'the' Global-NEWS model ...**

Added 'the'.

**p7 ln 15 ct .... and 'are' added to the nitrate pool .... (otherwise it is not clear if this is only assumed)**

Removed 'assumed to be bio-available and'.

**p7 ln 16 ct Particle export (without plural s)**

Corrected.

**p8 ln 1 ct ....interior biogeochemistry 'using' the different .... (not 'in')**

Replaced 'in' with 'using'.

**p8 Eq. 4 dot between mu and max misplaced**

Corrected.

**p8 ln 14/15 change to: 1.25 moles dissolved oxygen and 1 mole of alkalinity...**

Changed as suggested.

**p9 ln 11 ct where the strength 'of' this ....**

Corrected.

**p 11 Eq. (12) dot after 0.5 misplaced**

Corrected (now Eg. 14).

**p11 Eq. (13) remove leading dot**

Done (now Eq. 15).

**p11 ln24 insight into**

Corrected.

**p11 ln29 correct 'allow for'**

Rephrased to 'The inclusion of natural tracers provides substantial saving …'.

**p12 ln 5 remove 'to' before (ii)**

Done.

**p12 ln10 correct 'allow for'**

Rephrased to 'In order to allow comparison between different …'

**p13 ln 27 ct ... different parameterizations, (add s)**

Done.

**p14 ln 16 ct , as follows (add s)**

Done.

**p14 ln 25 ct 'a quasi-equilibrium state' (remove pl. s)**

Done.

**p14 ln 30 ct ...carbon counterpart of 'the' respective ...**

Done.

**p15 ln 1 ct ...applied 'to' organic ... (not 'for')**

Done.

**p15 ln 3 –> atmospheric pCO2**

Done.

**p16 ln 8 correct 'the the'**

Done.

**p16 ln11 either 'the majority' or 'the major part' (but not 'majority part')**

Used 'the majority of the mean …".

**p17 ln 29 ct 'In the subpolar North Atlantic'....**

Corrected.

**p17 ln 31 ct ....'a' deep MLD ....**

Done.

**p18 ln 5 delete 'depth' after upper 1 km**

The sentence has been rephrased: "Similar to the observations, high values of CFC-11 are generally simulated in the upper 1 km in both the Atlantic and the Pacific, …"

**p18 ln 16 ct .... material being ....**

Corrected.

**p18 ln 17 ct ... 'at' intermediate depth**

Corrected.

**p18 ln 18 ct ... 100 and 1500 m depth).**

Done.

**p18 ln 20 ct ... concentrations of all nutrients ...**

Replaced 'in' with 'of'.

**p18 ln 27 ct ....NADW as the main watermass ...**

Corrected.

**p19 ln 3 ct ...(see also Fig 8i). - check if 8i was meant, 8e is Atlantic NorESM2**

*Yes, we meant Fig. 8i (not 8e). Thank you.*

Corrected.

**p19 ln 8/9 ct: ...low levels.... limit the phytoplankton growth.**

Corrected.

**p19 ln 21 correct 'Fig. 9' to 'Fig. 12'**

*Thank you.*

Done.

**p19 ln 29 ct ... and at high latitudes during summer months.**

Done.

**p20 ln 3 ct ...the spring blooms....**

Done.

**p20 ln 4 delete 'during the boreal spring months' (redundant in sentence)**

Deleted.

**p20 ln 9 coastal areas (not grids)**

Corrected.

**p20 ln 20/21 move ref to Weber after 'biogeochemistry' ct ....still simulates too high transfer efficiencies...**

Both corrected.

**p20 ln 22 ct ...are comparable with observations.**

Done.

**p20 ln 26 correct 'Fig. 2c' to Fig. 2d; ct ....simulates a lower ...**

Both corrected.

**p21 ln 8 ct ...at the lower end...**

Done.

**p21 ln 24 ct As at the surface...**

Done. Replaced "As in the surface …" with "As at the surface …".

**p22 ln 12 ct ...translates into stronger carbon sinks...**

Done.

**p22 ln 20 ct of DIC-rich deep watermasses...**

Done.

**p22 ln 21 ct ...strong bias ... is considerably reduced, ....**

Corrected as suggested.

**p22 ln 22 ct ... is approximately reversed compared to observations:**

Corrected as suggested.

**p22 ln 26 ct Nevertheless, the ....**

Done.

**p23 ln 25 ct 0.7 oC, comparable to that from obs...**

Replaced " … relatively comparable with that from the …" with "comparable to that from …".

**p23 ln 26 ct the warming in esm-hist ....**

Done.

**p23 ln 30 ct (i.e., is lower than)...**

Done.

**p24 ln 7 ct For the 1980s and 1990s ...**

Done.

**p24 ln 9 correct allowing for**

Done.

**p24 ln 15 ct ...by the ocean for 1850-1994 and 1994-2011 is ....**

Done.

**p24 ln 27 correct allow for**

Corrected to 'which enable us to'.

**p24 ln 28 ct , and (iii) carbon isotopes that can be used e.g. ....**

Corrected.

**p25 ln 8 ct the equatorial Pacific OMZ**

Corrected.

**p25 ln 9 ct ...Southern Ocean, and the equatorial and North Pacific.**

Corrected.

**p25 ln 14 replace allowing for by 'resulting in'**

Done.

**p25 ln 20 ct simulates a considerable bias**

Corrected.

**p25 ln 25 ct the CO2-fluxes' seasonal cycle...**

Corrected.

**p25 ln 26 attention to (not of)**

Revised "attention of" to "attention to".

**p25 ln 30 ct ... depth of the equatorial Pacific ....**

Corrected.

**p25 ln 33 ct penetrating too far north..**

Corrected.

**p26 ln 1 ct biogeochemical components in ESMs...**

Corrected.

**p26 ln 2 ct the mean climatological state. However, .... (check if the references have to be moved)**

Checked and corrected.

**p26 ln 3 ct a lot of effort**

Corrected.

**p26 ln 5 replace allowed for by e.g., 'provided'**

Replaced 'allowed for additional constraints' with 'provided additional valuable constraints'

**p26 ln 15 correct allow for**

Replaced 'allow for efficient optimization of the current ecosystem parameterization'
with
'enable us to perform optimization of the current ecosystem parameterization more efficiently'.

**p26 ln 19 ct ..currently, we use fixed particulate organic carbon emissions ...**

Corrected.

**p26 ln 27 ct to investigate the sensitivity of ....**

Corrected.

**p26 ln 35 ct as the results of more complex model simulations**

Corrected.

**Table 1: ct ... ecosystem parameterisations that have been changed....**

Corrected.

**Table 2: ct ... simulation periods over which their climatological values have been averaged. ct Average of the three remote sensing products...**

Both phrases have been corrected.

**Table 3: ct Annual mean biology-related metrics... (not only primary production is listed)**

Agree, and corrected.

**Fig. 1 ct Blue depicts components, processes, .... (there is only one blue)**

Corrected.

**Fig. 3 caption: ct 'Differences between ... are ....**
**also Fig. 4, 7, 8, 9, 11, 13, 16, 17, 21, 22, 23**

Corrected for all mentioned Figures.

**Fig. 14 averaged over all months.**
**also Fig. 19**

Done.

**Fig. 20 ct relative to the 1850-1879 period, (b)... Green depicts results from the ....**
**simulation with NorESM2. The purple line in panel (c) represents .... (only one green, only**
**one esm-hist with NorESM2-LM)**

Corrected. Thank you.

---

## Author Comment (AC2) · 22 Mar 2020

We thank Dr. Julien Palmieri for his prompt and constructive review of our manuscript. In particular, Dr. Palmieri raised important comments regarding the carbon isotopes evaluation in our manuscript. We agree with him that the stand-alone ocean simulation used to produce the carbon isotope results are not directly comparable with the coupled ESM configuration runs, which are the main focus of the paper. However, we believe and hope Dr. Palmieri concur, that the carbon isotope feature in NorESM2 is an important novel component and would still fit in the current manuscript. In the revision, we have emphasized the difference in the experimental setups. We have also shown (in new Supplementary figures) that despite the differences in model configurations and resolutions, both the stand-alone and coupled ESM simulations share very similar quasi-equilibrium interior biogeochemistry states (as seen in the interior distributions of phosphate and DIC concentrations). In addition to this, we have also addressed all of Dr. Palmieri's comments (in **bold**) in the revised manuscript. Point-by-point (i) responses (in *italics*) and/or (ii) description of changes made in the revision (in blue) are detailed below. Changes in the revision can also be seen in the attached revised manuscript with highlighted changes (red and blue depicts older and revised version, respectively).

**The paper mainly describes the improvements made to the model since the previous version used for the CMIP5 exercise, comparing both to observation, and showing that the new version is mainly better the its predecessor. Some difficulties come from this presentation choice as sometime the authors consider that some description being made in the NorESM1 description papers, the reader is assumed to know them. In results, some basic description are almost missing. For example, the ocean grid description of which we only know it is tripolar.**

*We understand the reviewer concern and have added more basic description of the model, e.g., information on the ocean grid has been added (see also below). Indeed, the ocean carbon cycle model in NorESM2 has gone through several update iterations with respective documentations (e.g., Tjiputra et al., 2013: Schwinger et al., 2016). In order to avoid overlaps, we have opted to focus the current paper on key improvements in process representations and parameterizations. Improvements related to this comment can also be seen further below.*

We have added the horizontal and vertical grid information in the last paragraph of Sect. 2.1.

**Apart some little corrections of that kind, listed bellow, which should hopefully help to improve the paper, i have one concern about the isotope run. The isotopes are run on an ocean-only run, with a lower resolution grid than the coupled run, and a different atmosphere to force the run. I don't think this run should be used to evaluate the coupled run, unless the authors show that both model DIC steady states are comparable, or mention in the isotope results paragraph that these results are only informative, because from a different run. This needs - at least - to be reminded to the reader. Written as it is, the isotope results are in the middle of historical and esm-hist results. The reader can easily misunderstand and think they all are from the same simulation.**

*Thank you for this thoughtful comment. Even though the isotope run was configured differently, we still think that the results of our isotope run represent an important advancement to the model*

*and would still fit in the current manuscript. We agree with Dr. Palmieri that a clearer separation is needed.*

We have moved the discussion on the isotope results toward the end and reminded again the reader that this is not from a coupled configuration run, in contrast to the other variables. Following your suggestion, we have also shown the interior DIC (and phosphate) concentrations under the preindustrial state (see also below) from the ocean-only simulation.

**Overall, i have positive feelings about this paper. Most comments are minor like rephrasing or asking for missing details, what should translate in minor revision.**

*Thank you very much for all the positive and constructive feedbacks.*

**P2 l1 : rephrasing : absorber of heat and the greenhouse gas CO2 to absorber of heat and of the greenhouse gas CO2 or absorber of heat and CO2 greenhouse gas ?? But the one in the paper sounds weird to my ears.**

We have rephrase this to: "… absorber of heat and of the greenhouse gas CO2 …".

**P2 l30 to P3 l3 : Is there an equivalent paper to this one for NorESM1? It seems there is no reference description paper for the version 1 of the model. You list different papers of different development stages, but which one is the one that would describe the version of NorESM1 you use in this paper?**

*Yes, there is a model description paper for NorESM1 (Tjiputra et al., 2013).*

we have clarified in the revision, i.e., towards the end of this paragraph, that the comparison presented here is relative to the version described in Tjiputra et al. (2013).

**P3 l3 : "which will contribute to CMIP6". You can probably write it at present tense.**

*We agree.* Rephrased to "… which contributes to CMIP6."

**P3 l24-33 : This can not replace having 2 separate runs. Which nutrient does the biology see ? How important could that be within the scenario ? (PI carbonate system evolves with biology that feels high CO2 world, pH,... ) probably not important for historical runs. that's a problem we faced within OMIP, and we could not have a definite answer (should run both to be sure...).**

*The reviewer is correct that the enhanced carbon sinks alter the carbonate chemistry in the model and, in turn, the dissolutions of $CaCO_3$ in our model. Aside from this, the biological formulation such as phytoplankton and nutrient dynamics are not affected by the high $CO_2$ (or other scenario runs). To address this, we have also added respective 'natural' components of alkalinity and $CaCO_3$ as described in subsection '3.2 Natural inorganic carbon tracers'. To further clarify this and the 'non-effect' on biological production, we have added the following statements in subsection 3.2:*

"This is because the anthropogenic $CO_2$ uptake alters the carbonate system and therefore the dissolution of $CaCO_3$. We also note that anthropogenic carbon does not influence the biological production (e.g., nutrient concentrations and phytoplankton growth rate) in our model."

**P3 l34 to p4 l6 : 2 remarks : you say " The only external source [in NorESM1] is through atmospheric nitrogen fixation" you mean there is no dust (iron/Si/P) deposition in NorESM1? from this paragraph it sounds like you only add riverine nutrients, but you talk of dust somewhere else... could you clarify? Also, what is included in riverine nutrients? (actually you answer later on - maybe add a "see paragraph 2.4 for details")**

*You're right, we do have atmospheric deposition of dust (converted into dissolved iron).*

We have corrected this error in the revised manuscript. In addition, in the last paragraph of the introduction, we have emphasized that "… more detailed descriptions of all improvements in biogeochemical processes are described in Section 2."

**2.1 : you give several details in this paragraph but we miss some details of the ocean grid, like the basic ones: resolution, number of vertical levels,...**

We have added the horizontal and vertical grid information in the last paragraph of Sect. 2.1.

**p7 l4-7 : i don't understand what you explain there to get the riverine fluxes of DOC and POC, maybe you could try to make it clearer ?**

*We have rephrased the sentence as follows:*

"In our model, the organic form of dissolved carbon (DOC) is connected to nutrients through the Redfield ratio (C:N:P=122:16:1), and therefore other forms of dissolved organic matters (e.g., DON, and DOP) are not explicitly simulated. Since Global-NEWS provides estimates of dissolved organic matter in carbon, nitrogen, and phosphate forms ($DOC_{riv}$, $DON_{riv}$, and $DOP_{riv}$) separately, only the minimum of the three riverine dissolved organic constituents is added to the DOC term in the model (i.e., $DOC=DOC+min(DOC_{riv}, r_{C:N}*DON_{riv}, r_{C:P}DOP_{riv})$). Any excess or remaining organic matter of the three constituents is then added to the corresponding inorganic pools (DIC, $NO_3$ or $PO_4$). The same concept also applies to riverine inputs of particulate organic carbon (POC) (see also Bernard et al., 2011)."

**P9 paragraph 2.9 – So aerial dust deposition was already in NorESM1 - don't forget to add it P3.**

This information has now been added to Page 4 (introduction).

**P9 l11 - typo : strength *of* this**

Corrected.

**P10 l12 : "during the spin-up" might be misleading. Maybe "for the spin-up" or "to start the spin-up" or just "initialized to zero" might be enough.**

Rephrased to "at the beginning of the spin-up".

**P12 l14 : "... and CaCO3 formation, but the latter is neglected in our implementation". Why ? any idea of the impact ?**

*This was addressed further in subsection 3.3.2:*

*"Isotope equilibrium fractionation during CaCO₃ formation increases $\delta^{13}C$ of CaCO₃ and decreases seawater of $\delta^{13}C$. Nevertheless, the fractionation effect during CaCO₃ formation is relatively small compared to the effects of air-sea gas exchange and photosynthesis and therefore is often omitted in modelling (Lynch-Stieglitz et al., 1995; Schmittner et al., 2013)."*

*We have further revised this statement to:*

"Isotope equilibrium fractionation during CaCO₃ formation increases $\delta^{13}C$ of CaCO₃ and therefore depletes seawater of $^{13}C$ (thereby lowering seawater $\delta^{13}C$). Nevertheless, the fractionation effect during CaCO₃ formation is relatively small (i.e., in the order of -2 to +3‰, depending on species and environmental conditions; Grossman and Ku, 1986; Ziveri et al., 2003; Zeebe and Wolf-Gladrow, 2001) compared to the effects of air-sea gas exchange and photosynthesis. Therefore, fractionation during CaCO₃ formation is commonly omitted in modelling studies (e.g., Lynch-Stieglitz et al., 1995; Schmittner et al., 2013)."

**P14 l24 to P15 l2 : The method to initialize the C isotopes is very clever, and must save a lot of computing time. But i wonder the impact of initializing them this way. Don't you introduce bias compare to a proper model integration? Do you note drifts in the isotopes concentration during the historical period?**

*Yes, the method will introduce a bias, but avoids a long-term drift towards the same bias. Under this configuration, the drifts for $DI^{12}C$ and $DI^{13}C$ at the end of spin-up are in the order of 0.005% per 1000 years. Secondly, we only performed a spin-up stand-alone ocean simulation where the carbon isotopes are activated (hence no 'historical' simulation). The results shown in the paper represent values at the end of the 5000 years spin-up which is a close representation of the preindustrial state.*

We have clarified that only a preindustrial spin-up was performed at the end of revised Section 4 and beginning of Section 5.13.

**P15 l19-23 : might need a bit of rephrasing here. explanation about (3) are a bit confusing. i would suggest to call it emission-driven simulation as (1) and (2) are also "esm" runs. Or if you absolutely want to call it esm-... explain straight what it is like " (3) esm- experiments, the atmospheric CO2 is prognosticaly computed from ocean-atmosphere and land-atmosphere CO2 -fluxes, as well as from prescribed anthropogenic emissions (for the historical period). (3) is composed of a esm-piControl-spinup simulation (100 model years)**

**which is then branched off into (3a) an esm-piControl simulation (for 250 years), and (3b) a transient esm-hist (years 1850-2014) simulation." But calling this experiment "esm" is confusing for the reader.**

*The use of the 'esm-'naming in our emission-driven simulations follows the standard CMIP6 protocol (Eyring et al., 2016), so we would like to keep it.*

As suggested by both reviewers, we have revised the text describing experiments (3), to make it clearer and easier to understand.

**P15 l30 : "The atmospheric forcing of the spin-up is the CORE normal year forcing (Large and Yeager, 2004), which represents a climatological mean year with a smooth transition between end and start of the year" Why did you use the CORE climatology ? that's a surprising choice, it removes all substantial link between your C-isotope/ocean- only run and the esm. Using CORE remove all potential isotope insights on the esm runs... there are other ways that could provide this insight possibility: you could extract atm fields from the end of the spin-up and use them as forcing field for this experiment, for example. Using CORE, the ocean dynamics will be different from the coupled, the spin-up steady state will have nothing to do with the esm's steady state. Already changing the ocean resolution affect the dynamic, but resolution plus forcing completely change the ocean. I don't think you can use the isotopes to evaluate the esm run with such a different set-up.**

*Yes, we completely agree with you that the stand-alone ocean simulation with CORE normal year forcing (in addition to the different spatial resolution) is not directly comparable with the simulation performed in a coupled ESM configuration. The main intention was simply to describe the carbon isotope implementation and furthermore we think that a first evaluation can still be done with a, computationally cheap, CORE forcing setup (e.g., the large-scale circulation pattern is not completely different).*

We have emphasized this in the revised Section 5.1.3, and in addition, added two supplemental figures S6 and S7, which illustrate the similarity between the large-scale pattern of interior nutrient (phosphate) and DIC in the stand-alone ocean configuration and that in the coupled ESM setup.

**P15 l30 : Subsequent question concerning the isotope run. What happens after the spin-up? Is there anything special to run the historical period? Do you change the atm forcing field (i really wished you used atmospheric fields extracted from the coupled historical run)? or do you simply follow the historical atmospheric CO2 records for the last ~150years ?**

*As stated above, the simulation with carbon isotopes does not include any historical simulation (only preindustrial spin-up) and was meant to assess the performance of the new carbon isotopes under the preindustrial climate.*

In the revision, we have clarified this (i.e., in Sections 4 and 5.13).

**5.3 Salinity - P17 - no ref to Fig. 4**

References to Fig. 4 have been added.

**5.6 Nutrients - P18 - No ref to Fig. 7 and 9 The fig 2 is understandably well discussed as it is a very instructive figure, but all following figures are almost forgotten, and sometimes not introduced at all.**

We have included references to Figs. 7-9 in the revised subsection '5.6 Nutrients', as suggested. In the revised manuscript, we have also included more direct references to specific Figures, in addition to Fig. 2.

**P19 l3 - if you refer to NorESM1, you should point to fig 8h and 8i**

*The previous sentence was referring to the Pacific Ocean, so a reference to Fig. 8h is not necessary.*

We have, nevertheless, added a reference to NorESM2 (Fig. 8f) in the preceding sentence.

**5.7 Dissolved oxygen – P19 - No ref to Fig. 11 .**

We have included references to Fig. 11 in the revision.

**P19 l21 - Fig. 9 is the silicic acid; AOU is Fig. 12**

Corrected.

**5.12 - P22 : The Analysis done here are really great, but coming from a stand alone run forced with a different forcing field, these results do not inform about the main NorESM2 MM and LM runs i am afraid.**

*Although these simulations were done in a stand-alone ocean configuration, we think that the implementation and carbon isotopes capability in NorESM2 is novel and have decided to include them in this manuscript. While this simulation is not directly comparable with those coupled setups done with NorESM2-LM and NorESM2-MM, their interior biogeochemical mean states share very similar features.*

To support this, we have added two new figures of phosphate and DIC from the stand-alone run (in the supplemental materials, Figs. S13 and S14) and referred to them in the main manuscript.

**5.12 and 5.13 - double-check the figure order, you switched both section figures.**

Checked and followed your suggestion.

We have switched subsections 5.12 and 5.13 such that the carbon isotopes discussions are now presented last. We have also reminded the readers that the carbon isotopes analysis is based on ocean-only, and not coupled ESM, configuration.

**Fig.1 some features are missing. For instance the riverine inputs (with the different materials they contain). Also the silicic acid is missing.**

*Thank you for noticing this.*

We have revised Fig. 1, adding arrows to indicate riverine inputs. Silicate has been added in the nutrient box. Figure caption has been revised accordingly.

**Fig. 10 : The modeled Diatoms are never Si limited ? You say biological productivity, how is that done ? as both phyto are not necessarily limited by the same nutrient. Might be good to say 2 words in the figure caption about how you calculate that.**

*In our model, diatoms are not explicitly modeled. We only implement one prognostic bulk phytoplankton class. Silicate concentration in the model is used to determine the opal export, as stated in subsection 2.3. The bulk phytoplankton growth rate formulation is documented in Schwinger et al. (2016).*

We have revised subsection 5.6 (last paragraph) to reflect this information:

"In NorESM, the bulk phytoplankton growth rate is limited by multiple nutrients (i.e., phosphate, nitrate, and dissolved iron), in addition to temperature and light (see also Section 2.3.2 of Schwinger et al. (2016). The model does not explicitly simulate diatom and calcifier classes."

As suggested, we have also revised Fig. 10 caption to inform how the limiting nutrient is determined.

**Fig. 13 : I have difficulties to realize what's going on with contours superposed to colormaps and showing different things. It is OK-ish on vertical sections, but on global horizontal maps it is difficult to appreciate the differences, and spot what we should see. it is not straight forward. could you add a colored pictures for NorESM1 so it is not superposed as contour, for fig 13, 15, 18. I would ask to do that for all fig of that kind, but that would mean almost all pictures... i really find it difficult to interpret that way. it save space, but doesn't make it simpler for the reader, i think.**

*We have followed your suggestion.*

We have revised Figs. 13, 15, and 18 and removed the contours in these figures. Respective captions have been revised to reflect these changes. For Fig. 15, DMS was not simulated in NorESM1-ME. For $CO_2$ fluxes, since the seasonal cycles of NorESM1-ME in selected regions are already shown in Fig. 19, we decided not to include them again in Fig. 18 to reduce the number of figures. A map of $CO_2$ fluxes from NorESM1-ME is also already available in Tjiputra et al. (2013).

**Fig 14. and all alike : what is the difference between the solid red and dotted red lines ? i guess the dotted one is NorESM-LM but it is not written in the caption. you probably can remove the dotted line as there is no reference to it at all.**

*You're correct. Those are values for NorESM2-LM. We agree.*

We have removed the red dotted-lines.